# ON-POLICY MODEL ERRORS
# IN REINFORCEMENT LEARNING

**Lukas P. Fröhlich**[*]
Institute for Dynamic Systems and Control
ETH Zürich
Zurich, Switzerland
`lukasfro@ethz.ch`

**Maksym Lefarov**
Bosch Center for Artificial Intelligence
Renningen, Germany
`Maksym.Lefarov@de.bosch.com`

**Melanie N. Zeilinger**
Institute for Dynamic Systems and Control
ETH Zürich
Zurich, Switzerland
`mzeilinger@ethz.ch`

**Felix Berkenkamp**
Bosch Center for Artificial Intelligence
Renningen, Germany
`Felix.Berkenkamp@de.bosch.com`

## ABSTRACT

Model-free reinforcement learning algorithms can compute policy gradients given sampled environment transitions, but require large amounts of data. In contrast, model-based methods can use the learned model to generate new data, but model errors and bias can render learning unstable or suboptimal. In this paper, we present a novel method that combines real-world data and a learned model in order to get the best of both worlds. The core idea is to exploit the real-world data for on-policy predictions and use the learned model only to generalize to different actions. Specifically, we use the data as time-dependent on-policy correction terms on top of a learned model, to retain the ability to generate data without accumulating errors over long prediction horizons. We motivate this method theoretically and show that it counteracts an error term for model-based policy improvement. Experiments on MuJoCo- and PyBullet-benchmarks show that our method can drastically improve existing model-based approaches without introducing additional tuning parameters.

## 1 INTRODUCTION

Model-free reinforcement learning (RL) has made great advancements in diverse domains such as single- and multi-agent game playing (Mnih et al., 2015; Silver et al., 2016; Vinyals et al., 2019), robotics (Kalashnikov et al., 2018), and neural architecture search (Zoph & Le, 2017). All of these model-free approaches rely on large numbers of interactions with the environment to ensure successful learning. While this issue is less severe for environments that can easily be simulated, it limits the applicability of model-free RL to (real-world) domains where data is scarce.

Model-based RL (MBRL) reduces the amount of data required for policy optimization by approximating the environment with a learned model, which we can use to generate simulated state transitions (Sutton, 1990; Racanière et al., 2017; Moerland et al., 2020). While early approaches on low-dimensional tasks by Schneider (1997); Deisenroth & Rasmussen (2011) used probabilistic models with closed-form posteriors, recent methods rely on neural networks to scale to complex tasks on discrete (Kaiser et al., 2020) and continuous (Chua et al., 2018; Kurutach et al., 2018) action spaces. However, the learned representation of the true environment always remains imperfect, which introduces approximation errors to the RL problem (Atkeson & Santamaria, 1997; Abbeel et al., 2006). Hence, a key challenge in MBRL is *model-bias*; small errors in the learned models that compound over multi-step predictions and lead to lower asymptotic performance than model-free methods.

To address these challenges with both model-free and model-based RL, Levine & Koltun (2013); Chebotar et al. (2017) propose to combine the merits of both. While there are multiple possibilities to

---

[*]Work done partially while at the Bosch Center for Artificial Intelligence.

combine the two methodologies, in this work we focus on improving the model's predictive state distribution such that it more closely resembles the data distribution of the true environment.

**Contributions**    The main contribution of this paper is on-policy corrections (OPC), a novel hyperparameter-free methodology that uses on-policy transition data on top of a separately learned model to enable accurate long-term predictions for MBRL. A key strength of our approach is that it does not introduce any new parameters that need to be hand-tuned for specific tasks. We theoretically motivate our approach by means of a policy improvement bound and show that we can recover the true state distribution when generating trajectories on-policy with the model. We illustrate how OPC improves the quality of policy gradient estimates in a simple toy example and evaluate it on various continuous control tasks from the MuJoCo control suite and their PyBullet variants. There, we demonstrate that OPC improves current state-of-the-art MBRL algorithms in terms of data-efficiency, especially for the more difficult PyBullet environments.

**Related Work**    To counteract model-bias, several approaches combine ideas from model-free and model-based RL. For example, Levine & Koltun (2013) guide a model-free algorithm via model-based planning towards promising regions in the state space, Kalweit & Boedecker (2017) augment the training data by an adaptive ratio of simulated transitions, Talvitie (2017) use 'hallucinated' transition tuples from simulated to observed states to self-correct the model, and Feinberg et al. (2018); Buckman et al. (2018) use a learned model to improve the value function estimate. Janner et al. (2019) mitigate the issue of compounding errors for long-term predictions by simulating short trajectories that start from real states. Cheng et al. (2019) extend first-order model-free algorithms via adversarial online learning to leverage prediction models in a regret-optimal manner. Clavera et al. (2020) employ a model to augment an actor-critic objective and adapt the planning horizon to interpolate between a purely model-based and a model-free approach. Morgan et al. (2021) combine actor-critic methods with model-predictive rollouts to guarantee near-optimal simulated data and retain exploration on the real environment. A downside of most existing approaches is that they introduce additional hyperparameters that are critical to the learning performance (Zhang et al., 2021).

In addition to empirical performance, recent work builds on the theoretical guarantees for model-free approaches by Kakade & Langford (2002); Schulman et al. (2015) to provide guarantees for MBRL. Luo et al. (2019) provide a general framework to show monotonic improvement towards a local optimum of the value function, while Janner et al. (2019) present a lower-bound on performance for different rollout schemes and horizon lengths. Yu et al. (2020) show guaranteed improvement in the offline MBRL setting by augmenting the reward with an uncertainty penalty, while Clavera et al. (2020) present improvement guarantees in terms of the model's and value function's gradient errors.

Moreover, Harutyunyan et al. (2016) propose a similar correction term as the one introduced in this paper in the context of off-policy policy evaluation and correct the state-action value function instead of the transition dynamics. Similarly, Fonteneau et al. (2013) consider the problem of off-policy policy evaluation but in the batch RL setting and propose to generate 'artificial' trajectories from observed transitions instead of using an explicit model for the dynamics.

A related field to MBRL that also combines models with data is iterative learning control (ILC) (Bristow et al., 2006). While RL typically focuses on finding parametric feedback policies for general reward functions, ILC instead seeks an open-loop sequence of actions with fixed length to improve state tracking performance. Moreover, the model in ILC is often derived from first principles and then kept fixed, whereas in MBRL the model is continuously improved upon observing new data. The method most closely related to RL and our approach is optimization-based ILC (Owens & Hätönen, 2005; Schöllig & D'Andrea, 2009), in which a linear dynamics model is used to guide the search for optimal actions. Recently, Baumgärtner & Diehl (2020) extended the ILC setting to nonlinear dynamics and more general reward signals. Little work is available that draws connections between RL and ILC (Zhang et al., 2019) with one notable exception: Abbeel et al. (2006) use the observed data from the last rollout to account for a mismatch in the dynamics model. The limitations of this approach are that deterministic dynamics are assumed, the policy optimization itself requires a line search procedure with rollouts on the true environment and that it was not combined with model learning. We build on this idea and extend it to the stochastic setting of MBRL by making use of recent advances in RL and model learning.

---

**Algorithm 1** General Model-based Reinforcement Learning

---

1: **for** $n = 1, \dots$ **do**
2:      **for** $b = 1, \dots, B$ **do**
3:          Rollout policy $\pi_n$ on environment and store transitions $\mathcal{D}_n^b = \{(\hat{\mathbf{s}}_t^{n,b}, \hat{\mathbf{a}}_t^{n,b}, \hat{\mathbf{s}}_{t+1}^{n,b})\}_{t=0}^{T-1}$
4:      $\tilde{p}_n(\mathbf{s}_{t+1} \mid \mathbf{s}_t, \mathbf{a}_t)$: Learn a global dynamics model given all data $\mathcal{D}_{1:n} = \bigcup_{i=1}^n \bigcup_{b=1}^B \mathcal{D}_i^b$
5:      $\theta_{n+1} = \theta_n + \alpha \nabla \tilde{\eta}_n$: Optimize the policy based on the model $\tilde{p}$ with any RL algorithm

---

## 2 PROBLEM STATEMENT AND BACKGROUND

We consider the Markov decision process (MDP) $(\mathcal{S}, \mathcal{A}, p, r, \gamma, \rho)$, where $\mathcal{S} \subseteq \mathbb{R}^{d_\mathcal{S}}$ and $\mathcal{A} \subseteq \mathbb{R}^{d_\mathcal{A}}$ are the continuous state and action spaces, respectively. The unknown environment dynamics are described by the transition probability $p(\mathbf{s}_{t+1} \mid \mathbf{s}_t, \mathbf{a}_t)$, an initial state distribution $\rho(\mathbf{s}_0)$ and the reward signal $r(\mathbf{s}, \mathbf{a})$. The goal in RL is to find a policy $\pi_\theta(\mathbf{a}_t \mid \mathbf{s}_t)$ parameterized by $\theta$ that maximizes the expected return discounted by $\gamma \in [0, 1]$ over episodes of length $T$,

$$\eta = \mathbb{E}_{\mathbf{s}_{0:T}, \mathbf{a}_{0:T}} \left[ \sum_{t=0}^T \gamma^t r(\mathbf{s}_t, \mathbf{a}_t) \right], \quad \mathbf{s}_{t+1} \sim p(\mathbf{s}_{t+1} \mid \mathbf{s}_t, \mathbf{a}_t), \quad \mathbf{s}_0 \sim \rho, \quad \mathbf{a}_t \sim \pi_\theta(\mathbf{a}_t \mid \mathbf{s}_t). \quad (1)$$

The expectation is taken with respect to the trajectory under the stochastic policy $\pi_\theta$ starting from a stochastic initial state $\mathbf{s}_0$. Direct maximization of Eq. (1) is challenging, since we do not know the environment's transition model $p$. In MBRL, we learn a model for the transitions and reward function from data, $\tilde{p}(\mathbf{s}_{t+1} \mid \mathbf{s}_t, \mathbf{a}_t) \approx p(\mathbf{s}_{t+1} \mid \mathbf{s}_t, \mathbf{a}_t)$ and $\tilde{r}(\mathbf{s}_t, \mathbf{a}_t) \approx r(\mathbf{s}_t, \mathbf{a}_t)$, respectively. Subsequently, we maximize the model-based expected return $\tilde{\eta}$ as a surrogate problem for the true RL setting, where $\tilde{\eta}$ is defined as in Eq. (1) but with $\tilde{p}$ and $\tilde{r}$ instead. For ease of exposition, we assume a known reward function $\tilde{r} = r$, even though we learn it jointly with $\tilde{p}$ in our experiments.

We let $\eta_n$ denote the return under the policy $\pi_n = \pi_{\theta_n}$ at iteration $n$ and use $\hat{\mathbf{s}}$ and $\hat{\mathbf{a}}$ for states and actions that are observed on the true environment. Algorithm 1 summarizes the overall procedure for MBRL: At each iteration $n$ we store $B$ on-policy trajectories $\mathcal{D}_n^b = \{(\hat{\mathbf{s}}_t^{n,b}, \hat{\mathbf{a}}_t^{n,b}, \hat{\mathbf{s}}_{t+1}^{n,b})\}_{t=0}^{T-1}$ obtained by rolling out the current policy $\pi_n$ on the real environment in Line 3. Afterwards, we approximate the environment with a learned model $\tilde{p}_n(\mathbf{s}_{t+1} \mid \mathbf{s}_t, \mathbf{a}_t)$ based on the data $\mathcal{D}_{1:n}$ in Line 4, and optimize the policy based on the proxy objective $\tilde{\eta}$ in Line 5. Note that the policy optimization algorithm can be off-policy and employ its own, separate replay buffer.

**Model Choices** The choice of model $\tilde{p}$ plays a key role, since it is used to predict sequences $\tau$ of states transitions and thus defines the surrogate problem in MBRL. We assume that the model comes from a distribution family $\mathcal{P}$, which for each state-action pair $(\mathbf{s}_t, \mathbf{a}_t)$ models a distribution over the next state $\mathbf{s}_{t+1}$. The model is then trained to summarize all past data $\mathcal{D}_{1:n} = \cup_{i=1}^n \cup_{b=1}^B \mathcal{D}_i^b$ by maximizing the marginal log-likelihood $\mathcal{L}$,

$$\tilde{p}_n^{\text{model}}(\mathbf{s}_{t+1} \mid \mathbf{s}_t, \mathbf{a}_t) = \arg\max_{\tilde{p} \in \mathcal{P}} \sum_{(\hat{\mathbf{s}}_t, \hat{\mathbf{a}}_t, \hat{\mathbf{s}}_{t+1}) \in \mathcal{D}_{1:n}} \mathcal{L}(\hat{\mathbf{s}}_{t+1}; \hat{\mathbf{s}}_t, \hat{\mathbf{a}}_t). \quad (2)$$

For a sampled trajectory index $b \sim \mathcal{U}(\{1, \dots, B\})$, sequences $\tau$ start from the initial state $\hat{\mathbf{s}}_0^{n,b}$ and are distributed according to $\tilde{p}_n^{\text{model}}(\tau \mid b) = \delta(\mathbf{s}_0 - \hat{\mathbf{s}}_0^{n,b}) \prod_{t=0}^{T-1} \tilde{p}_n^{\text{model}}(\mathbf{s}_{t+1} \mid \mathbf{s}_t, \mathbf{a}_t) \pi_\theta(\mathbf{a}_t \mid \mathbf{s}_t)$, where $\delta(\cdot)$ denotes the Dirac-delta distribution. Using model-data for policy optimization is in contrast to model-free methods, which only use observed environment data by replaying past transitions from a recent on-policy trajectory $b \in \{1, \dots, B\}$. In our model-based framework, this replay buffer is equivalent to the non-parametric model

$$\tilde{p}_n^{\text{data}}(\tau \mid b) = \delta(\mathbf{s}_0 - \hat{\mathbf{s}}_0^{n,b}) \prod_{t=0}^{T-1} \tilde{p}_n^{\text{data}}(\mathbf{s}_{t+1} \mid t, b), \text{ where } \tilde{p}_n^{\text{data}}(\mathbf{s}_{t+1} \mid t, b) = \delta(\mathbf{s}_{t+1} - \hat{\mathbf{s}}_{t+1}^{n,b}), \quad (3)$$

where we only replay observed transitions instead of sampling new actions from $\pi_\theta$.

## 3 ON-POLICY CORRECTIONS

In this section, we analyze how the choice of model impacts policy improvement, develop OPC as a model that can eliminate one term in the improvement bound, and analyze its properties. In the following, we drop the $n$ sub- and superscript when the iteration is clear from context.

### 3.1 POLICY IMPROVEMENT

Independent of whether we use the data directly in $\tilde{p}^{\mathrm{data}}$ or summarize it in a world model $\tilde{p}^{\mathrm{model}}$, our goal is to find an optimal policy that maximizes Eq. (1) via the corresponding model-based proxy objective. To this end, we would like to know how a policy improvement $\tilde{\eta}_{n+1} - \tilde{\eta}_n \geq 0$ based on the model $\tilde{p}$, which is what we optimize in MBRL, relates to the true gain in performance $\eta_{n+1} - \eta_n$ on the environment with unknown transitions $p$. While the two are equal without model errors, in general the larger the model error, the worse we expect the proxy objective to be (Lambert et al., 2020). Specifically, we show in Appendix B.1 that the policy improvement can be decomposed as

$$\underbrace{\eta_{n+1} - \eta_n}_{\text{True policy improvement}} \geq \underbrace{\tilde{\eta}_{n+1} - \tilde{\eta}_n}_{\text{Model policy improvement}} - \underbrace{|\eta_{n+1} - \tilde{\eta}_{n+1}|}_{\text{Off-policy model error}} - \underbrace{|\eta_n - \tilde{\eta}_n|}_{\text{On-policy model error}}, \quad (4)$$

where a performance improvement on our model-based objective $\tilde{\eta}$ only translates to a gain in Eq. (1) if two error terms are sufficiently small. These terms depend on how well the performance estimate based on our model, $\tilde{\eta}$, matches the true performance, $\eta$. If the reward function is known, this term only depends on the model quality of $\tilde{p}$ relative to $p$. Note that in contrast to the result by Janner et al. (2019), Eq. (4) is a bound on the policy improvement instead of a lower bound on $\eta_{n+1}$.

The first error term compares $\eta_{n+1}$ and $\tilde{\eta}_{n+1}$, the performance estimation gap under the optimized policy $\pi_{n+1}$ that we obtain in Line 5 of Algorithm 1. Since at this point we have only collected data with $\pi_n$ in Line 3, this term depends on the generalization properties of our model to new data; what we call the *off-policy model error*. For our data-based model $\tilde{p}^{\mathrm{data}}$ that just replays data under $\pi_n$ independently of the action, this term can be bounded for stochastic policies. For example, Schulman et al. (2015) bound it by the average KL-divergence between $\pi_n$ and $\pi_{n+1}$. For learned models $\tilde{p}^{\mathrm{model}}$, it depends on the generalization properties of the model (Luo et al., 2019; Yu et al., 2020). While understanding model generalization better is an interesting research direction, we will assume that our learned model is able to generalize to new actions in the following sections.

While the first term hinges on model-generalization, the second term is the *on-policy model error*, i.e., the deviation between $\eta_n$ and $\tilde{\eta}_n$ under the current policy $\pi_n$. This error term goes to zero for $\tilde{p}^{\mathrm{data}}$ as we use more on-policy data $B \to \infty$, since the transition data are sampled from the true environment, c.f., Appendix B.2. While the learned model is also trained with on-policy data, small errors in our model compound as we iteratively predict ahead in time. Consequently, the on-policy error term grows as $\mathcal{O}\big(\min(\gamma/(1-\gamma)^2, H/(1-\gamma), H^2)\big)$, c.f., (Janner et al., 2019) and Appendix B.3.

### 3.2 COMBINING LEARNED MODELS AND REPLAY BUFFER

The key insight of this paper is that the learned model in Eq. (2) and the replay buffer in Eq. (3) have opposing strengths: The replay buffer has low error on-policy, but high error off-policy since it replays transitions from past data, i.e., they are independent of the actions chosen under the new policy. In contrast, the learned model can generalize to new actions by extrapolating from the data and thus has lower error off-policy, but errors compound over multi-step predictions.

An ideal model would combine the model-free and model-based approaches in a way such that it retains the unbiasedness of on-policy generated data, but also generalizes to new policies via the model. To this end, we propose to use the model to predict how observed transitions would *change* for a new state-action pair. In particular, we use the model's mean prediction $\tilde{f}_n(\mathbf{s}, \mathbf{a}) = \mathbb{E}[\tilde{p}_n^{\mathrm{model}}(\,\cdot\mid\mathbf{s}, \mathbf{a})]$ to construct the joint model

$$\tilde{p}_n^{\mathrm{opc}}(\mathbf{s}_{t+1}\mid\mathbf{s}_t, \mathbf{a}_t, b) = \underbrace{\delta\big(\mathbf{s}_{t+1} - \hat{\mathbf{s}}_{t+1}^{n,b}\big)}_{\tilde{p}_n^{\mathrm{data}}(\mathbf{s}_{t+1}\mid t, b)} * \underbrace{\delta\big(\mathbf{s}_{t+1} - [\tilde{f}_n(\mathbf{s}_t, \mathbf{a}_t) - \tilde{f}_n(\hat{\mathbf{s}}_t^{n,b}, \hat{\mathbf{a}}_t^{n,b})]\big)}_{\text{Model mean correction to generalize to } \mathbf{s}_t, \mathbf{a}_t}, \quad (5)$$

where $*$ denotes the convolution of the two distributions and $b$ refers to a specific rollout stored in the replay buffer that was observed in the true environment. Given a trajectory-index $b$, $\tilde{p}_n^{\mathrm{opc}}$ in Eq. (5) transitions deterministically according to $\mathbf{s}_{t+1} = \hat{\mathbf{s}}_{t+1}^{n,b} + \tilde{f}_n(\mathbf{s}_t, \mathbf{a}_t) - \tilde{f}_n(\hat{\mathbf{s}}_t^{n,b}, \hat{\mathbf{a}}_t^{n,b})$, resembling the equations in ILC (c.f., Baumgärtner & Diehl (2020) and Appendix E). If we roll out $\tilde{p}_n^{\mathrm{opc}}$ along a trajectory, starting from a state $\hat{\mathbf{s}}_t^{n,b}$ and apply the recorded actions from the replay buffer, $\hat{\mathbf{a}}_t^{n,b}$, the correction term on the right of Eq. (5) cancels out and we have $\tilde{p}_n^{\mathrm{opc}}(\mathbf{s}_{t+1}\mid \hat{\mathbf{s}}_t^{n,b}, \hat{\mathbf{a}}_t^{n,b}, b) = \tilde{p}_n^{\mathrm{data}}(\mathbf{s}_{t+1}\mid t, b) = \delta(\mathbf{s}_{t+1} - \hat{\mathbf{s}}_{t+1}^{n,b})$. Thus OPC retrieves the true on-policy data

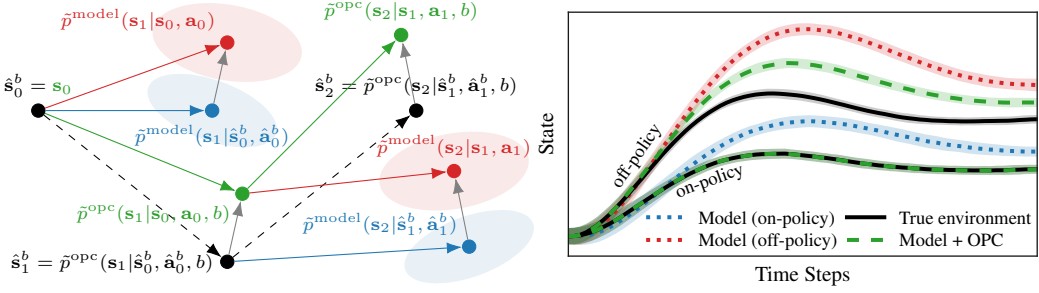

(a) Multi-step predictions with OPC for a given $n$.

(b) State distribution over time.

Figure 1: Illustration to compare predictions of the three models Eqs. (2), (3) and (5) starting from the same state $\hat{\mathbf{s}}_0^b$. In Fig. 1a, we see that on-policy, i.e., using actions $(\hat{\mathbf{a}}_0^b, \hat{\mathbf{a}}_1^b)$, $\tilde{p}^{\mathrm{data}}$ returns environment data, while $\tilde{p}^{\mathrm{model}}$ (blue) is biased. We correct this on-policy bias in expectation to obtain $\tilde{p}^{\mathrm{opc}}$. This allows us to retain the true state distribution when predicting with these models recursively (c.f., bottom three lines in Fig. 1b). When using OPC for off-policy actions $(\mathbf{a}_0, \mathbf{a}_1)$, $\tilde{p}^{\mathrm{opc}}$ does not recover the true off-policy state distribution since it relies on the biased model. However, the corrections generalize locally and reduce prediction errors in Fig. 1b (top three lines).

distribution independent of the prediction quality of the model, which is why we refer to this method as *on-policy corrections* (OPC). This behavior is illustrated in Fig. 1a, where the model (blue) is biased on-policy, but OPC corrects the model's prediction to match the true data. In Fig. 1b, we show how this affects predicted rollouts on a simple stochastic double-integrator environment. Although small on-policy errors in $\tilde{p}^{\mathrm{model}}$ (blue) compound over time, the corresponding $\tilde{p}^{\mathrm{opc}}$ matches the ground-truth environment data closely. Note that even though the model in Eq. (5) is deterministic, we retain the environment's stochasticity from the data in the transitions to $\hat{\mathbf{s}}_{t+1}$, so that we recover the on-policy aleatoric uncertainty (noise) from sampling different reference trajectories via indexes $b$.

When our actions $\mathbf{a}_t$ are different from $\hat{\mathbf{a}}_t^b$, $\tilde{p}^{\mathrm{opc}}$ still uses the data from the environment's transitions, but the correction term in Eq. (5) uses the learned model to predict how the next state *changes* in expectation relative to the prediction under $\hat{\mathbf{a}}_t^b$. That is, in Fig. 1a for a new $\mathbf{a}_t$ the model predicts the state distribution shown in red. Correspondingly, we shift the static prediction $\hat{\mathbf{s}}_{t+1}^b$ by the difference in means (gray arrow) between the two predictions; i.e., the *change in trajectory* by changing from $\hat{\mathbf{a}}_t^b$ to $\mathbf{a}_t$. Since we shift the model predictions by a time-dependent but constant offset, this does not recover the true state distribution unless the model has zero error. However, empirically it can still help with long-term predictions in Fig. 1b by shifting the model off-policy predictions (red) to the OPC predictions (green), which are closer to the environment's state distribution under the new policy.

### 3.3 THEORETICAL ANALYSIS

In the previous sections, we have introduced OPC to decrease the on-policy model error in Eq. (4) and tighten the improvement bound. In this section, we analyze the on-policy performance gap from a theoretical perspective and show that with OPC this error can be reduced independently of the learned model's error. To this end, we assume infinitely many on-policy reference trajectories, $B \to \infty$, which is equivalent to a variant of $\tilde{p}^{\mathrm{opc}}$ that considers $\hat{\mathbf{s}}_{t+1}^b$ as a random variable that follows the true environment's transition dynamics. While impossible to implement in practice, this formulation is useful to understand our method. We define the generalized OPC-model as

$$\tilde{p}_\star^{\mathrm{opc}}(\mathbf{s}_{t+1} \mid \mathbf{s}_t, \mathbf{a}_t, b) = \underbrace{p(\hat{\mathbf{s}}_{t+1} \mid \hat{\mathbf{s}}_t^b, \hat{\mathbf{a}}_t^b)}_{\text{Environment on-policy transition}} * \underbrace{\delta\left(\mathbf{s}_{t+1} - \left[\tilde{f}(\mathbf{s}_t, \mathbf{a}_t) - \tilde{f}(\hat{\mathbf{s}}_t^b, \hat{\mathbf{a}}_t^b)\right]\right)}_{\text{OPC correction term}}, \quad (6)$$

which highlights that it transitions according to the true on-policy dynamics conditioned on data from the replay buffer, combined with a correction term. We provide a detailed derivation for the generalized model in Appendix B, Lemma 4. With Eq. (6), we have the following result:

**Theorem 1.** *Let $\tilde{\eta}_\star^{\mathrm{opc}}$ and $\eta$ be the expected return under the generalized OPC-model Eq. (6) and the true environment, respectively. Assume that the learned model's mean transition function $\tilde{f}(\mathbf{s}_t, \mathbf{a}_t) = \mathbb{E}[\tilde{p}^{\mathrm{model}}(\mathbf{s}_{t+1} \mid \mathbf{s}_t, \mathbf{a}_t)]$ is $L_f$-Lipschitz and the reward $r(\mathbf{s}_t, \mathbf{a}_t)$ is $L_r$-Lipschitz. Further,*

if the policy $\pi(\mathbf{a}_t \mid \mathbf{s}_t)$ is $L_\pi$-Lipschitz with respect to $\mathbf{s}_t$ under the Wasserstein distance and its (co-)variance $\mathrm{Var}[\pi(\mathbf{a}_t \mid \mathbf{s}_t)] = \Sigma_\pi(\mathbf{s}_t) \in \mathbb{S}_+^{d_\mathcal{A}}$ is finite over the complete state space, i.e., $\max_{\mathbf{s}_t \in \mathcal{S}} \mathrm{trace}\{\Sigma_\pi(\mathbf{s}_t)\} \leq \bar{\sigma}_\pi^2$, then with $C_1 = \sqrt{2(1 + L_\pi^2)} L_f L_r$ and $C_2 = \sqrt{L_f^2 + L_\pi^2}$

$$|\eta - \tilde{\eta}_\star^{\mathrm{opc}}| \leq \frac{\bar{\sigma}_\pi}{1 - \gamma} d_\mathcal{A}^{\frac{1}{4}} C_1 C_2^T \sqrt{T}. \tag{7}$$

We provide a proof in Appendix B.4. From Theorem 1, we can observe the key property of OPC: for deterministic policies, the on-policy model error from Eq. (4) is zero and independent of the learned models' predictive distribution $\tilde{p}^{\mathrm{model}}$, so that $\eta = \tilde{\eta}_\star^{\mathrm{opc}}$. For policies with non-zero variance, the bound scales exponentially with $T$, highlighting the problem of compounding errors. In this case, as in the off-policy case, the model quality determines how well we can generalize to different actions. We show in Appendix B.5 that, for one-step predictions, OPC's prediction error scales as the minimum of policy variance and model error. To further alleviate the issue of compounding errors, one could extend Theorem 1 with a branched rollout scheme similarly to the results by Janner et al. (2019), such that the rollouts are only of length $H \ll T$.

In practice, $\tilde{p}_\star^{\mathrm{opc}}$ cannot be realized as it requires the true (unknown) state transition model $p$. However, as we use more on-policy reference trajectories for $\tilde{p}^{\mathrm{opc}}$ in Eq. (5), it also converges to zero on-policy error in probability for deterministic policies.

**Lemma 1.** *Let $M$ be a MDP with dynamics $p(\mathbf{s}_{t+1} \mid \mathbf{s}_t, \mathbf{a}_t)$ and reward $r < r_{\max}$. Let $\tilde{M}$ be another MDP with dynamics $\tilde{p}^{\mathrm{model}} \neq p$. Assume a deterministic policy $\pi : \mathcal{S} \mapsto \mathcal{A}$ and a set of trajectories $\mathcal{D} = \bigcup_{b=1}^B \{(\hat{\mathbf{s}}_t^b, \hat{\mathbf{a}}_t^b, \hat{\mathbf{s}}_{t+1}^b)\}_{t=0}^{T-1}$ collected from $M$ under $\pi$. If we use OPC Eq. (5) with data $\mathcal{D}$, then*

$$\lim_{B \to \infty} \mathrm{Pr}\left(|\eta - \tilde{\eta}^{\mathrm{opc}}| > \varepsilon\right) = 0 \quad \forall \varepsilon > 0 \quad \text{with convergence rate} \quad \mathcal{O}(1/B). \tag{8}$$

We provide a proof in Appendix B.4. Lemma 1 states that given sufficient reference on-policy data, the performance gap due to model errors becomes arbitrarily small for *any* model $\tilde{p}^{\mathrm{model}}$ when using OPC. While the assumption of infinite on-policy data in Lemma 1 is unrealistic for practical applications, we found empirically that OPC drastically reduces the on-policy model error even when the assumptions are violated. In our implementation, we use stochastic policies as well as trajectories from previous policies, i.e., off-policy data, for the corrections in OPC (see also Section 4.2).

## 3.4 DISCUSSION

**Epistemic uncertainty** So far, we have only considered aleatoric uncertainty (noise) in our transition models. In practice, modern methods additionally distinguish epistemic uncertainty that arises from having seen limited data (Deisenroth & Rasmussen, 2011; Chua et al., 2018). This leads to a distribution (or an ensemble) of models, where each sample could explain the data. In this setting, we apply OPC by correcting *each sample individually*. This allows us to retain epistemic uncertainty estimates after applying OPC, while the epistemic uncertainty is zero on-policy.

**Limitations** Since OPC uses on-policy data, it is inherently limited to local policy optimization where the policy changes slowly over time. As a consequence, it is not suitable for global exploration schemes like the one proposed by Curi et al. (2020). Similarly, since OPC uses the observed data and corrects it only with the expected learned model, $\tilde{p}^{\mathrm{opc}}$ always uses the on-policy transition noise (aleatoric uncertainty) from the data, even if the model has learned to represent it. While not having to learn a representation for aleatoric uncertainty can be a strength, it limits our approach to environments where the aleatoric uncertainty does not vary significantly with states/actions. It is possible to extend the method to the heteroscedastic noise setting under additional assumptions that enable distinguishing model error from transition noise (Schöllig & D'Andrea, 2009). Lastly, our method applies directly only to MDPs, since we rely on state-observations. Extending the ideas to partially observed environments is an interesting direction for future research.

## 4 EXPERIMENTAL RESULTS

We begin the experimental section with a motivating example on a toy problem to highlight the impact of OPC on the policy gradient estimates in the presence of model errors. The remainder of the section focuses on comparative evaluations and ablation studies on complex continuous control tasks.

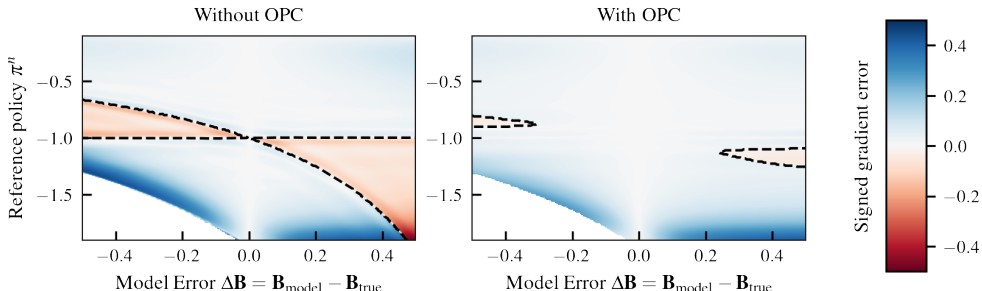

Figure 2: Signed gradient error when using inaccurate models Eq. (9) to estimate the policy gradient without (left) and with (right) OPC. The background's opacity depicts the error's magnitude, whereas color denotes if the sign of estimated and true gradient differ (red) or coincide (blue). OPC improves the gradient estimate in the presence of model errors. Note that the optimal policy is $\theta^* = -1.0$.

## 4.1 ILLUSTRATIVE EXAMPLE

In Section 3.3, we investigate the influence of the model error directly on the expected return from a theoretical perspective. From a practical standpoint, another relevant question is how the policy optimization and the respective policy gradients are influenced by model errors. For general environments and reward signals, this question is difficult to answer, due to the typically high-dimensional state/action spaces and large number of parameters governing the dynamics model as well as the policy. To shed light on this open question, we resort to a simple low-dimensional example and investigate how OPC improves the gradient estimates under a misspecified dynamics model.

In particular, we assume a one-dimensional deterministic environment with linear transitions and a linear policy. The benefits of this example are that we can 1) compute the true policy gradient based on a single rollout (determinism), 2) determine the environment's closed-loop stability under the policy (linearity), and 3) visualize the gradient error as a function of all relevant parameters (low dimensionality). The dynamics and initial state distribution are specified by $p(\mathbf{s}_{t+1} \mid \mathbf{s}_t, \mathbf{a}_t) = \delta(\mathbf{A}\mathbf{s}_t + \mathbf{B}\mathbf{a}_t \mid \mathbf{s}_t, \mathbf{a}_t)$ with $\rho(\mathbf{s}_0) = \delta(\mathbf{s}_0)$ where $\mathbf{A}, \mathbf{B} \in \mathbb{R}$ and $\delta(\cdot)$ denotes the Dirac-delta distribution. We define a deterministic linear policy $\pi_\theta(\mathbf{a}_t \mid \mathbf{s}_t) = \delta(\theta\mathbf{s}_t \mid \mathbf{s}_t)$ that is parameterized by the scalar $\theta \in \mathbb{R}$. The objective is to drive the state to zero, which we encode with an exponential reward $r(\mathbf{s}_t, \mathbf{a}_t) = \exp\left\{-(\mathbf{s}_t/\sigma_r)^2\right\}$. Further, we assume an approximate dynamics model

$$\tilde{p}(\mathbf{s}_{t+1} \mid \mathbf{s}_t, \mathbf{a}_t) = \delta((\mathbf{A} + \Delta\mathbf{A})\mathbf{s}_t + (\mathbf{B} + \Delta\mathbf{B})\mathbf{a}_t \mid \mathbf{s}_t, \mathbf{a}_t), \tag{9}$$

where $\Delta\mathbf{A}, \Delta\mathbf{B}$ quantify the mismatch between the approximate model and the true environment. In practice, the mismatch can arise due to noise-corrupted observations of the true state or, in the case of stochastic environments, due to a finite amount of training data.

With the setting outlined above, we investigate how model errors influence the estimation of policy gradients. To this end, we roll out different policies under models with varying degrees of error $\Delta\mathbf{B}$. For each policy/model combination, we compute the model-based policy gradient and compare it to the true gradient. The results are summarized in Fig. 2, where the background's opacity depicts the gradient error's magnitude and its color indicates whether the respective signs of the gradients are the same (blue, $\geq 0$) or differ (red, $< 0$). For policy optimization, the sign of the gradient estimate is paramount. However, we see in the left-hand image that even small model errors can lead to the wrong sign of the gradient. OPC significantly reduces the magnitude of the gradient error and increases the robustness towards model errors. See also Appendix C for a more in-depth analysis.

## 4.2 EVALUATION ON CONTINUOUS CONTROL TASKS

In the following section, we investigate the impact of OPC on a range of continuous control tasks. To this end, we build upon the current state-of-the-art model-based RL algorithm MBPO (Janner et al., 2019). Further, we aim to answer the important question about how data diversity affects MBRL, and OPC in particular. While having a model that can generate (theoretically) an infinite amount of data is intriguing, the benefit of having more data is limited by the model quality in terms

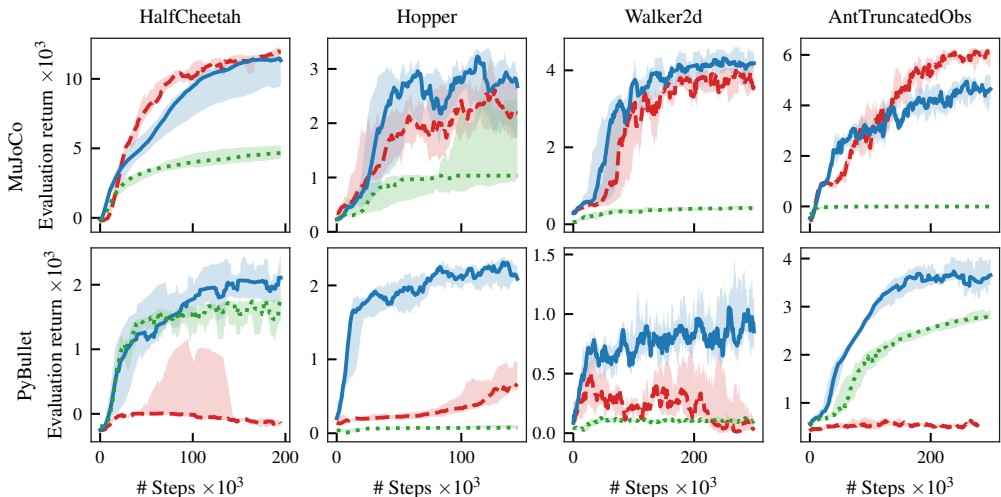

Figure 3: Comparison of OPC (——), MBPO($\star$) (– – –) and SAC (·····) on four environments from the MuJoCo control suite (top row) and their respective PyBullet implementations (bottom row).

of being representative of the true environment. For OPC, the following questions arise from this consideration: Do longer rollouts help to generate better data? And is there a limit to the value of simulated transition data, i.e., is more always better?

For the dynamics model $\tilde{p}^{\mathrm{model}}$, we follow Chua et al. (2018); Janner et al. (2019) and use a probabilistic ensemble of neural networks, where each head predicts a Gaussian distribution over the next state and reward. For policy optimization, we employ the soft actor critic (SAC) algorithm by Haarnoja et al. (2018). All learning curves are presented in terms of the median (lines) and interquartile range (shaded region) across ten independent experiments, where we smooth the evaluation return with a moving average filter to accentuate the results of particularly noisy environments. Apart from small variations in the hyperparameters, the only difference between OPC and MBPO is that our method uses $\tilde{p}^{\mathrm{opc}}$, while MBPO uses $\tilde{p}^{\mathrm{model}}$. We provide pseudo-code for the model rollouts of MBPO and OPC in Algorithm 2 in Appendix A.1. Generally, we found that OPC was more robust to the choice of hyperparameters. The rollout horizon to generate training data is set to $H = 10$ for all experiments. Note that when using $\tilde{p}^{\mathrm{data}}$ to generate data, we retain the standard (model-free) SAC algorithm.

Our implementation is based upon the code from Janner et al. (2019). We made the following changes to the original implementation: 1) The policy is only updated at the end of an epoch, not during rollouts on the true environment. 2) The replay buffer retains data for a fixed number of episodes, to clearly distinguish on- and off-policy data. 3) For policy optimization, MBPO uses a small number of environment transitions in addition to those from the model. We found that this design choice did not consistently improve performance and added another level of complexity. Therefore, we refrain from mixing environment and model transitions and only use simulated data for policy optimization. While we stay true to the key ideas of MBPO under these changes, we denote our variant as MBPO($\star$) to avoid ambiguity. See Appendices D.6 and D.7 for a comparison to the original MBPO algorithm.

**Comparative Evaluation**     We begin our analysis with a comparison of our method to MBPO($\star$) and SAC on four continuous control benchmark tasks from the MuJoCo control suite (Todorov et al., 2012) and their respective PyBullet variants (Ellenberger, 2018–2019). The results are presented in Fig. 3. We see that the difference in performance between both methods is only marginal when evaluated on the MuJoCo environments (Fig. 3, top row). Notably, the situation changes drastically for the PyBullet environments (Fig. 3, bottom row). Here, MBPO($\star$) exhibits little to no learning progress, whereas OPC succeeds at learning a good policy with few interactions in the environment. One of the main differences between the environments (apart from the physics engine itself) is that the PyBullet variants have initial state distributions with significantly larger variance.

**Influence of State Representation**     In general, the success of RL algorithms should be agnostic to the way an environment represents its state. In robotics, joint angles $\vartheta$ are often re-parameterized by a sine/cosine transformation, $\vartheta \mapsto [\sin(\vartheta), \cos(\vartheta)]$. We show that even for the simple CartPole envi-

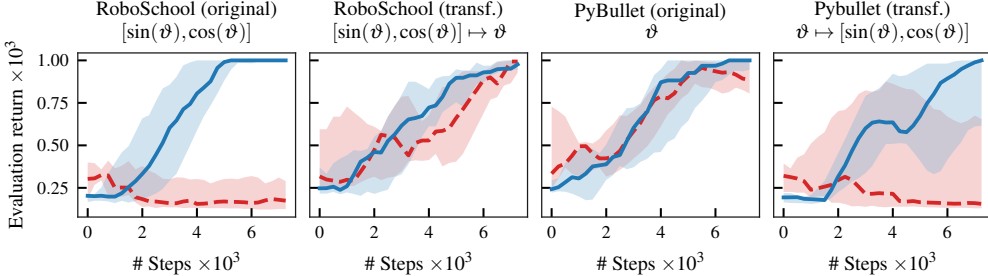

Figure 4: Comparison of OPC (———) and MBPO($\star$) (- - -) on different variants of the CartPole environment. When the pole's angle $\vartheta$ is observed directly (center plots), both algorithms successfully learn a policy. With the sine/cosine transformations (outer plots), MBPO($\star$) fails to solve the task.

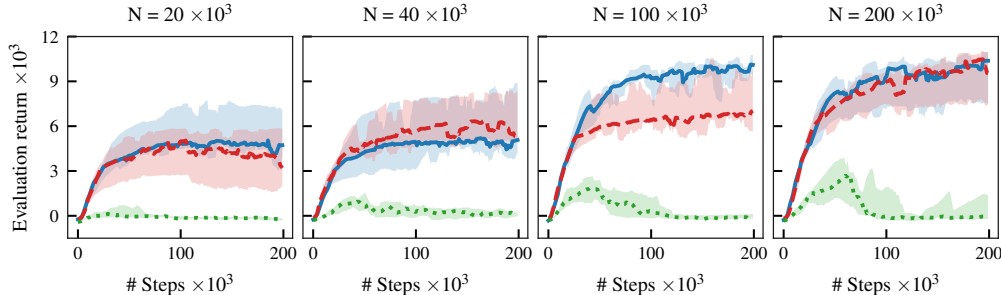

Figure 5: Ablation study for OPC on the HalfCheetah environment. In each plot, we fix the number of simulated transitions $N$ and vary the rollout lengths $H = \{1(\cdots\cdots), 5(\text{- - -}), 10(\text{———})\}$.

ronment, the parameterization of the pole's angle has a large influence on the performance of MBRL. In particular, we compare OPC and MBPO($\star$) on the RoboSchool and PyBullet variants of the CartPole environment, which represent the pole's angle with and without the sine/cosine transformation. The results are shown in Fig. 4. To rule out other effects than the angle's representation, we repeat the experiment for each implementation but transform the state to the other representation, respectively. Notably, OPC successfully learns a policy irrespective of the state's representation, whereas MBPO($\star$) fails if the angle of the pole is represented by the sine/cosine transformation.

**Influence of Data Diversity** Here, we investigate whether multi-step predictions are in fact beneficial compared to single-step predictions during the data generation process. To this end, we keep the total number of simulated transitions $N$ for training constant, but choose different horizon lengths $H = \{1, 5, 10\}$. The corresponding numbers of simulated rollouts are then given by $n_{\text{rollout}} = N/H$. The results for $N = \{20, 40, 100, 200\} \times 10^3$ on the HalfCheetah environment are shown in Fig. 5. First, note that more data leads to a higher asymptotic return, but after a certain point more data only leads to diminishing returns. Further, the results indicate that one-step predictions are not enough to generate sufficiently diverse data. Note that this result contradicts the findings by Janner et al. (2019) that one-step predictions are often sufficient for MBPO.

## 5 CONCLUSION

In this paper, we have introduced *on-policy corrections* (OPC), a novel method that combines observed transition data with model-based predictions to mitigate model-bias in MBRL. In particular, we extend a replay buffer with a learned model to account for state-action pairs that have not been observed on the real environment. This approach enables the generation of more realistic transition data to more closely match the true state distribution, which was further motivated theoretically by a tightened improvement bound on the expected return. Empirically, we demonstrated superior performance on high-dimensional continuous control tasks as well as robustness towards state representations.

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

# SUPPLEMENTARY MATERIAL

In the appendix we provide additional details on our method, ablation studies, and the detailed hyperparameter configurations used in the paper. An overview is shown below.

## Table of Contents

# A   IMPLEMENTATION DETAILS AND COMPUTATIONAL RESOURCES

## A.1   DETAILED ALGORITHM FOR ROLLOUTS WITH OPC

In Section 3.2 and Fig. 1a, we have introduced the OPC transition model and how to roll out trajectories with the model. Here, we will give more details on the algorithmic implementation for the generation of simulated data. Algorithm 2 follows the branched rollout scheme from MBPO (Janner et al., 2019). Differences to MBPO are highlighted in blue.

Generally, OPC only requires a deterministic transition function $\tilde{f}$ to compute the corrective term in Line 8 in Algorithm 2. For models that include aleatoric uncertainty, we choose $\tilde{f}(\mathbf{s}_t, \mathbf{a}_t) = \mathbb{E}_{\mathbf{s}_{t'}}[\tilde{p}(\mathbf{s}_{t'} \mid \mathbf{s}_t, \mathbf{a}_t)]$. If, in addition, the model includes epistemic uncertainty, we refer to the comment in Section 3.4 in the main part of the paper.

In practice, rollouts on the true environment are terminated early if, for instance, a particular state exceeds a user-defined boundary. Consequently, not all trajectories in the replay buffer are necessarily of length $T$. Since the prediction in Line 8 requires valid transition tuples for the correction term, we additionally check in Line 10 whether the next reference state is a terminal state. Thus, in contrast to MBPO, for OPC we terminate the inner loop in Algorithm 2 early if either a simulated or reference state is a terminal state.

---

**Algorithm 2** Branched rollout scheme with OPC model (differences to MBPO highlighted in blue)

**Input:** Required parameters:

- Set of trajectories $\mathcal{D}_n^b = \{(\hat{\mathbf{s}}_t^{n,b}, \hat{\mathbf{a}}_t^{n,b}, \hat{\mathbf{s}}_{t+1}^{n,b})\}_{t=0}^{T-1}$ for $b \in \{1, \dots, B\}$
- Environment model $\tilde{p}(\mathbf{s}_{t'} \mid \mathbf{s}_t, \mathbf{a}_t)$. Define $\tilde{f}(\mathbf{s}, \mathbf{a}) = \mathbb{E}_{\mathbf{s}_{t'}}[\tilde{p}(\mathbf{s}_{t'} \mid \mathbf{s}, \mathbf{a})]$.
- Policy $\pi_\theta$
- Prediction horizon $H$
- Number of simulated transitions $N$

1: $\mathcal{D}^{\mathrm{sim}} \leftarrow \emptyset$: Initialize empty buffer for simulated transitions
2: **while** $|\mathcal{D}^{\mathrm{sim}}| < N$ **do**
3:     $b \sim \mathcal{U}\{1, B\}$: Sample random reference trajectory
4:     $t \sim \mathcal{U}\{0, T-H\}$: Sample random starting state
5:     $h \leftarrow 0, \mathbf{s}_t \leftarrow \hat{\mathbf{s}}_t^b$: Initialize starting state
6:     **while** $h < H$ **do**
7:         $\mathbf{a}_{t+h} \sim \pi_\theta(\mathbf{a} \mid \mathbf{s}_{t+h})$: Sample action from policy
8:         $\mathbf{s}_{t+h+1} \leftarrow \hat{\mathbf{s}}_{t+h+1} + \tilde{f}(\mathbf{s}_{t+h}, \mathbf{a}_{t+h}) - \tilde{f}(\hat{\mathbf{s}}_{t+h}^b, \hat{\mathbf{a}}_{t+h}^b)$: Do one-step prediction with OPC
9:         $\mathcal{D}^{\mathrm{sim}} \leftarrow \mathcal{D}^{\mathrm{sim}} \cup (\mathbf{s}_{t+h}, \mathbf{a}_{t+h}, \mathbf{s}_{t+h+1})$: Store new transition tuple
10:        **if** ($\mathbf{s}_{t+h+1}$ is terminal) or ($\hat{\mathbf{s}}_{t+h+1}^b$ is terminal) **then**
11:            break
12:        $h \leftarrow h + 1$: Increase counter
    **return** $\mathcal{D}^{\mathrm{sim}}$

---

## A.2 HYPERPARAMETER SETTINGS

Below, we list the most important hyperparameter settings that were used to generate the results in the main paper.

Table 1: Hyperparameter settings for OPC (blue) and MBPO($\star$) (red) for results shown in Fig. 3. Note that the respective hyperparameters for each environment are shared across the different implementations, i.e., MuJoCo and PyBullet.

|  | HalfCheetah | Hopper | Walker2D | AntTruncatedObs |
|---|---|---|---|---|
| epochs | 200 | 150 | 300 | 300 |
| env steps per epoch | 1000 | | | |
| retain epochs | 50 / 5 | 50 | 50 | 5 |
| policy updates per epoch | 40 | 20 | 20 | 20 |
| model horizon | 10 | | | |
| model rollouts per epoch | 100'000 | | | |
| mix-in ratio | 0.0 | | | |
| model network | ensemble of 7 with 5 elites | | | |
| policy network | MLP with 2 hidden layers of size 64 | | | |

## A.3 IMPLEMENTATION AND COMPUTATIONAL RESOURCES

Our implementation is based on the code from MBPO (Janner et al., 2019), which is open-sourced under the MIT license. All experiments were run on an HPC cluster, where each individual experiment used one Nvidia V100 GPU and four Intel Xeon CPUs. All experiments (including early debugging and evaluations) amounted to a total of 84'713 hours, which corresponds to roughly 9.7 years if the jobs ran sequentially. Most of this compute was required to ensure reproducibility (ten random seeds per job and ablation studies over the effects of parameters). The Bosch Group is carbon-neutral. Administration, manufacturing and research activities do no longer leave a carbon footprint. This also includes GPU clusters on which the experiments have been performed.

## B  THEORETICAL ANALYSIS OF ON-POLICY CORRECTIONS

In this section, we analyze OPC from a theoretical perspective and how it affects policy improvement.

**Notation**   In the following, we drop the $n$ superscript for states and actions for ease of exposition. That is, $\hat{\mathbf{s}}^{n,b} = \hat{\mathbf{s}}^b$ and $\hat{\mathbf{a}}^{n,b} = \hat{\mathbf{a}}^b$.

### B.1  GENERAL POLICY IMPROVEMENT BOUND

We begin by deriving inequality Eq. (4), which serves as motivation for OPC and is the foundation for the theoretical analysis. Our goal is to bound the difference in expected return for the policies before and after the policy optimization step, i.e., $\eta_{n+1} - \eta_n$. Since we are considering the MBRL setting, it is natural to express the improvement bound in terms of the expected return under the model $\tilde{\eta}$ and thus obtain the following

$$\eta_{n+1} - \eta_n = \eta_{n+1} - \eta_n + \tilde{\eta}_{n+1} - \tilde{\eta}_{n+1} + \tilde{\eta}_n - \tilde{\eta}_n$$
$$= \tilde{\eta}_{n+1} - \tilde{\eta}_n + \eta_{n+1} - \tilde{\eta}_{n+1} + \tilde{\eta}_n - \eta_n$$
$$\underbrace{\eta_{n+1} - \eta_n}_{\substack{\text{True policy} \\ \text{improvement}}} \geq \underbrace{\tilde{\eta}_{n+1} - \tilde{\eta}_n}_{\substack{\text{Model policy} \\ \text{improvement}}} - \underbrace{|\eta_{n+1} - \tilde{\eta}_{n+1}|}_{\substack{\text{Off-policy} \\ \text{model error}}} - \underbrace{|\tilde{\eta}_n - \eta_n|}_{\substack{\text{On-policy} \\ \text{model error}}}.$$

According to this bound, the improvement of the policy under the true environment is governed by the three terms on the LHS:

- *Model policy improvement*: This term is what we are directly optimizing in MBRL offset by the return of the previous iteration $\tilde{\eta}_n$, which is constant given the current policy $\pi_n$. Assuming that we are not at an optimum, standard policy optimization algorithms guarantee that this term is non-negative.

- *Off-policy model error*: The last term is the difference in return for the true environment and model under the improved policy $\pi_{n+1}$. This depends largely on the generalization properties of our model, since it is not trained on data under $\pi_{n+1}$.

- *On-policy model error*: This term compares the on-policy return under $\pi_n$ between the true environment and the model and it is zero for any model $\tilde{p} = p$. Since we have access to transitions from the true environment under the $\pi_n$, the replay buffer Eq. (3) fulfills this condition under certain circumstances and the on-policy model error vanishes, see Lemma 2.

Note that the learned model Eq. (2) is able to generalize to unseen state-action pairs better than the replay buffer Eq. (3) and accordingly will achieve a lower off-policy model error. The motivation behind OPC is therefore to combine the best of the learned model and the replay buffer to reduce both on- and off-policy model errors.

### B.2  PROPERTIES OF THE REPLAY BUFFER

The benefit of the replay buffer Eq. (3) is that it can never introduce any model-bias such that any trajectory sampled from this model is guaranteed to come from the true state distribution. Accordingly, if we have collected sufficient data under the same policy, the on-policy model error vanishes.

**Lemma 2.** *Let $M$ be the true MDP with (stochastic) dynamics $p$, bounded reward $r < r_{\max}$ and let $\tilde{p}^{\text{data}}$ be the transition model for the replay buffer Eq. (3). Further, consider a set of trajectories $\mathcal{D} = \bigcup_{b=1}^{B} \{(\hat{\mathbf{s}}_t^b, \hat{\mathbf{a}}_t^b, \hat{\mathbf{s}}_{t+1}^b)\}_{t=0}^{T-1}$ collected from $M$ under policy $\pi$. If we collect more and more on-policy training data under the same policy, then*

$$\lim_{B \to \infty} \Pr\left(|\eta - \tilde{\eta}^{\text{replay}}| > \varepsilon\right) = 0 \quad \forall \varepsilon > 0$$

*where $\tilde{\eta}^{\text{replay}}$ is the expected return under the replay buffer using the collected trajectories $\mathcal{D}$.*

*Proof.* First, note that the corresponding expected return for the replay-buffer model is given by

$$\tilde{\eta}^{\text{replay}} = \frac{1}{B} \sum_{b=1}^{B} \sum_{t=0}^{T-1} r(\hat{\mathbf{s}}_t^b, \hat{\mathbf{a}}_t^b),$$

which is a sample-based approximation of the true reward $\eta$. By the weak law of large numbers (see e.g., Blitzstein & Hwang (2019, Theorem 10.2.2)), the Lemma then holds. □

### B.3 PROPERTIES OF THE LEARNED MODEL

Following Janner et al. (2019, Lemma B.3), a general bound on the performance gap between two MDPs with different dynamics can be given by

$$|\eta_1 - \eta_2| \leq 2r_{\max}\epsilon_m \sum_{t=1}^{H} t\gamma^t. \tag{10}$$

where $\epsilon_m \geq \max_t \mathbb{E}_{\mathbf{s} \sim p_1^t(\mathbf{s})} \mathrm{KL}(p_1(\mathbf{s}', \mathbf{a}) || p_2(\mathbf{s}', \mathbf{a}))$ bounds the mismatch between the respective transition models. Now, the final form of Eq. (10) depends on the horizon length $H$. For $H \to \infty$, we obtain the original result form Janner et al. (2019) with $\sum_{t \geq 1} t\gamma^t = \gamma/(1-\gamma)^2$. For the finite horizon case one can obtain tigther bounds when $H$ is smaller than the effective horizon, $H < \gamma/(1-\gamma)$, encoded in the discount factor:

$$\sum_{t=1}^{H} t\gamma^t \leq \min\left\{\frac{H(H+1)}{2}, \frac{H}{1-\gamma}, \frac{\gamma}{(1-\gamma)^2}\right\}, \tag{11}$$

which can be verified by upper bounding $t \leq H$ to obtain $H/(1-\gamma)$ or by bounding $\gamma \leq 1$ to obtain $\mathcal{O}(H^2)$.

Note that this bound is vacuous for deterministic policies, since the KL divergence between two distributions with non-overlapping support is infinite. In the following we focus on the Wasserstein metric under the Euclidean distance.

### B.4 PROPERTIES OF OPC

In this section, we analyze the properties of OPC relative to the true, unknown environment's transition distribution $p$ and a learned representation $\tilde{p}$. In general, the OPC-model mixes observed transitions from the environment with the learned model. The resulting transitions are then a combination of the mean transitions from the learned model, the aleatoric noise from the data (the environment), and the mean-error between our learned model and the environment.

#### B.4.1 PROOF OF LEMMA 1

In this section, we prove Lemma 1 by showing that the OPC-model coincides with the replay-buffer Eq. (3) in the case of a deterministic policy and thus lead to the same expected return $\tilde{\eta}$.

**Lemma 3.** *Let $M$ be the true MDP with (stochastic) dynamics $p$ and let $\tilde{M}$ be a MDP with the same reward function $r$ and initial state distribution $\rho_0$, but different dynamics $\tilde{p}^{\mathrm{model}}$, respectively. Further, assume a deterministic policy $\pi : \mathcal{S} \mapsto \mathcal{A}$ and a set of trajectories $\mathcal{D} = \bigcup_{b=1}^{B}\{(\hat{\mathbf{s}}_t^b, \hat{\mathbf{a}}_t^b, \hat{\mathbf{s}}_{t+1}^b)\}_{t=0}^{T-1}$ collected from $M$ under $\pi$. If we extend the approximate dynamics $\tilde{p}^{\mathrm{model}}$ by OPC with data $\mathcal{D}$, then*

$$\tilde{\eta}^{\mathrm{replay}} = \eta^{\mathrm{opc}},$$

*where $\tilde{\eta}^{\mathrm{replay}}$ and $\tilde{\eta}^{\mathrm{opc}}$ are the model-based returns following models Eqs. (3) and (5), respectively.*

*Proof.* For the proof, it suffices to show that the resulting state distributions of the two transition models $\tilde{p}^{\mathrm{data}}$ and $\tilde{p}^{\mathrm{opc}}$ under the deterministic policy $\pi$ are the same for all $b$ with $1 \leq b \leq B$. We show this by induction:

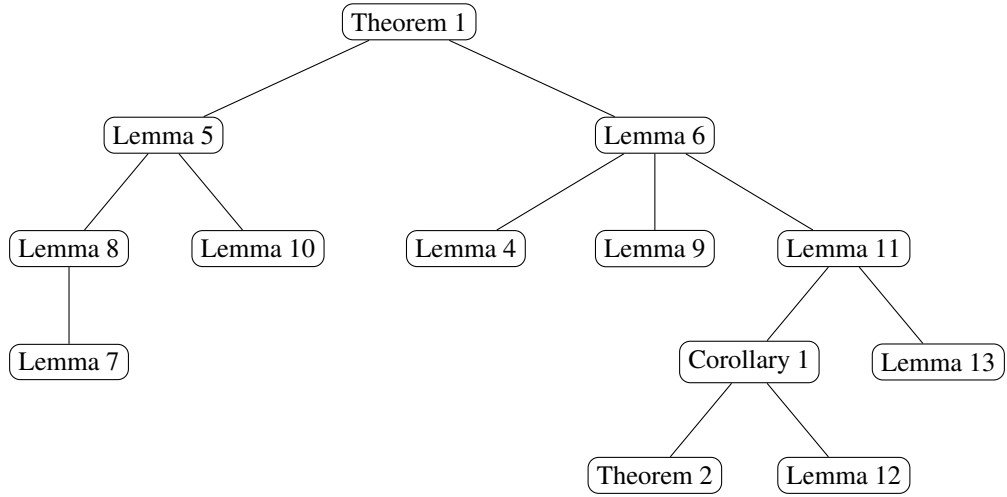

Figure 6: Overview about the supporting Lemmas for the proof of Theorem 1.

For $t = 0$ the states are sampled from the initial state distribution, thus we have $\mathbf{s}_0 = \hat{\mathbf{s}}_0^b$ by definition. Assume that $\mathbf{s}_t = \hat{\mathbf{s}}_t^b$ as an induction hypothesis. Then

$$\tilde{p}^{\mathrm{opc}}(\mathbf{s}_{t+1} \mid \mathbf{s}_t, \pi(\mathbf{s}_t), b) = \delta\Big(\mathbf{s}_{t+1} - \hat{\mathbf{s}}_{t+1}^b\Big) * \delta\Big(\mathbf{s}_{t+1} - \Big[f(\mathbf{s}_t, \pi(\mathbf{s}_t)) - f(\hat{\mathbf{s}}_t^b, \hat{\mathbf{a}}_t^b)\Big]\Big)$$

$$= \delta\Big(\mathbf{s}_{t+1} - \Big[\hat{\mathbf{s}}_{t+1}^b + f(\mathbf{s}_t, \pi(\mathbf{s}_t)) - f(\hat{\mathbf{s}}_t^b, \hat{\mathbf{a}}_t^b)\Big]\Big)$$

$$= \delta\Big(\mathbf{s}_{t+1} - \Big[\hat{\mathbf{s}}_{t+1}^b + f(\hat{\mathbf{s}}_t^b, \hat{\mathbf{a}}_t^b) - f(\hat{\mathbf{s}}_t^b, \hat{\mathbf{a}}_t^b)\Big]\Big)$$

$$= \delta\Big(\mathbf{s}_{t+1} - \hat{\mathbf{s}}_{t+1}^b\Big)$$

$$= \tilde{p}^{\mathrm{data}}(\mathbf{s}_{t+1} \mid t + 1, b)$$

where the second step follows by the induction hypothesis and due to the deterministic policy. Thus, for any index $b$ we have $\tau_b^{\mathrm{opc}} = \tau_b$ and the result follows. □

Now, combining Lemmas 2 and 3 proofs the result in Lemma 1.

### B.4.2 PROOF OF THEOREM 1

In this section we prove our main result. An overview of the lemma dependencies is shown in Fig. 6.

**Remark on Notation** In the main paper, we unify the notation for state sequence probabilities of the different models Eqs. (2), (3) and (5) as $\tilde{p}^{\mathrm{x}}(\tau_{t:t+H} \mid t, b)$ with $\mathrm{x} \in \{\mathrm{replay}, \mathrm{model}, \mathrm{opc}\}$. This allows for a consistent description of the respective rollouts independent of the model being a learned representation, a replay buffer or the OPC-model. For that notation, the index $b$ denotes the sampled trajectory from the collected data on the real environment. Implicitly, we therefore condition the state sequence probability on the observed transition tuples $[(\hat{\mathbf{s}}_{t+1}^b, \hat{\mathbf{s}}_t^b, \hat{\mathbf{a}}_t^b), \ldots, (\hat{\mathbf{s}}_{t+H}^b, \hat{\mathbf{s}}_{t+H-1}^b, \hat{\mathbf{a}}_{t+H-1}^b)]$, i.e.,

$$\tilde{p}^{\mathrm{x}}(\tau_{t:t+H} \mid t, b) = \tilde{p}^{\mathrm{x}}(\tau_{t:t+H} \mid (\hat{\mathbf{s}}_{t+1}^b, \hat{\mathbf{s}}_t^b, \hat{\mathbf{a}}_t^b), \ldots, (\hat{\mathbf{s}}_{t+H}^b, \hat{\mathbf{s}}_{t+H-1}^b, \hat{\mathbf{a}}_{t+H-1}^b)), \tag{12}$$

where we omit the explicit conditioning for the sake of brevity in the main paper. Similarly, we can write the one-step model Eq. (5) for OPC in an explicit form as

$$\tilde{p}^{\mathrm{opc}}(\mathbf{s}_{t+1} \mid \mathbf{s}_t, \mathbf{a}_t, b) = \tilde{p}^{\mathrm{opc}}(\mathbf{s}_{t+1} \mid \mathbf{s}_t, \mathbf{a}_t, \hat{\mathbf{s}}_{t+1}^b, \hat{\mathbf{s}}_t^b, \hat{\mathbf{a}}_t^b). \tag{13}$$

Note that with the explicit notation, the relation between the OPC-model Eq. (5) and *generalized* OPC-model Eq. (6) becomes clear:

$$\tilde{p}_\star^{\mathrm{opc}}(\mathbf{s}_{t+1} \mid \mathbf{s}_t, \mathbf{a}_t, b) = \tilde{p}^{\mathrm{opc}}(\mathbf{s}_{t+1} \mid \mathbf{s}_t, \mathbf{a}_t, \hat{\mathbf{s}}_t^b, \hat{\mathbf{a}}_t^b) = \int \tilde{p}^{\mathrm{opc}}(\mathbf{s}_{t+1} \mid \mathbf{s}_t, \mathbf{a}_t, \hat{\mathbf{s}}_{t+1}^b \hat{\mathbf{s}}_t^b, \hat{\mathbf{a}}_t^b) \, \mathrm{d}\hat{\mathbf{s}}_{t+1}^b. \tag{14}$$

For the following proofs, we stay with the explicit notation for sake of clarity and instead omit the conditioning on $b$.

**Generalized OPC-Model** In this section, we have a closer look at the *generalized* OPC-model Eq. (6). The main difference between Eqs. (5) and (6) is that the former is in fact transitioning deterministically (the stochasticity arises from the environment's aleatoric uncertainty which manifests itself in the observed transitions). The two models can be related via marginalization of $\hat{s}_{t+1}^b$, see Eq. (14). The resulting generalized OPC-model can then be related to the true transition distribution according to the following Lemma.

**Lemma 4.** *For all $\hat{s}_t^b, \hat{a}_t^b \in \mathcal{S} \times \mathcal{A}$ with $\hat{s}_{t+1}^b \sim p(\hat{s}_{t+1} \mid \hat{s}_t^b, \hat{a}_t^b)$ it holds that*

$$\tilde{p}^{\mathrm{opc}}(\mathbf{s}_{t+1} \mid \mathbf{s}_t, \mathbf{a}_t, \hat{s}_t^b, \hat{a}_t^b) = p\Big(\mathbf{s}_{t+1} - \underbrace{\big[\tilde{f}(\mathbf{s}_t, \mathbf{a}_t) - \tilde{f}(\hat{s}_t^b, \hat{a}_t^b)\big]}_{\text{Mean correction}} \mid \mathbf{s}_t, \mathbf{a}_t, \hat{s}_t^b, \hat{a}_t^b\Big), \qquad (15)$$

*where $\tilde{f}(\mathbf{s}_t, \mathbf{a}_t) = \mathbb{E}[\tilde{p}^{\mathrm{model}}(\mathbf{s}_{t+1} \mid \mathbf{s}_t, \mathbf{a}_t)]$ is the mean transition of the learned model Eq. (2) and $\tilde{p}^{\mathrm{opc}}(\mathbf{s}_{t+1} \mid \mathbf{s}_t, \mathbf{a}_t, \hat{s}_t^b, \hat{a}_t^b)$ denotes the OPC-model if we marginalize over the distribution for $\hat{s}_{t+1}^b$ instead of using its observed value.*

*Proof.* Using the explicit notation Eq. (13), the OPC-model from Eq. (5) is defined as

$$\tilde{p}^{\mathrm{opc}}(\mathbf{s}_{t+1} \mid \mathbf{s}_t, \mathbf{a}_t, b) = \tilde{p}^{\mathrm{opc}}(\mathbf{s}_{t+1} \mid \mathbf{s}_t, \mathbf{a}_t, \hat{s}_t^b, \hat{a}_t^b, \hat{s}_{t+1}^b) \qquad (16)$$

$$= \delta\Big(\mathbf{s}_{t+1} - \hat{s}_{t+1}^b\Big) * \delta\Big(\mathbf{s}_{t+1} - \big[\tilde{f}(\mathbf{s}_t, \mathbf{a}_t) - \tilde{f}(\hat{s}_t^b, \hat{a}_t^b)\big]\Big), \qquad (17)$$

$$= \delta\Big(\mathbf{s}_{t+1} - \big[\hat{s}_{t+1}^b + \tilde{f}(\mathbf{s}_t, \mathbf{a}_t) - \tilde{f}(\hat{s}_t^b, \hat{a}_t^b)\big]\Big). \qquad (18)$$

With $\hat{s}_{t+1}^b \sim p(\hat{s}_{t+1} \mid \hat{s}_t^b, \hat{a}_t^b)$, marginalizing $\hat{s}_{t+1}$ yields (note that we use $\hat{s}_{t+1}$ instead of $\mathbf{s}_{t+1}$ to denote the random variable for the next state in order to distinguish it from the random variable for $\tilde{p}^{\mathrm{opc}}(\mathbf{s}_{t+1} \mid \mathbf{s}_t, \mathbf{a}_t, b)$ under the integral)

$$\tilde{p}^{\mathrm{opc}}(\mathbf{s}_{t+1} \mid \mathbf{s}_t, \mathbf{a}_t, \hat{s}_t^b, \hat{a}_t^b) = \int \delta\Big(\mathbf{s}_{t+1} - \big[\hat{s}_{t+1} + \tilde{f}(\mathbf{s}_t, \mathbf{a}_t) - \tilde{f}(\hat{s}_t^b, \hat{a}_t^b)\big]\Big) p(\hat{s}_{t+1} \mid \hat{s}_t^b, \hat{a}_t^b) \, \mathrm{d}\hat{s}_{t+1},$$

$$= p\Big(\mathbf{s}_{t+1} - \big[\tilde{f}(\mathbf{s}_t, \mathbf{a}_t) - \tilde{f}(\hat{s}_t^b, \hat{a}_t^b)\big] \mid \hat{s}_t^b, \hat{a}_t^b\Big).$$

$\square$

*Remark* 1. An alternative way of writing the general OPC-model is the following,

$$\tilde{p}^{\mathrm{opc}}(\mathbf{s}_{t+1} \mid \mathbf{s}_t, \mathbf{a}_t, \hat{s}_t^b, \hat{a}_t^b) = \underbrace{p(\hat{s}_{t+1} \mid \hat{s}_t^b, \hat{a}_t^b)}_{\text{On-policy transition}} * \underbrace{\delta\Big(\mathbf{s}_{t+1} - \big[\tilde{f}(\mathbf{s}_t, \mathbf{a}_t) - \tilde{f}(\hat{s}_t^b, \hat{a}_t^b)\big]\Big)}_{\text{Mean correction term}}, \qquad (19)$$

which highlights that the $\tilde{p}^{\mathrm{opc}}$ transitions according to the true on-policy dynamics conditioned on data from the replay buffer, combined with a correction term. We can further explicitly see why an implementation of this model wouldn't be possible due to its dependency on the true transition probabilities. Thus, in practice, we're limited to the sample-based approximation shown in the paper.

The fundamental idea for the proof of Theorem 1 lies in the following Lemma 5, which is the foundation for bounding the on-policy error. The Wasserstein distance naturally arises in bounding this type of error model as it depends on the expected return under two different distributions. The final result is then summarized in Theorem 1.

**Lemma 5.** *Let $\tilde{p}^{\mathrm{opc}}$ be the generalized OPC-model (cf. Lemma 4) and $\tilde{\eta}^{\mathrm{opc}}$ be its corresponding expected return. Assume that the return $r(\mathbf{s}_t, \mathbf{a}_t)$ is $L_r$-Lipschitz and the policy $\pi(\mathbf{a}_t \mid \mathbf{s}_t)$ is $L_\pi$-Lipschitz with respect to $\mathbf{s}_t$ under the Wasserstein distance, then*

$$|\eta - \tilde{\eta}^{\mathrm{opc}}| \leq L_r \sqrt{1 + L_\pi^2} \sum_{t \geq 0} \gamma^t \mathcal{W}_2(p(\mathbf{s}_t), \tilde{p}^{\mathrm{opc}}(\mathbf{s}_t)) \qquad (20)$$

*Proof.*

$$|\eta - \tilde{\eta}^{\mathrm{opc}}| = \Big| \underset{\tau \sim p}{\mathbb{E}} \Big[ \sum_{t \geq 0} \gamma^t r(\hat{\mathbf{s}}_t, \hat{\mathbf{a}}_t) \Big] - \underset{\tau \sim \tilde{p}^{\mathrm{opc}}}{\mathbb{E}} \Big[ \sum_{t \geq 0} \gamma^t r(\mathbf{s}_t, \mathbf{a}_t) \Big] \Big| \tag{21}$$

$$= \Big| \sum_{t \geq 0} \gamma^t \Big( \underset{\hat{\mathbf{s}}_t, \hat{\mathbf{a}}_t \sim p(\hat{\mathbf{s}}_t, \hat{\mathbf{a}}_t)}{\mathbb{E}} \big[ r(\hat{\mathbf{s}}_t, \hat{\mathbf{a}}_t) \big] - \underset{\mathbf{s}_t, \mathbf{a}_t \sim \tilde{p}^{\mathrm{opc}}(\mathbf{s}_t, \mathbf{a}_t)}{\mathbb{E}} \big[ r(\mathbf{s}_t, \mathbf{a}_t) \big] \Big) \Big| \tag{22}$$

$$\leq \sum_{t \geq 0} \gamma^t \Big| \underset{\hat{\mathbf{s}}_t, \hat{\mathbf{a}}_t \sim p(\hat{\mathbf{s}}_t, \hat{\mathbf{a}}_t)}{\mathbb{E}} \big[ r(\hat{\mathbf{s}}_t, \hat{\mathbf{a}}_t) \big] - \underset{\mathbf{s}_t, \mathbf{a}_t \sim \tilde{p}^{\mathrm{opc}}(\mathbf{s}_t, \mathbf{a}_t)}{\mathbb{E}} \big[ r(\mathbf{s}_t, \mathbf{a}_t) \big] \Big| \tag{23}$$

Applying Lemma 8:

$$\leq \sum_{t \geq 0} \gamma^t L_r \mathcal{W}_2(p(\hat{\mathbf{s}}_t, \hat{\mathbf{a}}_t), \tilde{p}^{\mathrm{opc}}(\mathbf{s}_t, \mathbf{a}_t)) \tag{24}$$

Writing the joint distributions for state/action in terms of their conditional (i.e., policy) and marginal distributions $p(\mathbf{s}_t, \mathbf{a}_t) = \pi(\mathbf{a}_t \mid \mathbf{s}_t) p(\mathbf{s}_t)$:

$$= \sum_{t \geq 0} \gamma^t L_r \mathcal{W}_2(\pi(\hat{\mathbf{a}}_t \mid \hat{\mathbf{s}}_t) p(\hat{\mathbf{s}}_t), \pi(\mathbf{a}_t \mid \mathbf{s}_t) \tilde{p}^{\mathrm{opc}}(\mathbf{s}_t)) \tag{25}$$

Under the assumption that the policies $\pi$ are $L_\pi$-Lipschitz under the Wasserstein distance, application of Lemma 10 concludes the proof:

$$\leq \sum_{t \geq 0} \gamma^t L_r \sqrt{1 + L_\pi^2} \mathcal{W}_2(p(\hat{\mathbf{s}}_t), \tilde{p}^{\mathrm{opc}}(\mathbf{s}_t)). \tag{26}$$

$\square$

**Lemma 6** (Wasserstein Distance between Marginal State Distributions). *Let $\tilde{p}^{\mathrm{opc}}(\mathbf{s}_t)$ and $p(\hat{\mathbf{s}}_t)$ be the marginal state distributions at time $t$ when rolling out from the same initial state $\hat{\mathbf{s}}_0$ under the same policy with the OPC-model and the true environment, respectively. Assume that the underlying learned dynamics model is $L_f$-Lipschitz continuous with respect to both its arguments and the policy $\pi(\mathbf{a} \mid \mathbf{s})$ is $L_\pi$-Lipschitz with respect to $\mathbf{s}$ under the Wasserstein distance. If it further holds that the policy's (co-)variance $\mathrm{Var}[\pi(\mathbf{a} \mid \mathbf{s})] = \Sigma_\pi(\mathbf{s}) \in \mathbb{S}_+^{d_\mathcal{A}}$ is finite over the complete state space, i.e., $\max_{\mathbf{s} \in \mathcal{S}} \mathrm{trace}\{\Sigma_\pi(\mathbf{s})\} \leq \bar{\sigma}_\pi^2$, then the discrepancy between the marginal state distributions of the two models is bounded*

$$\mathcal{W}_2^2(\tilde{p}^{\mathrm{opc}}(\mathbf{s}_t), p(\hat{\mathbf{s}}_t)) \leq 2\sqrt{d_\mathcal{A}} \bar{\sigma}_\pi^2 L_f^2 \sum_{t'=0}^{t-1} (L_f^2 + L_\pi^2)^{t'} \tag{27}$$

*Proof.* We proof the Lemma by induction.

**Base Case:** $t = 1$  For the base case, we need to show that starting from the same initial state $\hat{\mathbf{s}}_0$ the following condition holds:

$$\mathcal{W}_2^2(\tilde{p}^{\mathrm{opc}}(\mathbf{s}_1), p(\hat{\mathbf{s}}_1)) \leq 2\sqrt{d_\mathcal{A}} \bar{\sigma}_\pi^2 L_f^2$$

For ease of readability, we define $\mathbf{z} = (\mathbf{s}, \mathbf{a})$ and use the notation $\mathrm{d}p(\mathbf{x}, \mathbf{y}) = p(\mathbf{x}, \mathbf{y}) \, \mathrm{d}\mathbf{x} \, \mathrm{d}\mathbf{y}$ whenever no explicit assumptions are made about the distributions.

$$\mathcal{W}_2(\tilde{p}^{\mathrm{opc}}(\mathbf{s}_1), p(\hat{\mathbf{s}}_1)) \tag{28}$$

$$= \mathcal{W}_2 \left( \iint \tilde{p}^{\mathrm{opc}}(\mathbf{s}_1 \mid \hat{\mathbf{z}}_0, \mathbf{z}_0) \, \mathrm{d}p(\hat{\mathbf{z}}_0, \mathbf{z}_0), \int p(\hat{\mathbf{s}}_1 \mid \hat{\mathbf{z}}_0) \, \mathrm{d}p(\hat{\mathbf{z}}_0) \right) \tag{29}$$

Recall that we can write both $\tilde{p}^{\mathrm{opc}}$ and $p$ as convolution between $p_\epsilon$ and a Dirac delta (see Lemma 4). Together with the identity Eq. (54), the Wasserstein distance for sums of random variables Eq. (53) and noting $\mathcal{W}_p(p_\epsilon(\epsilon), p_\epsilon(\epsilon)) = 0$:

$$\leq \mathcal{W}_2 \left( \iint \delta(\mathbf{s}_1 - [f(\hat{\mathbf{z}}_0) + \tilde{f}(\mathbf{z}_0) - \tilde{f}(\hat{\mathbf{z}}_0)]) \, \mathrm{d}p(\hat{\mathbf{z}}_0, \mathbf{z}_0), \int \delta(\hat{\mathbf{s}}_1 - f(\hat{\mathbf{z}}_0)) \, \mathrm{d}p(\hat{\mathbf{z}}_0) \right) \tag{30}$$

Squaring and using Lemma 9:

$$\mathcal{W}_2^2 \left( \iint \delta(\mathbf{s}_1 - [f(\hat{\mathbf{z}}_0) + \tilde{f}(\mathbf{z}_0) - \tilde{f}(\hat{\mathbf{z}}_0)]) \, \mathrm{d}p(\hat{\mathbf{z}}_0) p(\mathbf{z}_0), \int \delta(\hat{\mathbf{s}}_1 - f(\hat{\mathbf{z}}_0)) \, \mathrm{d}p(\hat{\mathbf{z}}_0) \right)$$

$$\leq \iint \| f(\hat{\mathbf{z}}_0) + \tilde{f}(\mathbf{z}_0) - \tilde{f}(\hat{\mathbf{z}}_0) - f(\hat{\mathbf{z}}_0) \|^2 \, \mathrm{d}p(\hat{\mathbf{z}}_0, \mathbf{z}_0) \tag{31}$$

$$= \iint \| \tilde{f}(\mathbf{z}_0) - \tilde{f}(\hat{\mathbf{z}}_0) \|^2 \, \mathrm{d}p(\hat{\mathbf{z}}_0, \mathbf{z}_0) \tag{32}$$

$$\leq L_f^2 \int \| \mathbf{s}_0 - \hat{\mathbf{s}}_0 \|^2 + \| \mathbf{a}_0 - \hat{\mathbf{a}}_0 \|^2 \, \mathrm{d}p(\hat{\mathbf{s}}_0, \hat{\mathbf{a}}_0, \mathbf{s}_0, \mathbf{a}_0) \tag{33}$$

We are assuming that the initial states of the trajectory rollouts coincide. The joint state/action distribution can then be written as $p(\hat{\mathbf{s}}_0, \hat{\mathbf{a}}_0, \mathbf{s}_0, \mathbf{a}_0) = p(\hat{\mathbf{s}}_0)\pi(\hat{\mathbf{a}}_0 \mid \hat{\mathbf{s}}_0)\delta(\mathbf{s}_0 - \hat{\mathbf{s}}_0)\pi(\mathbf{a}_0 \mid \mathbf{s}_0)$. Integrating with respect to $\mathbf{s}_0$ leads to:

$$= L_f^2 \int \| \mathbf{a}_0 - \hat{\mathbf{a}}_0 \|^2 p(\hat{\mathbf{s}}_0)\pi(\hat{\mathbf{a}}_0 \mid \hat{\mathbf{s}}_0)\pi(\mathbf{a}_0 \mid \hat{\mathbf{s}}_0) \, \mathrm{d}\hat{\mathbf{s}}_0 \, \mathrm{d}\hat{\mathbf{a}}_0 \, \mathrm{d}\mathbf{a}_0 \tag{34}$$

This term describes the mean squared distance between two random actions. Since we condition $\pi$ on the same state $\hat{\mathbf{s}}_0$, the policy distributions coincide. Define $\Delta \mathbf{a} = \mathbf{a}_0 - \hat{\mathbf{a}}_0$,

$$= L_f^2 \, \mathbb{E}_{\hat{\mathbf{s}}_0} \left[ \, \mathbb{E}_{\Delta \mathbf{a}} [\| \Delta \mathbf{a} \|^2] \right] \tag{35}$$

$$= L_f^2 \, \mathbb{E}_{\hat{\mathbf{s}}_0} \left[ \, \mathrm{trace}\{\mathrm{Var}[\Delta \mathbf{a}]\} \right] \tag{36}$$

Now $\mathrm{Var}[\Delta \mathbf{a}] = \mathrm{Var}[\pi(\hat{\mathbf{a}}_0 \mid \hat{\mathbf{s}}_0)] + \mathrm{Var}[\pi(\mathbf{a}_0 \mid \hat{\mathbf{s}}_0)] = 2\,\mathrm{Var}[\pi(\mathbf{a}_0 \mid \hat{\mathbf{s}}_0)]$

$$= 2L_f^2 \, \mathbb{E}_{\hat{\mathbf{s}}_0} \left[ \, \mathrm{trace}\{\mathrm{Var}[\pi(\mathbf{a}_0 \mid \hat{\mathbf{s}}_0)]\} \right] \tag{37}$$

This term is in fact less than Eq. (27) for $t = 0$, thus proofing the base case.

$$\leq 2\sqrt{d_{\mathcal{A}}} \bar{\sigma}_\pi^2 L_f^2 \tag{38}$$

**Inductive Step**  We will show that if the hypothesis holds for $t$ then it holds for $t + 1$ as well. We explicitly write the following intermediate bound such that its application in the proof is more apparent, i.e.,

$$\mathcal{W}_2^2(p(\hat{\mathbf{s}}_t), \tilde{p}^{\mathrm{opc}}(\mathbf{s}_t)) \leq \int \| \tilde{f}(\mathbf{z}_{t-1}) - \tilde{f}(\hat{\mathbf{z}}_{t-1}) \|^2 \, \mathrm{d}p(\hat{\mathbf{z}}_{t-1}, \mathbf{z}_{t-1}) \tag{39}$$

$$\leq 2\sqrt{d_{\mathcal{A}}} \bar{\sigma}_\pi^2 L_f^2 \sum_{t'=0}^{t-1} (L_f^2 + L_\pi^2)^{t'}, \tag{40}$$

where the first inequality immediately follows from the same reasoning as in the base case Eq. (28)–Eq. (32).

$$\mathcal{W}_2^2(p(\hat{\mathbf{s}}_{t+1}, p(\mathbf{s}_{t+1})) \tag{41}$$

$$\leq \int \| \tilde{f}(\mathbf{z}_t) - \tilde{f}(\hat{\mathbf{z}}_t) \|^2 \, \mathrm{d}p(\hat{\mathbf{z}}_t, \mathbf{z}_t) \tag{42}$$

$$\leq L_f^2 \int \| \mathbf{s}_t - \hat{\mathbf{s}}_t \|^2 \, \mathrm{d}p(\hat{\mathbf{z}}_t, \mathbf{z}_t) + L_f^2 \int \| \mathbf{a}_t - \hat{\mathbf{a}}_t \|^2 \, \mathrm{d}p(\hat{\mathbf{z}}_t, \mathbf{z}_t) \tag{43}$$

Applying Lemma 11 to the second integral

$$\leq (L_f^2 + L_\pi^2) \int \| \mathbf{s}_t - \hat{\mathbf{s}}_t \|^2 \, \mathrm{d}p(\hat{\mathbf{z}}_t, \mathbf{z}_t) + 2\sqrt{d_{\mathcal{A}}} L_f^2 \bar{\sigma}_\pi^2 \tag{44}$$

We predict along a consistent trajectory, i.e., $\mathbf{s}_t = \hat{\mathbf{s}}_t + \tilde{f}(\mathbf{s}_{t-1}, \mathbf{a}_{t-1}) - \tilde{f}(\hat{\mathbf{s}}_{t-1}, \hat{\mathbf{a}}_{t-1})$

$$\leq (L_f^2 + L_\pi^2) \int \| \tilde{f}(\mathbf{z}_{t-1}) - \tilde{f}(\hat{\mathbf{z}}_{t-1}) \|^2 \, \mathrm{d}p(\hat{\mathbf{z}}_{t-1}, \mathbf{z}_{t-1}) + 2\sqrt{d_{\mathcal{A}}} L_f^2 \bar{\sigma}_\pi^2 \tag{45}$$

Assume that the hypothesis Eq. (39) holds for $t$

$$\leq (L_f^2 + L_\pi^2) \times 2\sqrt{d_\mathcal{A}}\bar{\sigma}_\pi^2 L_f^2 \sum_{t'=0}^{t-1}(L_f^2 + L_\pi^2)^{t'} + 2\sqrt{d_\mathcal{A}}L_f^2\bar{\sigma}_\pi^2 \tag{46}$$

$$= 2\sqrt{d_\mathcal{A}}\bar{\sigma}_\pi^2 L_f^2 \big[1 + \sum_{t'=0}^{t-1}(L_f^2 + L_\pi^2)^{t'}\big] \tag{47}$$

$$= 2\sqrt{d_\mathcal{A}}\bar{\sigma}_\pi^2 L_f^2 \sum_{t'=0}^{t}(L_f^2 + L_\pi^2)^{t'} \tag{48}$$

$\square$

**Theorem 1.** *Let $\tilde{\eta}_\star^{\mathrm{opc}}$ and $\eta$ be the expected return under the generalized OPC-model Eq. (6) and the true environment, respectively. Assume that the learned model's mean transition function $\tilde{f}(\mathbf{s}_t, \mathbf{a}_t) = \mathbb{E}[\tilde{p}^{\mathrm{model}}(\mathbf{s}_{t+1} \mid \mathbf{s}_t, \mathbf{a}_t)]$ is $L_f$-Lipschitz and the reward $r(\mathbf{s}_t, \mathbf{a}_t)$ is $L_r$-Lipschitz. Further, if the policy $\pi(\mathbf{a}_t \mid \mathbf{s}_t)$ is $L_\pi$-Lipschitz with respect to $\mathbf{s}_t$ under the Wasserstein distance and its (co-)variance $\mathrm{Var}[\pi(\mathbf{a}_t \mid \mathbf{s}_t)] = \Sigma_\pi(\mathbf{s}_t) \in \mathbb{S}_+^{d_\mathcal{A}}$ is finite over the complete state space, i.e., $\max_{\mathbf{s}_t \in \mathcal{S}} \mathrm{trace}\{\Sigma_\pi(\mathbf{s}_t)\} \leq \bar{\sigma}_\pi^2$, then with $C_1 = \sqrt{2(1 + L_\pi^2)}L_f L_r$ and $C_2 = \sqrt{L_f^2 + L_\pi^2}$*

$$|\eta - \tilde{\eta}_\star^{\mathrm{opc}}| \leq \frac{\bar{\sigma}_\pi}{1 - \gamma} d_\mathcal{A}^{\frac{1}{4}} C_1 C_2^T \sqrt{T}. \tag{7}$$

*Proof.* From combining Lemmas 5 and 6 it follows that

$$|\eta - \tilde{\eta}_\star^{\mathrm{opc}}| \leq \sqrt{2\sqrt{d_\mathcal{A}}(1 + L_\pi^2)}\bar{\sigma}_\pi L_f L_r \sum_{t \geq 0} \gamma^t \sqrt{\sum_{t'=0}^{t}(L_f^2 + L_\pi^2)^{t'}} \tag{49}$$

with the shorthand notations $C_1 = \sqrt{2(1 + L_\pi^2)}L_f L_r$ and $C_2 = L_f^2 + L_\pi^2$

$$= C_1 |\mathcal{A}|^{\frac{1}{4}}\bar{\sigma}_\pi \sum_{t=0}^{T} \gamma^t \sqrt{\sum_{t'=0}^{t} C_2^{t'}} \tag{50}$$

Since $t \leq T$, we have that $\sum_{t'=0}^{t} C_2^{t'} \leq T C_2^T$

$$\leq C_1 |\mathcal{A}|^{\frac{1}{4}}\bar{\sigma}_\pi C_2^{\frac{T}{2}} \sqrt{T} \sum_{t=0}^{T} \gamma^t \tag{51}$$

Since $T \leq \infty$, we have with the geometric series $\sum_{t=0}^{T} \gamma^t \leq 1/(1 - \gamma)$

$$\leq \frac{C_1}{1 - \gamma}|\mathcal{A}|^{\frac{1}{4}} C_2^{\frac{T}{2}} \sqrt{T}\bar{\sigma}_\pi \tag{52}$$

$\square$

### B.4.3 DEFINITIONS, HELPFUL IDENTITIES AND SUPPORTING LEMMAS

Here we briefly summarize some basic definitions and properties that will be used throughout the following.

- The Wasserstein distance fulfills the properties of a metric: $\mathcal{W}_p(p_1, p_3) \leq W_p(p_1, p_2) + W_p(p_2, p_3)$.
- Wasserstein distance of sums of random variables (see, e.g., Mariucci & Reiß (2018, Corollary 1) for a proof):

$$\mathcal{W}_p(p_1 * \cdots * p_n, q_1 * \cdots * q_n) \leq \sum_{i=1}^{n} \mathcal{W}_p(p_i, q_i) \tag{53}$$

- For any function $g(\mathbf{z}, \hat{\mathbf{z}})$ we have

$$\iint p(\mathbf{s}_{t+1} - g(\mathbf{z}, \hat{\mathbf{z}}))\nu(\mathbf{z}, \hat{\mathbf{z}}) \, \mathrm{d}\mathbf{z} \, \mathrm{d}\hat{\mathbf{z}} = p * \iint \delta(\mathbf{s}_{t+1} - g(\mathbf{z}, \hat{\mathbf{z}}))\nu(\mathbf{z}, \hat{\mathbf{z}}) \, \mathrm{d}\mathbf{z} \, \mathrm{d}\hat{\mathbf{z}} \quad (54)$$

- For any multivariate random variable $\mathbf{z}_1$ and $\mathbf{z}_2$ with probability distributions $p(\mathbf{z}_1) = p_1(\mathbf{x})q(\mathbf{y})$ and $p(\mathbf{z}_2) = p_2(\mathbf{x})q(\mathbf{y})$, respectively, we have that (Panaretos & Zemel, 2019)

$$\mathcal{W}_2^2(p_1(\mathbf{x})q(\mathbf{y}), p_2(\mathbf{x})q(\mathbf{y})) = \mathcal{W}_2^2(p_1(\mathbf{x}), p_2(\mathbf{x})). \quad (55)$$

Further, the following Lemmas are helpful for the proof of Theorem 1.

**Lemma 7** (Kantorovich-Rubinstein (cf. Mariucci & Reiß (2018) Proposition 1.3)). *Let $X$ and $Y$ be integrable real random variables. Denote by $\mu$ and $\mu$ their laws [...]. Then the following characterization of the Wasserstein distance of order 1 holds:*

$$\mathcal{W}_1(\nu, \mu) = \sup_{\|\phi\|_{\mathrm{Lip}} \leq 1} \mathbb{E}_{x \sim \nu(\cdot)}[\phi(x)] - \mathbb{E}_{y \sim \mu(\cdot)}[\phi(y)], \quad (56)$$

*where the supremum is being taken over all $\phi$ satisfying the Lipschitz condition $|\phi(x) - \phi(y)| \leq |x - y|$, for all $x, y \in \mathbb{R}$.*

**Lemma 8.** *Let $f$ be $L_f$-Lipschitz with respect to a metric $d$. Then*

$$|\mathbb{E}_{x \sim \nu(\cdot)}[f(x)] - \mathbb{E}_{y \sim \mu(\cdot)}[f(y)]| \leq L_f \mathcal{W}_1(\nu, \mu) \leq L_f \mathcal{W}_2(\nu, \mu) \quad (57)$$

*Proof.* The first inequality is a direct consequence of Lemma 7 and the second inequality comes from the well-known fact that if $1 \leq p \leq q$, then $\mathcal{W}_p(\mu, \nu) \leq \mathcal{W}_q(\mu, \nu)$ (cf. Mariucci & Reiß (2018, Lemma 1.2)). $\square$

**Lemma 9.** *For any two functions $f(\mathbf{s})$ and $g(\mathbf{s})$ and probability density $p(\mathbf{s})$ that govern the distributions defined by*

$$p_1(\mathbf{x}_1) = \int \delta(\mathbf{x}_1 - f(\mathbf{s}))p(\mathbf{s}) \, \mathrm{d}\mathbf{s} \quad and \quad p_2(\mathbf{x}_2) = \int \delta(\mathbf{x}_2 - g(\mathbf{s}))p(\mathbf{s}) \, \mathrm{d}\mathbf{s}, \quad (58)$$

*it holds for any $q \geq 1$ that*

$$\mathcal{W}_q^q(p_1, p_2) \leq \int \|f(\mathbf{s}) - g(\mathbf{s})\|^q p(\mathbf{s}) \, \mathrm{d}\mathbf{s}. \quad (59)$$

*Proof.* We have

$$\mathcal{W}_q^q(p_1(\mathbf{x}_1), p_2(\mathbf{x}_2)) = \inf_{\gamma \in \Gamma(p_1, p_2)} \iint \|\xi_1 - \xi_2\|^q \gamma(\xi_1, \xi_2) \, \mathrm{d}\xi_1 \, \mathrm{d}\xi_2 \quad (60)$$

Enforcing the following structure on $\gamma(\xi_1, \xi_2)$ reduces the space of possible distributions: $\gamma(\xi_1, \xi_2) = \int \delta(\xi_1 - f(\mathbf{s}))\delta(\xi_2 - g(\mathbf{s}))p(\mathbf{s}) \, \mathrm{d}\mathbf{s}$, so that $\gamma(\xi_1, \xi_2) \in \Gamma(p_1, p_2)$ and thus

$$\leq \int \|\xi_1 - \xi_2\|^q \int \delta(\xi_1 - f(\mathbf{s}))\delta(\xi_2 - g(\mathbf{s}))p(\mathbf{s}) \, \mathrm{d}\mathbf{s} \, \mathrm{d}\xi_1 \, \mathrm{d}\xi_2 \quad (61)$$

Integrating over $\xi_1$ and $\xi_2$ yields

$$= \int \|f(\mathbf{s}) - g(\mathbf{s})\|^q p(\mathbf{s}) \, \mathrm{d}\mathbf{s} \quad (62)$$

$\square$

**Lemma 10.** *If the policy $\pi(\mathbf{a}_t \mid \mathbf{s}_t)$ is $L_\pi$-Lipschitz with respect to $\mathbf{s}_t$ under the Wasserstein distance, then with $p(\hat{\mathbf{s}}, \hat{\mathbf{a}}) = \pi(\hat{\mathbf{a}} \mid \hat{\mathbf{s}})p(\hat{\mathbf{s}})$ and $\tilde{p}(\mathbf{s}, \mathbf{a}) = \pi(\mathbf{a} \mid \mathbf{s})\tilde{p}(\mathbf{s})$,*

$$\mathcal{W}_2^2(p(\hat{\mathbf{s}}, \hat{\mathbf{a}}), \tilde{p}(\mathbf{s}, \mathbf{a})) \leq (1 + L_\pi^2)\mathcal{W}_2^2(p(\hat{\mathbf{s}}), \tilde{p}(\mathbf{s})) \quad (63)$$

*Proof.*

$$\mathcal{W}_2^2(p(\hat{\mathbf{s}}, \hat{\mathbf{a}}), \tilde{p}(\mathbf{s}, \mathbf{a})) \tag{64}$$

$$= \inf_{\gamma \in \Gamma(p(\hat{\mathbf{s}}, \hat{\mathbf{a}}), \tilde{p}(\mathbf{s}, \mathbf{a}))} \int \left[ \|\hat{\mathbf{a}} - \mathbf{a}\|^2 + \|\hat{\mathbf{s}} - \mathbf{s}\|^2 \right] \gamma(\hat{\mathbf{a}}, \mathbf{a}, \hat{\mathbf{s}}, \mathbf{s}) \, d\hat{\mathbf{a}} \, d\mathbf{a} \, d\hat{\mathbf{s}} \, d\mathbf{s} \tag{65}$$

Enforcing the following structure on $\gamma$ reduces the space of possible distributions: $\gamma(\hat{\mathbf{s}}, \mathbf{s}, \hat{\mathbf{a}}, \mathbf{a}) = \gamma(\hat{\mathbf{s}}, \mathbf{s})\gamma(\hat{\mathbf{a}}, \mathbf{a} \mid \hat{\mathbf{s}}, \mathbf{s})$ with $\gamma(\hat{\mathbf{s}}, \mathbf{s}) \in \Gamma(p(\hat{\mathbf{s}}), \tilde{p}(\mathbf{s}))$ and $\gamma(\hat{\mathbf{a}}, \mathbf{a}|\hat{\mathbf{s}}, \mathbf{s}) \in \Gamma(\pi(\hat{\mathbf{a}} \mid \hat{\mathbf{s}}), \pi(\mathbf{a} \mid \mathbf{s}))$.

$$\leq \inf_{\substack{\gamma(\hat{\mathbf{s}},\mathbf{s}) \in \Gamma(p(\hat{\mathbf{s}}), \tilde{p}(\mathbf{s})) \\ \gamma(\hat{\mathbf{a}},\mathbf{a}|\hat{\mathbf{s}},\mathbf{s}) \in \Gamma(\pi(\hat{\mathbf{a}}|\hat{\mathbf{s}}), \pi(\mathbf{a}|\mathbf{s}))}} \int \left\{ \int \left[ \|\hat{\mathbf{a}} - \mathbf{a}\|^2 \gamma(\hat{\mathbf{a}}, \mathbf{a} \mid \hat{\mathbf{s}}, \mathbf{s}) \, d\hat{\mathbf{a}} \, d\mathbf{a} \right\} + \|\hat{\mathbf{s}} - \mathbf{s}\|^2 \right] \gamma(\hat{\mathbf{s}}, \mathbf{s}) \, d\hat{\mathbf{s}} \, d\mathbf{s} \tag{66}$$

Interchange infimum and the integral: Rockafellar (1976, Theorem 3A)

$$= \inf_{\gamma(\hat{\mathbf{s}},\mathbf{s})} \int \left\{ \inf_{\gamma(\hat{\mathbf{a}},\mathbf{a}|\hat{\mathbf{s}},\mathbf{s})} \int \|\hat{\mathbf{a}} - \mathbf{a}\|^2 \gamma(\hat{\mathbf{a}}, \mathbf{a} \mid \hat{\mathbf{s}}, \mathbf{s}) \, d\hat{\mathbf{a}} \, d\mathbf{a} + \|\hat{\mathbf{s}} - \mathbf{s}\|^2 \right\} \gamma(\hat{\mathbf{s}}, \mathbf{s}) \, d\hat{\mathbf{s}} \, d\mathbf{s} \tag{67}$$

$$= \inf_{\gamma(\hat{\mathbf{s}},\mathbf{s}) \in \Gamma(p(\hat{\mathbf{s}}), \tilde{p}(\mathbf{s}))} \int \left\{ \mathcal{W}_2^2(\pi(\hat{\mathbf{a}} \mid \hat{\mathbf{s}}), \pi(\mathbf{a} \mid \mathbf{s})) + \|\hat{\mathbf{s}} - \mathbf{s}\|^2 \right\} \gamma(\hat{\mathbf{s}}, \mathbf{s}) \, d\hat{\mathbf{s}} \, d\mathbf{s} \tag{68}$$

Using the assumption that the action distribution is $L_\pi$-Lipschitz continuous under the Wasserstein metric with respect to the state $\mathbf{s}$:

$$\leq \inf_{\gamma(\hat{\mathbf{s}},\mathbf{s}) \in \Gamma(p(\hat{\mathbf{s}}), \tilde{p}(\mathbf{s}))} \int (1 + L_\pi^2) \|\hat{\mathbf{s}} - \mathbf{s}\|^2 ] \gamma(\hat{\mathbf{s}}, \mathbf{s}) \, d\hat{\mathbf{s}} \, d\mathbf{s} \tag{69}$$

$$= (1 + L_\pi^2) \mathcal{W}_2^2(p(\hat{\mathbf{s}}), \tilde{p}(\mathbf{s})) \tag{70}$$

$\square$

**Lemma 11** (Average Squared Euclidean Distance Between Actions). *If the policy $\pi(\mathbf{a} \mid \mathbf{s})$ is $L_\pi$-Lipschitz with respect to $\mathbf{s}$ under the Wasserstein distance and the policy's (co-)variance $\mathrm{Var}[\pi(\mathbf{a} \mid \mathbf{s})] = \Sigma_\pi(\mathbf{s}) \in \mathbb{S}_+^{d_\mathcal{A}}$ is finite over the complete state space, i.e., $\max_{\mathbf{s} \in \mathcal{S}} \mathrm{trace}\{\Sigma_\pi(\mathbf{s})\} \leq \bar{\sigma}_\pi^2$, then*

$$\mathbb{E}_{\substack{\hat{\mathbf{a}}_t \sim \pi(\hat{\mathbf{s}}_t) \\ \mathbf{a}_t \sim \pi(\mathbf{s}_t)}} \left[ \|\hat{\mathbf{a}}_t - \mathbf{a}_t\|^2 \right] \leq L_\pi^2 \|\hat{\mathbf{s}}_t - \mathbf{s}_t\|^2 + 2\sqrt{d_\mathcal{A}} \bar{\sigma}_\pi^2 \tag{71}$$

*Proof.* Straightforward application of Corollary 1 and Lemma 13 $\square$

**Lemma 12** (Average Squared Euclidean Distance). *Consider two random variables $\mathbf{x}, \mathbf{y}$ with distributions $p_\mathbf{x}, p_\mathbf{y}$, mean vectors $\mu_\mathbf{x}, \mu_\mathbf{y} \in \mathbb{R}^m$ and covariance matrices $\Sigma_\mathbf{x}, \Sigma_\mathbf{y} \in \mathbb{S}_+^m$, respectively. Then the average squared Euclidean distance between the two is*

$$\mathbb{E}_{\mathbf{x},\mathbf{y}} \left[ \|\mathbf{x} - \mathbf{y}\|^2 \right] = \|\mu_\mathbf{x} - \mu_\mathbf{y}\|^2 + \mathrm{trace}\{\Sigma_\mathbf{x} + \Sigma_\mathbf{y}\}. \tag{72}$$

*Proof.* Define $\mathbf{z} = \mathbf{x} - \mathbf{y}$ with mean $\mu_\mathbf{z} = \mu_\mathbf{x} - \mu_\mathbf{y}$ and variance $\Sigma_\mathbf{z} = \Sigma_\mathbf{x} + \Sigma_\mathbf{y}$.

$$\mathbb{E}\left[\|\mathbf{z}\|^2\right] = \mathbb{E}\left[\sum_i \mathbf{z}_i^2\right]$$

$$= \sum_i \mathbb{E}\left[\mathbf{z}_i^2\right]$$

$$= \sum_i \mathbb{E}[\mathbf{z}_i]^2 + \mathrm{Var}[\mathbf{z}_i]$$

$$= \mu_\mathbf{z}^\top \mu_\mathbf{z} + \mathrm{trace}\{\Sigma_\mathbf{z}\}$$

$$= \|\mu_\mathbf{x} - \mu_\mathbf{y}\|^2 + \mathrm{trace}\{\Sigma_\mathbf{x} + \Sigma_\mathbf{y}\}.$$

$\square$

**Theorem 2** (Gelbrich Bound (from Kuhn et al. (2019))). *If $\|\cdot\|$ is the Euclidean norm, and the distributions $p_\mathbf{x}$ and $p_\mathbf{y}$ have mean vectors $\mu_\mathbf{x}, \mu_\mathbf{y} \in \mathbb{R}^m$ and covariance matrices $\Sigma_\mathbf{x}, \Sigma_\mathbf{y} \in \mathbb{S}_+^m$, respectively, then*

$$\mathcal{W}_2(p_\mathbf{x}, p_\mathbf{y}) \geq \sqrt{\|\mu_\mathbf{x} - \mu_\mathbf{y}\|^2 + \operatorname{trace}\left\{\Sigma_\mathbf{x} + \Sigma_\mathbf{y} - 2\big(\Sigma_\mathbf{x}^{1/2}\Sigma_\mathbf{y}\Sigma_\mathbf{x}^{1/2}\big)^{1/2}\right\}}. \tag{73}$$

*The bound is exact if $p_\mathbf{x}$ and $p_\mathbf{y}$ are elliptical distributions with the same density generator.*

**Corollary 1.** *Consider the same setting as in Lemma 12, then the average squared Euclidean distance is bounded by*

$$\mathbb{E}_{\mathbf{x},\mathbf{y}}\left[\|\mathbf{x} - \mathbf{y}\|^2\right] \leq \mathcal{W}_2^2(p_\mathbf{x}, p_\mathbf{y}) + 2\operatorname{trace}\left\{\big(\Sigma_\mathbf{x}^{1/2}\Sigma_\mathbf{y}\Sigma_\mathbf{x}^{1/2}\big)^{1/2}\right\}. \tag{74}$$

*Proof.* Straightforward application of the results from Lemma 12 and Theorem 2. $\square$

**Lemma 13.** *If the policy's (co-)variance $\operatorname{Var}[\pi(\mathbf{a} \mid \mathbf{s})] = \Sigma_\pi(\mathbf{s}) \in \mathbb{S}_+^{d_\mathcal{A}}$ is finite over the complete state space, i.e., $\max_{\mathbf{s} \in \mathcal{S}} \operatorname{trace}\{\Sigma_\pi(\mathbf{s})\} \leq \bar{\sigma}_\pi^2$, then*

$$\operatorname{trace}\left\{\big(\Sigma_\pi(\hat{\mathbf{s}})^{1/2}\Sigma_\pi(\mathbf{s})\Sigma_\pi(\hat{\mathbf{s}})^{1/2}\big)^{1/2}\right\} \leq \sqrt{d_\mathcal{A}}\,\bar{\sigma}_\pi^2 \tag{75}$$

*Proof.*

$$\operatorname{trace}\left\{\big(\Sigma_\pi(\hat{\mathbf{s}})^{1/2}\Sigma_\pi(\mathbf{s})\Sigma_\pi(\hat{\mathbf{s}})^{1/2}\big)^{1/2}\right\} \tag{76}$$

The trace of a matrix is the same as the sum of its eigenvalues, and the square root of a matrix has eigenvalues that are square root of its eigenvalues. From Jensen's inequality we know that $\sum_{i=1}^{d_\mathcal{A}} \sqrt{\lambda_i} \leq \sqrt{d_\mathcal{A} \sum_{i=1}^{d_\mathcal{A}} \lambda_i}$ and consequently it holds for a matrix $\mathbf{M} \in \mathbb{R}^{d_\mathcal{A} \times d_\mathcal{A}}$ that $\operatorname{trace}\{\mathbf{M}^{1/2}\} \leq \sqrt{d_\mathcal{A} \operatorname{trace}\{\mathbf{M}\}}$ , so that

$$\leq \sqrt{d_\mathcal{A} \operatorname{trace}\{\Sigma_\pi(\hat{\mathbf{s}})^{1/2}\Sigma_\pi(\mathbf{s})\Sigma_\pi(\hat{\mathbf{s}})^{1/2}\}} \tag{77}$$

The trace is invariant under cyclic permutation

$$= \sqrt{d_\mathcal{A} \operatorname{trace}\{\Sigma_\pi(\hat{\mathbf{s}})\Sigma_\pi(\mathbf{s})\}} \tag{78}$$

Since both matrices are positive semi-definite, it follows from the Cauchy-Schwartz inequality that

$$\leq \sqrt{d_\mathcal{A} \operatorname{trace}\{\Sigma_\pi(\hat{\mathbf{s}})\} \operatorname{trace}\{\Sigma_\pi(\mathbf{s})\}} \tag{79}$$

By assumption, the covariance matrices' traces are bounded

$$\leq \sqrt{d_\mathcal{A}}\,\bar{\sigma}_\pi^2 \tag{80}$$

$\square$

## B.5 Model Errors in OPC

While Theorem 1 highlights that OPC counteracts the on-policy error in predicted performance, for stochastic policies we use the Lipschitz continuity of the model to upper-bound errors. In this section, we look at the impact of model errors in combination with OPC. Specifically we focus on the one-step prediction case from a known initial state $\hat{\mathbf{s}}_0$. There, while for the model $\tilde{p}^{\text{model}}$ without OPC the prediction error only depends on the quality of the model, with OPC it is instead the minimum of the model error and the policy variance. This is advantageous, since typical environments tend to have more states than actions, so that the trace of the policy variance can be significantly smaller than the full-state model error.

**Lemma 14.** *Under the assumptions of Theorem 1, starting from an initial state $\hat{\mathbf{s}}_0$ the following condition holds:*

$$\mathcal{W}_2^2(\tilde{p}_\star^{\text{opc}}(\mathbf{s}_1), p(\hat{\mathbf{s}}_1))$$

$$\leq \min\left(\mathcal{O}\Big(\operatorname{trace}\{\operatorname{Var}[\pi(\cdot \mid \hat{\mathbf{s}}_0)]\}\Big), \mathcal{O}\Big(\underbrace{\int \|f(\hat{\mathbf{s}}_0, \mathbf{a}) - \tilde{f}(\hat{\mathbf{s}}_0, \mathbf{a})\|^2 \,\mathrm{d}\pi(\mathbf{a} \mid \hat{\mathbf{s}}_0)}_{\text{One-step model error}}\Big)\right)$$

*Proof.* The first term in the minimum follows directly from the base-case of Lemma 6. For the second term, follow the same derivation, but note that under the distribution $p(\hat{\mathbf{s}}_0, \hat{\mathbf{a}}_0, \mathbf{s}_0, \mathbf{a}_0) = p(\hat{\mathbf{s}}_0)\pi(\hat{\mathbf{a}}_0 \mid \hat{\mathbf{s}}_0)\delta(\mathbf{s}_0 - \hat{\mathbf{s}}_0)\pi(\mathbf{a}_0 \mid \mathbf{s}_0)$ we have $\int p(\hat{\mathbf{s}}_1 \mid \hat{\mathbf{z}}_0)\,\mathrm{d}p(\hat{\mathbf{z}}_0) = \int p(\mathbf{s}_1 \mid \mathbf{z}_0)\,\mathrm{d}p(\mathbf{z}_0)$. Inserting this into the r.h.s. of Eq. (28) and following the same steps we obtain

$$
\mathcal{W}_2^2(\tilde{p}_\star^{\mathrm{opc}}(\mathbf{s}_1), p(\hat{\mathbf{s}}_1))
$$
$$
\leq \mathcal{W}_2^2\left(\iint \delta(\mathbf{s}_1 - [f(\hat{\mathbf{z}}_0) + \tilde{f}(\mathbf{z}_0) - \tilde{f}(\hat{\mathbf{z}}_0)])\,\mathrm{d}p(\hat{\mathbf{z}}_0)p(\mathbf{z}_0), \int \delta(\mathbf{s}_1 - f(\mathbf{z}_0))\,\mathrm{d}p(\mathbf{z}_0)\right) \quad (81)
$$
$$
\leq \iint \|f(\hat{\mathbf{z}}_0) + \tilde{f}(\mathbf{z}_0) - \tilde{f}(\hat{\mathbf{z}}_0) - f(\mathbf{z}_0)\|^2\,\mathrm{d}p(\hat{\mathbf{z}}_0, \mathbf{z}_0) \quad (82)
$$
$$
= \iint \|f(\hat{\mathbf{z}}_0) - \tilde{f}(\hat{\mathbf{z}}_0)\|^2 + \|f(\mathbf{z}_0) - \tilde{f}(\mathbf{z}_0)\|^2\,\mathrm{d}p(\hat{\mathbf{z}}_0, \mathbf{z}_0) \quad (83)
$$
$$
= 2 \int \|f(\hat{\mathbf{z}}_0) - \tilde{f}(\hat{\mathbf{z}}_0)\|^2\,\mathrm{d}p(\mathbf{z}_0) \quad (84)
$$
$$
= 2 \int \|f(\hat{\mathbf{s}}_0, \mathbf{a}) - \tilde{f}(\hat{\mathbf{s}}_0, \mathbf{a})\|^2\,\mathrm{d}p(\mathbf{a} \mid \hat{\mathbf{s}}_0) \quad (85)
$$

$\square$

Note the additional factor of two in front of the upper bound on the model error, which comes from using the model 'twice': once with $\hat{\mathbf{z}}$ and once with $\mathbf{z}$. In practice we do not see any adverse effects of this error, presumably because either the variance of the policy is sufficiently small, or due to the upper bound being lose in practice.

## C   MOTIVATING EXAMPLE – IN-DEPTH ANALYSIS

In this section, we re-visit the motivating example presented in Section 4.1 of the main paper. For completeness, we re-state all assumptions that lead to the simplified system at hand. We continue with an analysis of the reward landscape and how OPC influences its shape. Next, we investigate how an increasing mismatch of the dynamics model impacts the gradient error. In addition to the result presented in the main paper, we here show the influence of different model errors, i.e., $\Delta\mathbf{A}$ as well as $\Delta\mathbf{B}$. While OPC is motivated for the use case of on-policy RL algorithms, we further show that the resulting gradients are robust with respect to differences in data-generating and evaluation policy, i.e., the off-policy setting. Lastly, we state the the signed gradient distance that we use for evaluation of the gradient errors, state the relevant theorem for determining the closed-loop stability of linear systems, as well as all numerical values used for the motivating example.

### C.1   SETUP

Here, we assume a linear system with deterministic dynamics
$$
p(\mathbf{s}_{t+1} \mid \mathbf{s}_t, \mathbf{a}_t) = \delta(\mathbf{A}\mathbf{s}_t + \mathbf{B}\mathbf{a}_t \mid \mathbf{s}_t, \mathbf{a}_t), \quad \rho_0(\mathbf{s}_0) = \delta(\mathbf{s}_0) \quad (86)
$$
with $\mathbf{A}, \mathbf{B} \in \mathbb{R}$ and $\delta(\cdot)$ denoting the Dirac-delta distribution. The linear policy and bell-shaped reward are given by the following equations
$$
\pi_\theta(\mathbf{a}_t \mid \mathbf{s}_t) = \delta(\theta\mathbf{s}_t \mid \mathbf{s}_t) \text{ with } \theta \in \mathbb{R} \quad \text{and} \quad r(\mathbf{s}_t, \mathbf{a}_t) = \exp\left\{-\left(\frac{\mathbf{s}_t}{\sigma_r}\right)^2\right\}. \quad (87)
$$

Further, we assume to have access to an approximate dynamics model $\tilde{p}$
$$
\tilde{p}(\mathbf{s}_{t+1} \mid \mathbf{s}_t, \mathbf{a}_t) = \delta((\mathbf{A} + \Delta\mathbf{A})\mathbf{s}_t + (\mathbf{B} + \Delta\mathbf{B})\mathbf{a}_t \mid \mathbf{s}_t, \mathbf{a}_t), \quad (88)
$$
where $\Delta\mathbf{A}, \Delta\mathbf{B}$ quantify the mismatch between the approximate model and the true system. For completeness, the (deterministic) policy gradient is defined as
$$
\nabla_\theta \frac{1}{T} \sum_{t=0}^{T-1} r(\mathbf{s}_t, \mathbf{a}_t), \quad (89)
$$

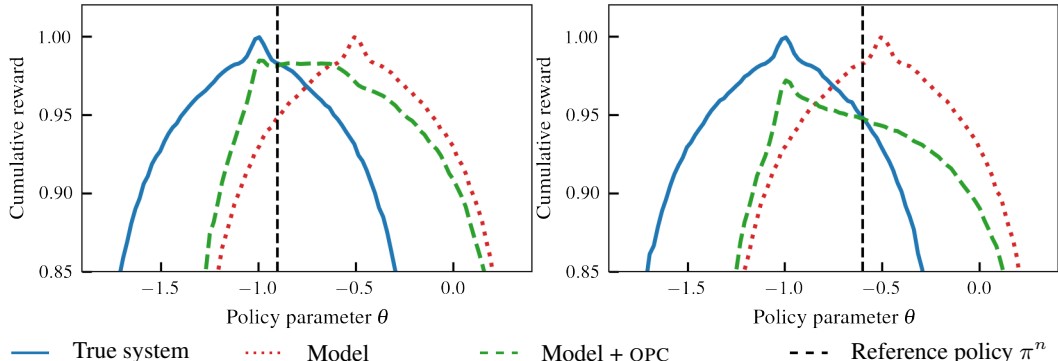

Figure 7: Cumulative reward for different systems as a function of the policy parameter. The reference trajectory that is used for OPC was generated by $\pi_\theta^n$ (denoted by the black dashed line). The model mismatch between the true system and the approximated model is $\Delta \mathbf{A} = 0.5, \Delta \mathbf{B} = 0.0$.

where the state/action pairs are obtained by simulating any of the two above models for $T$ time-steps and following policy $\pi_\theta^n$ resulting in the trajectory $\tilde{\tau}^n = \{(\mathbf{s}_t^n, \mathbf{a}_t^n)\}_{t=0}^{T-1}$. Because both the model and policy are deterministic, we can compute the analytical policy gradient from only one rollout.

## C.2 Reward Landscapes

In a first step, we will look at the cumulative rewards as a function of the policy parameter for the different systems at hand: 1) the true system, 2) the approximate model without OPC and 3) the approximate model with OPC. Further, let's assume that the model mismatch is fixed to some arbitrary value. The resulting reward landscapes are shown in Fig. 7. We would like to emphasize several key aspects in the plots: First, as one would expect the model mismatch leads to different optimal policies as well as misleading policy gradients for large parts of the policy parameter space. Second, the reward landscape for the model with OPC depends on the respective reference policy $\pi^n$ that was used to generate the data for the corrections. Consequently, the correct reward is recovered at $\theta = \theta^n$. More importantly, the OPC reshape the reward landscape such that the policy gradients point towards the correct optimum (left plot). Lastly, even when using OPC the policy gradient's sign is not guaranteed to have the correct sign (right plot). The extent of this effect strongly depends on the model mismatch, which we will investigate in the next section.

## C.3 Influence of Model Error

As shown in the previous section, the estimated policy gradient depends on the current policy as well as the mismatch between the true system and the approximate model. Fig. 2 depicts the (signed) differences between the true policy gradient as well as the approximated gradient as a function of model mismatch and the reference policy. Here, the opacity of the background denotes the magnitude of the error and the color denotes if the true and estimated gradient have the same (blue) or oppposite (red) sign. In the context of policy learning, the sign of the gradient is more relevant than the actual magnitude due to internal re-scaling of the gradients in modern implementations of stochastic optimizers such as Adam (Kingma & Ba, 2015). In our example, even for negligible model errors (either in $\Delta \mathbf{A}$ or $\Delta \mathbf{B}$), the model-based approach can lead to gradient estimates with the opposite sign, indicated by the large red areas for the left figures in Fig. 2. On the other hand, applying OPC to the model, we gradient estimates are significantly more robust with respect to errors in the dynamics.x

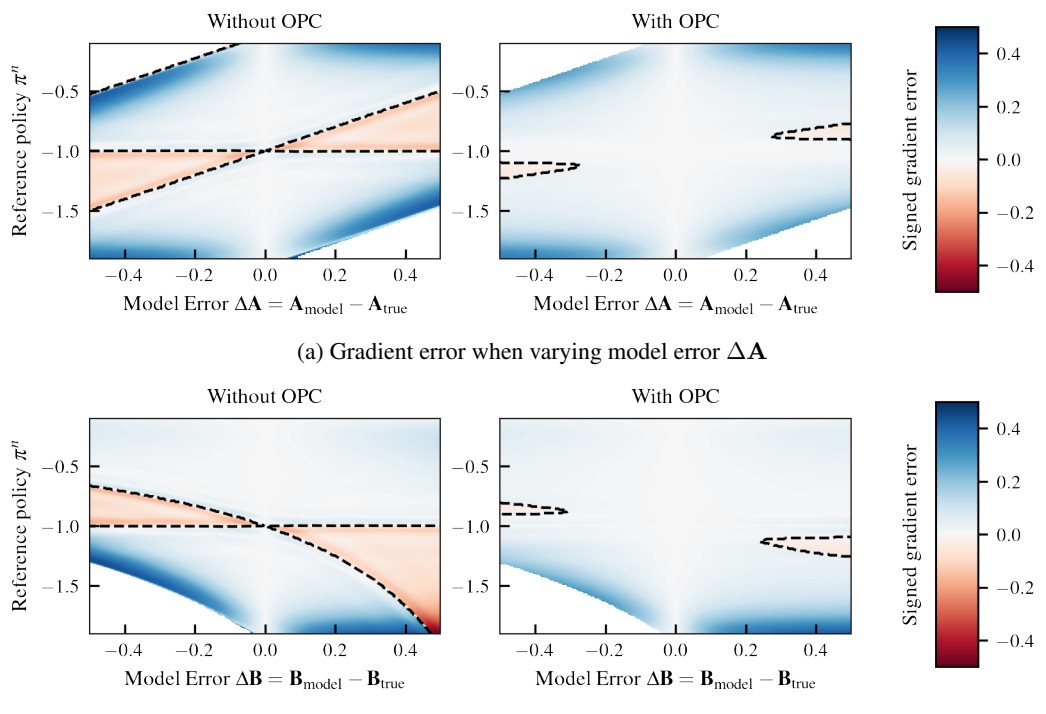

(a) Gradient error when varying model error $\Delta \mathbf{A}$

(b) Gradient error when varying model error $\Delta \mathbf{B}$

Figure 8: Signed gradient error (see Equation equation 90) when using the approximate model to estimate the policy gradient without (left) and with (right) on-policy corrections (OPCs). Using OPCs increases the robustness of the gradient estimate with respect to the model error.

## C.4 INFLUENCE OF OFF-POLICY ERROR

Until now we have considered the case in which the reference trajectory used for OPC is generated with the same policy as the one used for gradient estimation, i.e., the on-policy setting. In this case, we have observed that the true return could be recovered (see Fig. 7) when using OPC and that the gradient estimates are less sensitive to model errors (see Fig. 8). The off-policy case corresponds to the policy gains in Fig. 7 that are different from the reference policy $\pi^n$ indicated by the dashed line. Fig. 9 summarizes the results for the off-policy setting. Here, we varied the policy error and the reference policy itself for varying model errors. Note that for the correct model, we always recover the true gradient. But also for inaccurate models, the gradient estimates retain a good quality in most cases, with the exception for some model/policy combinations that are close to unstable.

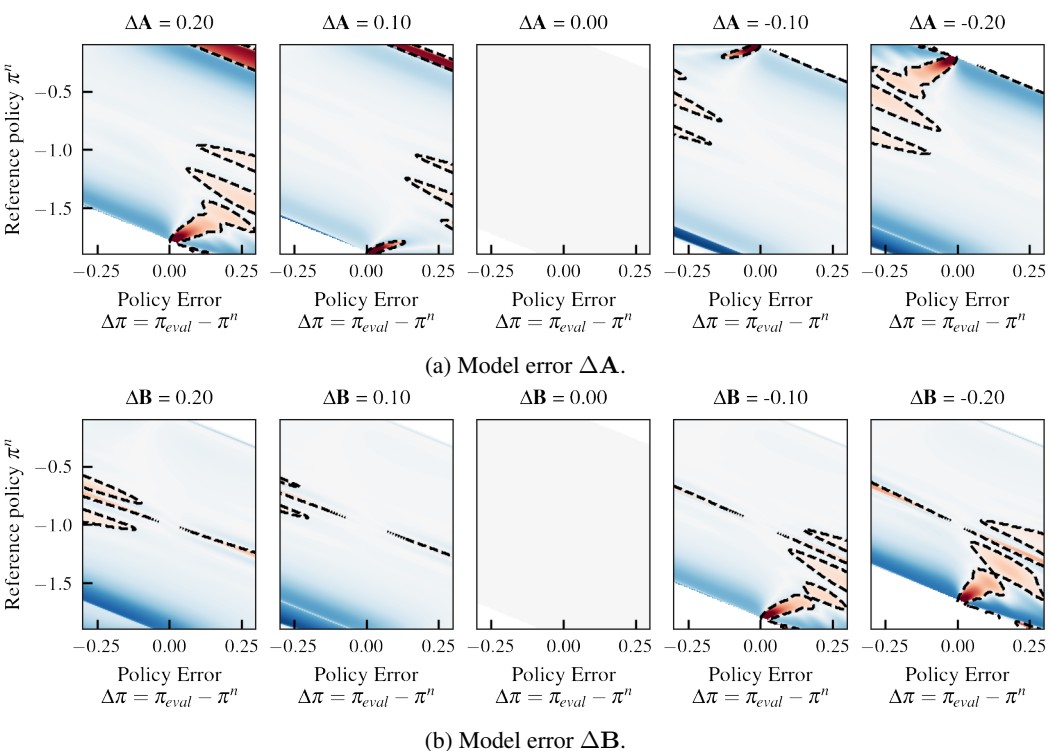

(a) Model error $\Delta\mathbf{A}$.

(b) Model error $\Delta\mathbf{B}$.

Figure 9: Signed gradient error due to off-policy data when using OPC. Note that we retain the true gradient in case of no model error.

## C.5 ADDITIONAL INFORMATION

Next, we provide some additional information about how we compute gradient distances, properties of linear systems, and exact numerical values used.

### C.5.1 COMPUTING THE SIGNED GRADIENT DISTANCE

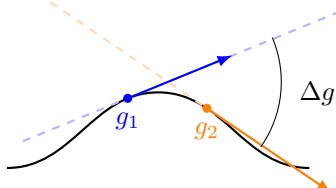

Figure 10: Sketch depicting the signed gradient distance Eq. (90). In this particular case, gradient $g_1$ is positive and $g_2$ is negative.

In order to compare two (1-dimensional) gradients in terms of sign and magnitude, we use the following formula

$$d(g_1, g_2) = \frac{1}{\pi} \begin{cases} \text{sign}(g_2) \cdot \Delta g, & \text{if } g_1 = 0 \\ \text{sign}(g_1) \cdot \Delta g, & \text{if } g_2 = 0 \\ \text{sign}(g_1 \cdot g_2) \cdot \Delta g, & \text{otherwise} \end{cases}, \quad \text{with} \quad \Delta g = |\arctan g_1 - \arctan g_2|. \quad (90)$$

The magnitude of this quantity depends on the normalized difference between the tangent's angles $\Delta g$ and is positive for gradients with the same sign and vice versa it is negative for gradients with opposing signs. See also Fig. 10 for a sketch.

### C.5.2 DETERMINING THE CLOSED-LOOP STABILITY FOR LINEAR SYSTEMS

For linear and deterministic systems, we can easily check if the system is (asymptotically) stable for a particular linear policy using the following standard result from linear system theory:

**Theorem 3** (Exponential stability for linear time-invariant systems (Callier & Desoer, 1991)). *The solution of $\mathbf{x}_{t+1} = \mathbf{F}\mathbf{x}_t$ is exponentially stable if and only if $\sigma(\mathbf{F}) \subset D(0, 1)$, i.e., every eigenvalue of $\mathbf{F}$ has magnitude strictly less than one.*

In our setting, this means that the closed-loop systems fulfilling the following are *unstable*,

$$|\mathbf{A} + \Delta\mathbf{A} + (\mathbf{B} + \Delta\mathbf{B})\theta| > 1, \tag{91}$$

i.e., the state and input grow exponentially. We therefore refrain from including unstable system in the results to avoid numerical issues for the gradients' computation. The respective areas in the plots are not colored, see e.g., bottom left corner in Fig. 8b.

### C.5.3 NUMERICAL VALUES

The numerical values for all parameters used in the motivating example are given as follows:

- True system dynamics: $\mathbf{A} = 1.0, \mathbf{B} = 1.0$
- Initial condition: $\mathbf{s}_0 = 1.0$
- Reward width parameter: $\sigma_r = 0.05$
- Optimal policy gain: $\theta^* = -1.0$
- Rollout horizon: $T = 60$

# D  ADDITIONAL EXPERIMENTAL RESULTS

In this section, we provide additional experimental results that did not fit into the main body of the paper.

## D.1  COMPARISON WITH OTHER BASELINE ALGORITHMS

Fig. 11 shows our method compared to a range of baseline algorithms. The results for all baselines were obtained from Janner et al. (2019) via personal communication. Note that all results are presented in terms of mean and standard deviation. The comparison includes the following methods:

- MBPO (Janner et al., 2019),
- PETS (Chua et al., 2018),
- SAC (Haarnoja et al., 2018),
- PPO (Schulman et al., 2017),
- STEVE (Buckman et al., 2018),
- SLBO (Luo et al., 2019).

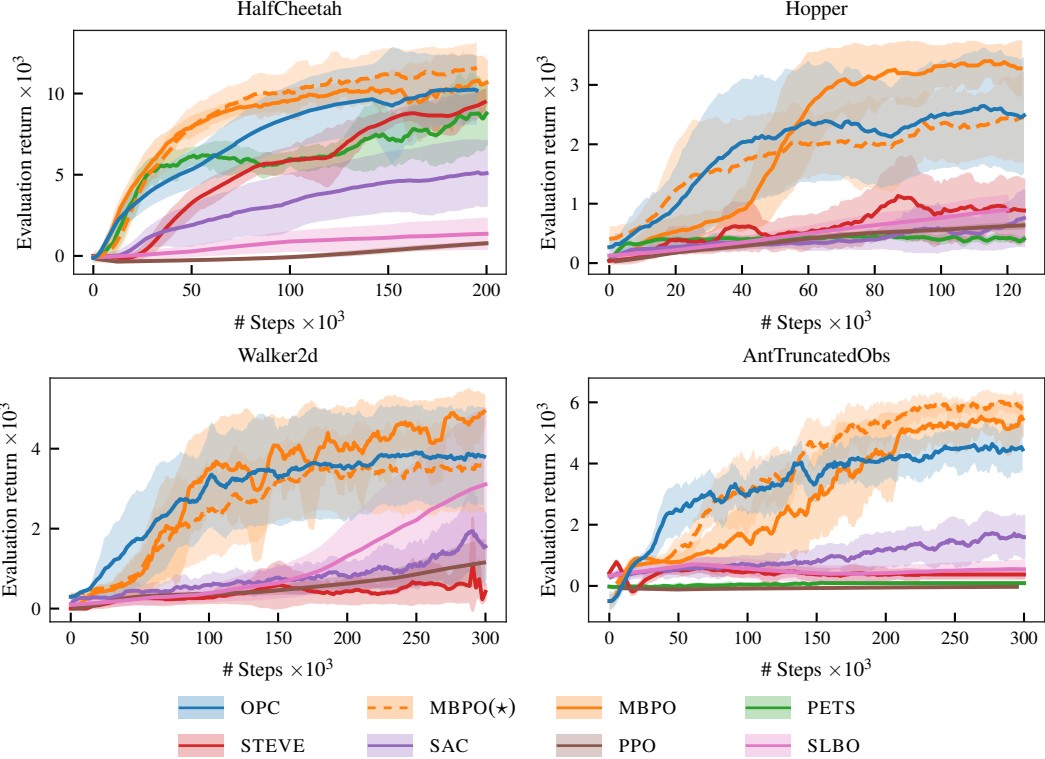

Figure 11: Comparison of OPC against a range of baseline methods on three MuJoCo environments. We present the mean and standard deviation across 5 independent experiments (10 for OPC and MBPO(⋆)). The original data for MBPO and the other baselines were provided by Janner et al. (2019). Solid lines represent the mean and the shaded areas correspond to mean ± one standard deviation.

## D.2 ABLATION - RETAIN EPOCHS

One of the hyperparameters that we found is critical to both OPC and MBPO($\star$), is `retain epochs`, i.e., the number of epochs that are kept in the data buffer for the simulated data generated with the $\tilde{p}^{\text{x}}$ with $\text{x} = \{\text{OPC}, \text{model}\}$. The results for a comparison are shown in Fig. 12. For MBPO($\star$), we found that for some environments (HalfCheetah, AntTruncatedObs) smaller values for `retain epochs` are helpful, i.e., simulated data is almost only on-policy, and for other environments larger values are beneficial (Hopper, Walker2d). For OPC on the other hand, we found that `retain epochs = 50` almost always leads to better results.

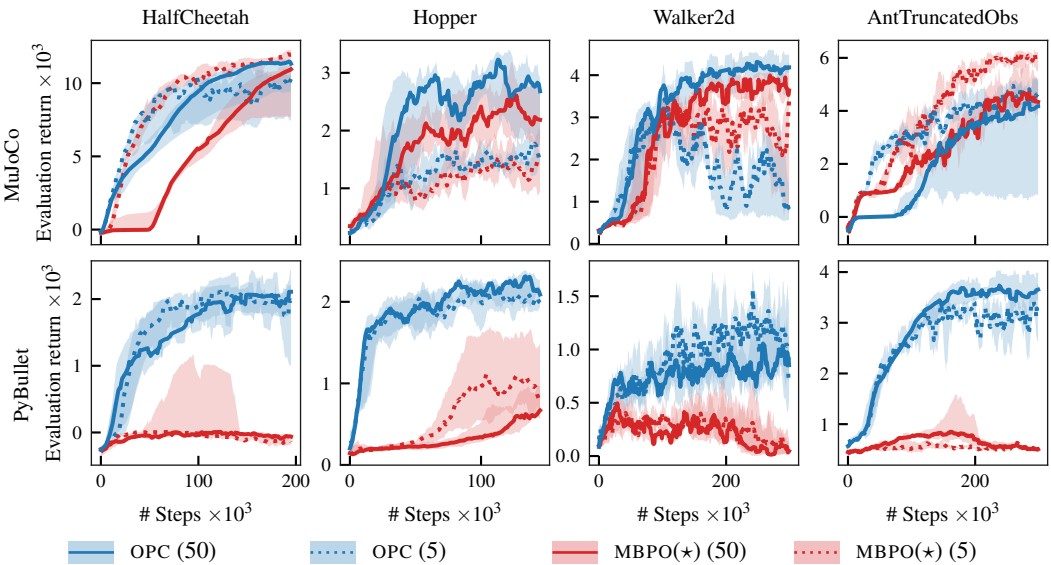

Figure 12: Ablation study for OPC and MBPO($\star$) on four environments from the MuJoCo control suite (top row) and their respective PyBullet implementations (bottom row). We vary the `retain epochs` hyperparameter (indicated by the number in the parentheses behind the legend entries), i.e., the number of epochs that are kept in the data buffer for the simulated data.

## D.3 INFLUENCE OF STATE REPRESENTATION: IN-DEPTH ANALYSIS

In the following, we will have a closer look at the surprising result from Fig. 4 (left). In there, the results indicate that MBPO($\star$) is not able to learn a stabilizing policy within the first 7'500 steps on the RoboSchool variant of the CartPole environment. We hypothesize that the failure of MBPO($\star$) is due to a mismatch of the simulated data and the true state distribution. As a result, the policy that is optimized with the simulated data cannot stabilize the pole in the true environment.

To validate our hypothesis, we perform the following experiment: First, we train a policy $\pi^*$ that we know performs well on the true environment, leading to the maximum evaluation return of 1000. With this policy, we roll out a reference trajectory on the true environment $\tau^{\text{ref}} = \{(\hat{\mathbf{s}}_t, \hat{\mathbf{a}}_t)\}_{t=0}^T$. To perform branched rollouts with the respective methods, OPC and MBPO($\star$), we use the learned transition models after 20 epochs (5'000 time steps) that were logged during a full learning process for each method. We then perform 100 branched rollouts of length $H = 20$ starting from randomly sampled initial states of the reference trajectories.

Fig. 13 shows the difference between the true and predicted state trajectories (median and 95$^{\text{th}}$ percentiles) for each state in $[x, \cos(\vartheta), \sin(\vartheta), \dot{x}, \dot{\vartheta}]$. Since we start each branched rollout from a state on the real environment, the initial error is always zero. Across all states, the errors are drastically reduced when using OPC. Fig. 14 shows the predicted trajectories for the cosine of the pole's angle, $\cos(\vartheta)$. For MBPO($\star$), we observe that the trajectories often diverge and attain values that are clearly out of distribution, i.e., $\cos(\vartheta) > 1$.

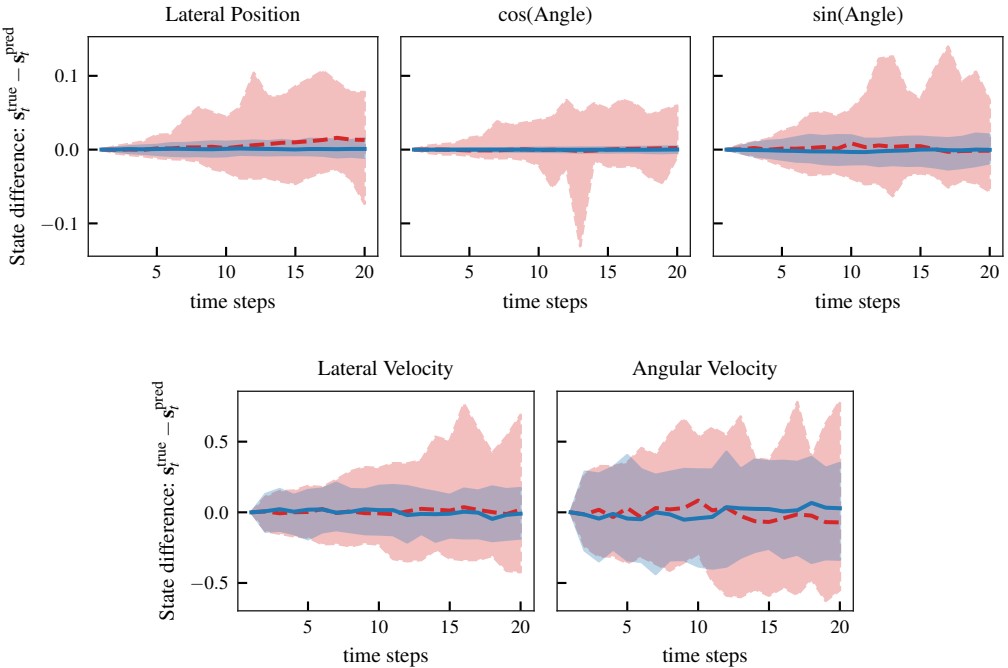

Figure 13: Difference in state trajectories $\mathbf{s}_t^{\mathrm{true}} - \mathbf{s}_t^{\mathrm{pred}}$ from branched rollouts using a fixed policy of length $H = 20$ on the RoboSchool environment (cf. Fig. 4 (left)) with $\mathbf{s} = [x, \cos(\vartheta), \sin(\vartheta), \dot{x}, \dot{\vartheta}]$. The solid lines show the median across 100 rollouts and the shaded areas represent the $95^{\mathrm{th}}$ percentiles. With OPC (——), the simulated rollouts follows the true state trajectories much closer, whereas with MBPO($\star$) (- - -) the prediction errors quickly accumulate over time.

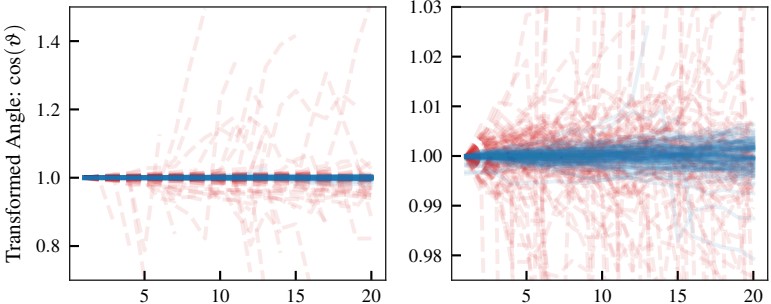

Figure 14: Trajectories of the second state $(\cos(\vartheta))$ from 100 branched rollouts using a fixed policy of length $H = 20$ on the RoboSchool environment (cf. Fig. 4 (left)). Both plots present the same data but the differ in terms of scaling of the ordinate. With OPC (——), the respective trajectories remain around values close to one, which corresponds to the upright position of the pendulum. When using the standard predictive model from MBPO($\star$) (- - -), the state trajectories often diverge and the rollouts are terminated prematurely.

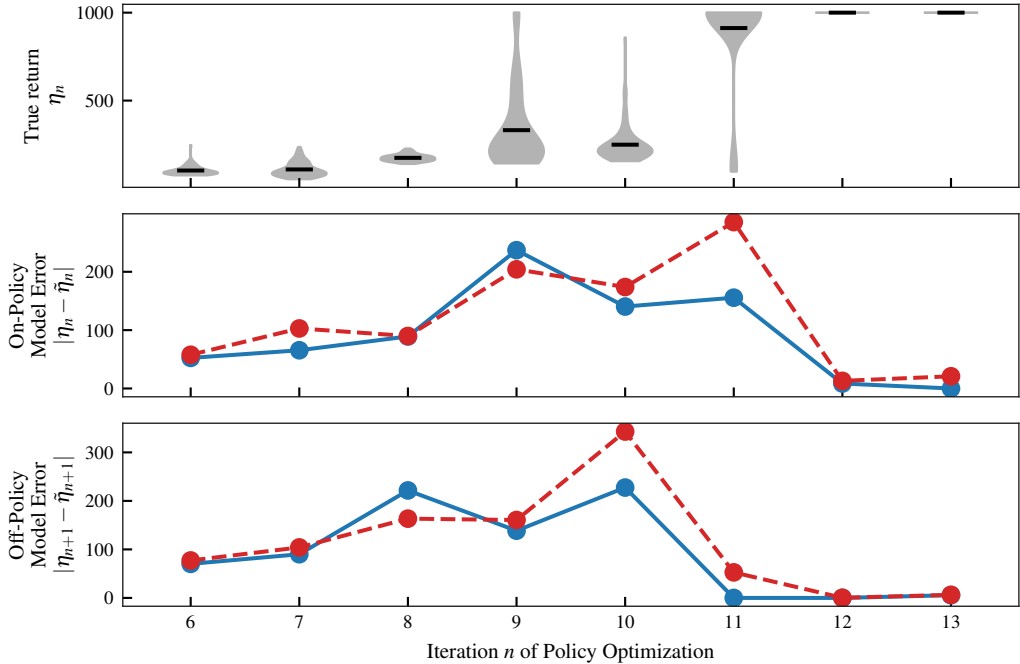

Figure 15: Empirical evaluation of the error terms in the policy improvement bound in Eq. (4) for the CartPole environment (PyBullet). We evaluate the respective terms for a sequence of policies that were obtained during different iterations $n$ from a full policy optimization run. The respective returns $\tilde{\eta}_{n+1}, \eta_n, \tilde{\eta}_{n+1}, \eta_{n+1}$ are approximated as the mean from 100 rollouts on the true environment and the respective models, $\tilde{p}_n^{\mathrm{opc}}$ (——) and $\tilde{p}_n^{\mathrm{model}}$ (- - -). For the return on the true environment $\eta_n$ (top), we additionally show the sample distribution of the rollouts' returns. This nicely demonstrates how the policy smoothly transitions from failing consistently ($n \leq 8$) to successfully stabilizing the pole ($n \geq 12$). Additionally, note that the on-policy model error is almost always smaller for OPC compared to MBPO($\star$), which supports the theoretical motivation that our method is build upon.

## D.4 IMPROVEMENT BOUND: EMPIRICAL INVESTIGATION

In this section, we empirically investigate to what extent OPC is able to tighten the policy improvement bound Eq. (4) compared to pure model-based approaches. As mentioned in the main paper, the motivation behind OPC is to reduce the on-policy error and we assume that the off-policy error is not affected too badly by the corrected transitions. Generally speaking, it is difficult to quantify a-priori how OPC compares to a purely model-based approach as this depends on the generalization capabilities of the learned dynamics model, the environment itself and the reward signal.

Here, we analyze the CartPole environment (PyBullet implementation) and estimate the respective error terms that appear in the policy improvement bound Eq. (4). To this end, we roll out a sequence of policies $\pi_n$ on the true environment (to estimate $\eta_n$ and $\eta_{n+1}$) and on the learned model with and without OPC (to estimate $\tilde{\eta}_n$ and $\tilde{\eta}_{n+1}$). The sequence of policies was obtained during a full policy optimization run (the policy was logged after each update) and we roll out each policy 100 times on the respective environment/models. The corresponding learned transition models were similarly logged during a full policy optimization. For OPC, we estimate the off-policy return $\tilde{\eta}_{n+1}$ using the learned model from iteration $n$ and reference trajectories from the true environment that were collected under $\pi_n$, but then roll out the model with $\pi_{n+1}$. The results are shown in Fig. 15.

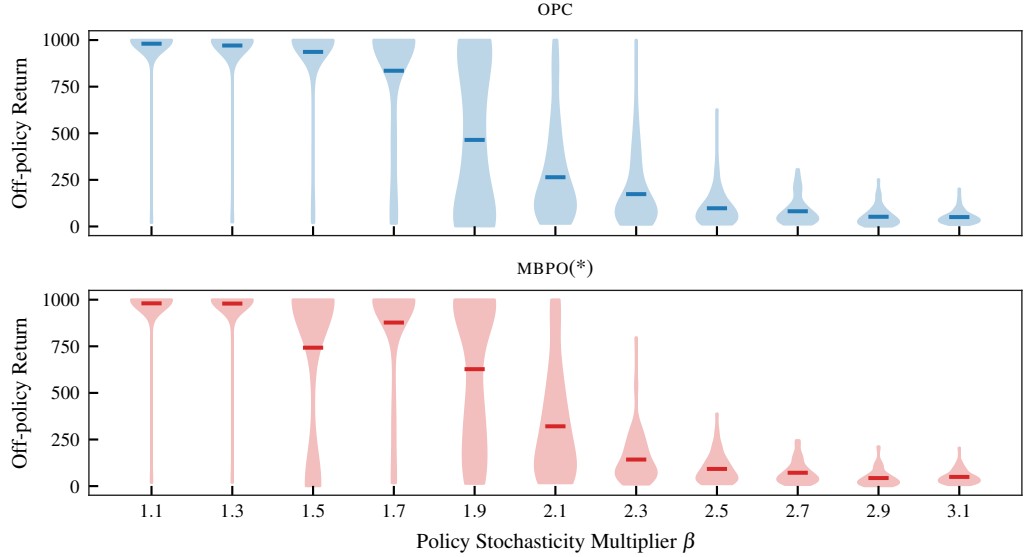

Figure 16: Sample distributions of the return on the CartPole environment (PyBullet) with increasing stochasticity of the behaviour policy $\pi_{\text{rollout}}$ when rolling out with $\tilde{p}_n^{\text{opc}}$/OPC (top) and $\tilde{p}_n^{\text{model}}$/MBPO($\star$) (bottom). The multiplier $\beta$ quantifies the stochasticity of $\pi_{\text{rollout}}$ relative to the reference policy $\pi_n$ (Eq. (92)) such that higher values lead to more 'off-policy-ness'.

## D.5 OFF-POLICY ANALYSIS

In this section, we investigate the robustness of OPC towards 'off-policy-ness' of the reference trajectories that are simulated under the data-generating policy $\pi_n$. To this end, we manually increase the stochasticity of the behavior policy $\pi_{\text{rollout}}$ by a factor $\beta$ such that

$$\mathbb{V}[\pi_{\text{rollout}}(\cdot \mid \mathbf{s})] = \beta^2 \mathbb{V}[\pi_n(\cdot \mid \mathbf{s})], \tag{92}$$

and we keep the mean the same for both policies. Fig. 16 shows the distributions of the returns from 100 rollouts with varying degree of 'off-policy-ness' on the CartPole environment (PyBullet). Note that the data-generating policy consistently leads to the maximum return of 1000. For $\beta$ close to one, both OPC and MBPO($\star$) lead to almost ideal behavior and correctly predict the return in more than 95% of the cases. As we increase the policy's stochasticity (from left to right), the respective performance of the policy decreases until all rollouts terminate prematurely ($\beta \geq 2.3$). Notably, the extent of this effect is almost identical between OPC and MBPO($\star$). We conclude that OPC is, at least empirically, robust towards 'off-policy-ness' of the reference trajectories. Otherwise, we should observe a more pronounced degradation of the policy's performance with increased stochasticity of the behavior policy, since this leads to observing more off-policy state/actions.

---

**Algorithm 3** Original MBPO algorithm

---

1: Initialize policy $\pi_\phi$, predictive model $p_\theta$, environment dataset $\mathcal{D}_{\text{env}}$, model dataset $\mathcal{D}_{\text{model}}$
2: **for** $N$ epochs **do**
3:     Train model $p_\theta$ on $\mathcal{D}_{\text{env}}$ via maximum likelihood
4:     **for** $E$ steps **do**
5:         Take action in environment according to $\pi_\phi$; add to $\mathcal{D}_{\text{env}}$
6:         **for** $M$ model rollouts **do**
7:             Sample $s_t$ uniformly from $\mathcal{D}_{\text{env}}$
8:             Perform $k$-step model rollout starting from $s_t$ using policy $\pi_\phi$; add to $\mathcal{D}_{\text{model}}$
9:         **for** $G$ gradient updates **do**
10:             Update policy parameters on model data: $\phi \leftarrow \phi - \lambda_\pi \hat{\nabla}_\phi J_\pi(\phi, \mathcal{D}_{\text{model}})$

---

**Algorithm 4** Our version of MBPO algorithm, denoted as MBPO($\star$)

---

1: Initialize policy $\pi_\phi$, predictive model $p_\theta$, environment dataset $\mathcal{D}_{\text{env}}$, model dataset $\mathcal{D}_{\text{model}}$
2: **for** $N$ epochs **do**
3:     Train model $p_\theta$ on $\mathcal{D}_{\text{env}}$ via maximum likelihood
4:     **for** $E$ steps **do**
5:         Take action in environment according to $\pi_\phi$; add to $\mathcal{D}_{\text{env}}$
6:     **for** $M$ model rollouts **do**
7:         Sample $s_t$ uniformly from $\mathcal{D}_{\text{env}}$
8:         Perform $k$-step model rollout starting from $s_t$ using policy $\pi_\phi$; add to $\mathcal{D}_{\text{model}}$
9:     **for** $G$ gradient updates **do**
10:         Update policy parameters on model data: $\phi \leftarrow \phi - \lambda_\pi \hat{\nabla}_\phi J_\pi(\phi, \mathcal{D}_{\text{model}})$

---

### D.6   Implementation Changes to Original MBPO

Here, we provide details to the changes we made to the original implementation of MBPO. We denote our variant as MBPO($\star$).

**Episodic Setting**   MBPO updates the policy during rollouts on the real environment. Algorithms 3 and 4 show the original and our version of MBPO, respectively. While the original version might be more sample efficient because it allows for more policy gradient updates based on more recent data, it is not realistic to update the policy during a rollout on the real environment.

**Mix-in of Real Data**   As mentioned in the main paper, one of the key problems with MBRL is that the generated data might exhibit so-called model-bias. Biased data can be problematic for policy optimization as the transition tuples possibly do not come from the true state distribution of the real environment, thus misleading the RL algorithm. In the original implementation of MBPO, the authors use a mix of simulated data from the model as well as observed data from the true environment for policy optimization (`https://github.com/jannerm/mbpo/blob/22cab517c1be7412ec33fbe5c510e018d5813ebf/mbpo/algorithms/mbpo.py#L430`) to alleviate the issue of model bias. While this design choice might help in practice, it adds another hyperparameter and the exact influence of this parameter is difficult to interpret. We therefore refrain from mixing in data from the true environment, but instead only use simulated transition tuples – staying true to the spirit of MBRL.

**Fix Replay Buffer**   The replay buffer that stores the simulated data $\mathcal{D}_{\text{model}}$ for policy optimization is allocated to a fixed size of `rollout length`×`rollout batch size`×`retain epochs`, meaning that per epoch `rollout length`×`rollout batch size` transition tuples are simulated and only the data from the last `retain epochs` are kept in the buffer (`https://github.com/jannerm/mbpo/blob/22cab517c1be7412ec33fbe5c510e018d5813ebf/mbpo/algorithms/mbpo.py#L351`). As the buffer is implemented as a FIFO queue and rollouts might terminate early such that the actual number of simulated transitions is less than `rollout length`, data from older episodes as specified by `retain epochs` are retained in the buffer. We assume that this is not the intended behavior and correct for that in our implementation.

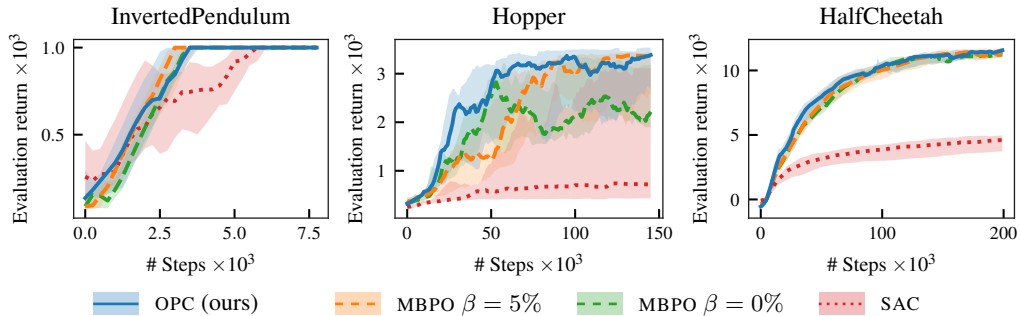

Figure 17: Results based on original implementation Appendix D.7: Comparison of OPC to the model-based MBPO and model-free SAC algorithms. The two MBPO variants differ in terms of the mix-in ratio $\beta$ of real off-policy transitions in the training data – a critical hyperparameter. All model-based approaches outperform SAC in terms of convergence for the high-dimensional tasks. Moreover, on the highly stochastic Hopper environment, OPC outperforms both MBPO variants and does not require additional real off-policy data.

### D.7 RESULTS FOR ORIGINAL MBPO IMPLEMENTATION

The following results were obtained with the original MBPO implementation without the changes described in Appendix D.6. Consequently, the results for OPC in this section also do not include these changes.

**Comparative Evaluation**

We evaluate the original implementation on three continuous control benchmark tasks from the MuJoCo control suite (Todorov et al., 2012). The results for OPC, two variants of MBPO and SAC are presented in Fig. 3. Both OPC and MBPO use a rollout horizon of $H = 10$ to generate the training data. The difference between the two MBPO variants lies in the mix-in ratio $\beta$ of real off-policy transitions to the simulated training data. Especially for highly stochastic environments such as the Hopper, this mix-in ratio is a critical hyperparameter that requires careful tuning (see also Fig. 18). Fig. 3 indicates that OPC is on par with MBPO on both the InvertedPendulum and HalfCheetah environments that exhibit little stochasticity. On the Hopper environment, OPC outperforms both MBPO variants. Note that the mix-in ratio for MBPO is critical for successful learning (the original implementation uses $\beta = 5\%$). OPC on the other hand, does not require any mixed-in real data. SAC learns slower on the more complex Hopper and HalfCheetah environments, re-iterating that model-based approaches are significantly more data-efficient than model-free methods.

**Large Ablation Study – Hopper**

The full study investigates the influence of the following hyperparameters and design choices:

- Rollout length $H$ and total number of simulated transitions $N$.

- Mix-in ratio of real transitions into the training data $\beta$.

- Deterministic or stochastic rollouts (for MBPO): Current state-of-the-art methods in MBRL rely on probabilistic dynamic models to capture both aleatoric and epistemic uncertainty. Accordingly, when rolling out the model, these two sources of uncertainty are accounted for. However, we show that in terms of evaluation return, the stochastic rollouts do not always lead to the best outcome.

- Re-setting the buffer of simulated data after a policy optimization step: We found that re-setting the replay buffer for simulated data after each iteration of Algorithm 3 can have a large influence. In particular, the replay buffer is implemented as a FIFO queue with a fixed size. Hence, if the buffer is not emptied after each iteration, it still contains (simulated) off-policy transitions.

Fig. 18 presents the results of the ablation study. We want to highlight a few core insights:

1. When choosing the best setting for each method, OPC improves MBPO by a large margin (bottom row, right). Generally, for long rollouts $H = 20$ (bottom row), OPC improves MBPO.

2. Across all settings, OPC performs well and is more robust with respect to the choices of hyperparameters (e.g., bottom row left and center, third row left). Only for few exceptions, re-setting the buffer can have detrimental effects (e.g., top right, third row right).

3. Mixing in real transition data can be highly beneficial for MBPO (second row left and center) but it can also have the opposite effect (bottom row).

4. Using deterministic rollouts can be beneficial (third row right and left), detrimental (third row center) or have no influence (bottom row left) for MBPO.

5. It is not clear, if re-setting the buffer after each iteration should overall be recommended or not. This remains an open question left for future research.

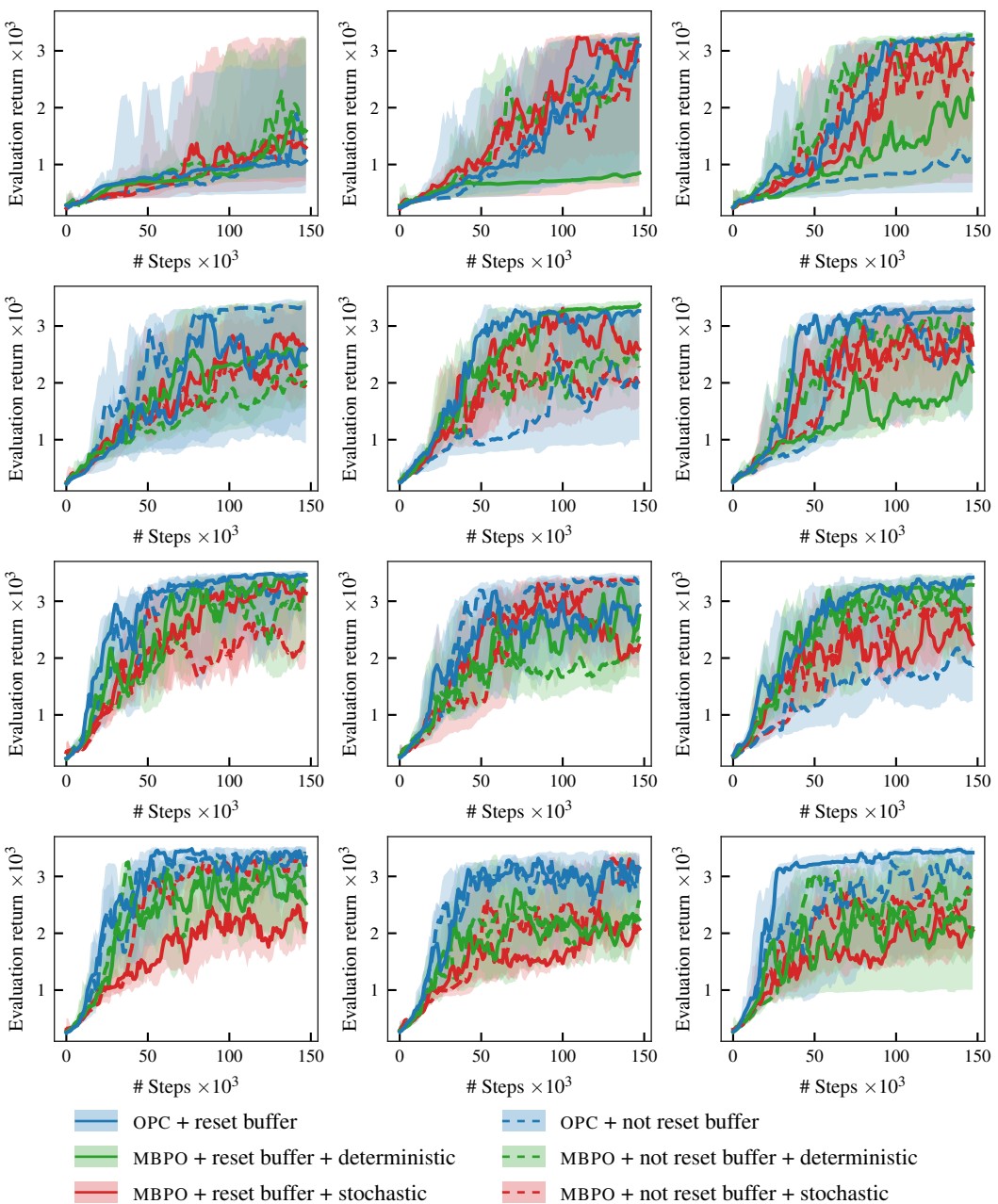

Figure 18: Results based on original implementation Appendix D.7: Large ablation study on the Hopper environment, investigating the influence various hyperparameters and design choices. Mix-in ratio of real data $\beta$ (columns): $0\%, 5\%, 10\%$ from left to right. Rollout length $H$ (rows): $1, 5, 10, 20$ from top to bottom.

# E  CONNECTION BETWEEN MBRL AND ILC

In this section we compare optimization-based or so-called *norm-optimal* ILC (NO-ILC) with MBRL. In particular, we show that under certain assumptions we can reduce the MBRL setting to NO-ILC. This comparison is structured as follows: First, we review the basic assumptions and notations for NO-ILC. While there are many variations on NO-ILC, we will only consider the very basic setting, i.e., linear dynamics, fully observed state and deterministic state evolution. Then, based on the lifted state representation of the problem, we derive the solution to the optimization problem that leads to the input sequence of the next iteration / rollout. Next, we will state the general MBRL problem and pose the simplifications that we need to make in order to be equivalent to the NO-ILC problem. Last, we show that the solution to the reduced MBRL problem is equivalent to the one of NO-ILC.

## E.1  NORM-OPTIMAL ILC

The goal of NO-ILC is to find a sequence of inputs $\mathbf{a} = [\mathbf{a}_0^\top, \ldots, \mathbf{a}_{T-1}^\top]^\top$ with length $T + 1$ such that the outputs $\mathbf{y} = [\mathbf{y}_0^\top, \ldots, \mathbf{y}_T^\top]^\top$ follow a desired output trajectory $\hat{\mathbf{y}} = [\hat{\mathbf{y}}_0^\top, \ldots, \hat{\mathbf{y}}_T^\top]^\top$. In the simplest setting we assume that the system evolves according to the following linear dynamics

$$\mathbf{s}_{t+1} = \mathbf{A}\mathbf{s}_t + \mathbf{B}\mathbf{a}_t + \mathbf{d}_t \tag{93}$$

$$\mathbf{y}_t = \mathbf{C}\mathbf{s}_t, \tag{94}$$

with state $\mathbf{s}_t \in \mathbb{R}^{d_{\mathcal{S}}}$, action $\mathbf{a}_t \in \mathbb{R}^{d_{\mathcal{A}}}$, output $\mathbf{y}_t \in \mathbb{R}^p$ and disturbance $\mathbf{d}_t \in \mathbb{R}^{d_{\mathcal{S}}}$. One of the major assumption in ILC is that the disturbances $\mathbf{d}_t$ are *repetitive*, meaning that the sequence $\mathbf{d} = [\mathbf{d}_0^\top, \ldots, \mathbf{d}_T^\top]^\top$ does not change (or only varies slightly) across multiple rollouts. While these disturbances can be considered to come from some exogenous error source, one can also interpret these as unmodeled effects of the dynamics, e.g., nonlinearities stemming from friction, aerodynamic effects, etc.

For the derivation, let's assume $\mathbf{C} = \mathbf{I}$ so that we're operating on an MDP. Consequently, the goal of ILC is to track a sequence of desired states $\hat{\mathbf{s}}$ instead of outputs $\hat{\mathbf{y}}$. In order to find an optimal input sequence, we minimize the squared 2-norm of a state's deviation from the reference for each time-step $t$ of the sequence, i.e., $\mathbf{e}_t = \hat{\mathbf{s}}_t - \mathbf{s}_t$. As is common in the ILC literature, we make use of the so-called *lifted system* formulation such that we can conveniently write the state evolution for one rollout using a single matrix/vector multiplication. Assuming zero initial state (which can always be done in the linear setting by just shifting the state by a constant offset), we obtain the following formulation

$$\mathbf{s} = \mathbf{F}\mathbf{a} + \mathbf{d}, \quad \text{with} \quad \mathbf{F} = \begin{bmatrix} 0 & \cdots & & \\ \mathbf{B} & 0 & & \\ \mathbf{AB} & \mathbf{B} & 0 & \\ \mathbf{A}^2\mathbf{B} & \mathbf{AB} & \mathbf{B} & \ddots \\ \vdots & & \ddots & \ddots \end{bmatrix} \in \mathbb{R}^{d_{\mathcal{S}}(T+1) \times d_{\mathcal{A}}T}. \tag{95}$$

Thus, we can write the state-error at the $j$-th iteration of ILC in the lifted representation as (recall that the disturbance $\mathbf{d}$ does not change across iterations)

$$\mathbf{e}^{(i)} = \hat{\mathbf{s}} - \mathbf{s}^{(i)} = \hat{\mathbf{s}} - \mathbf{F}\mathbf{a}^{(i)} - \mathbf{d}, \tag{96}$$

$$\mathbf{e}^{(i+1)} = \mathbf{e}^{(i)} - \mathbf{F}(\mathbf{a}^{(i+1)} - \mathbf{a}^{(i)}). \tag{97}$$

Now, the resulting optimization problem then becomes

$$\mathbf{a}_*^{(i+1)} = \underset{\mathbf{a}^{(i+1)}}{\arg\min} \, J\left(\mathbf{a}^{(i+1)}\right) \text{ with } J\left(\mathbf{a}^{(i+1)}\right) = \frac{1}{2}\|\mathbf{e}^{(i+1)}\|^2.$$

In order to be less sensitive to noise and the inherent stochasticity of real-world problems, one typically also adds a regularizing term that penalizes the changes in the input sequence. While this additional penalization term can slow down the learning process it makes it more robust by avoiding overcompensation to the disturbances. The full objective then becomes

$$J\left(\mathbf{a}^{(i+1)}\right) = \frac{1}{2}\|\mathbf{e}^{(i+1)}\|_{\mathbf{M}}^2 + \frac{1}{2}\|\mathbf{a}^{(i+1)} - \mathbf{a}^{(i)}\|_{\mathcal{W}}^2, \tag{98}$$

where we additionally added positive semi-definite cost matrices $\mathbf{M}, \mathcal{W}$ for the respective norms in order to facilitate tuning of the corresponding terms in the objective. Given the regularized cost function we obtain the optimal sequence of inputs at the next iteration as

$$\mathbf{a}_*^{(i+1)} = \mathbf{a}_*^{(i)} + \left(\mathbf{F}^\top \mathbf{M} \mathbf{F} + \mathcal{W}\right)^{-1} \mathbf{F}^\top \mathbf{M} \mathbf{e}^{(i)}. \tag{99}$$

### E.2 MODEL-BASED RL TO NORM-OPTIMAL ILC

**Assumptions**     In order to show the equivalence of MBRL and NO-ILC we need to make some assumptions to the above stated optimization problem.

- No aleatoric uncertainty in the model, i.e., no transition noise $\omega_t = 0 \quad \forall t$.
- Typically in RL we assume the policy to be stationary, however, in this setting we will allow for non-stationary policies that are indexed by time $t$ (the result will not necessarily depend on the state such that we essentially just obtain a feedforward control sequence. To include feedback one can always just combine the feedforward signal with a local controller that tracks the desired state/action trajectory).
- The reward is given as the negative quadratic error of the state w.r.t. a desired state trajectory, $r(\mathbf{s}_t, \mathbf{a}_t) = -\frac{1}{2}\|\hat{\mathbf{s}}_t - \mathbf{s}_t\|^2$
- Typically in RL we have constraints on the policy such that it does not change too much after every iteration, see e.g. TRPO, REPS, etc. While in the mentioned approaches, the policy are often constrained in terms of their parameterization vectors, we constrain it as $\|\pi_t^{(i+1)} - \pi_t^{(i)}\|^2 \le \epsilon_\pi \quad \forall t$.
- Assume that the model is given by $\tilde{f}_t(\mathbf{s}, \mathbf{a}) = \mathbf{A}\mathbf{s} + \mathbf{B}\mathbf{a} + \mathbf{d}_t$, where $\mathbf{A}$ and $\mathbf{B}$ are fixed system matrices and $\mathbf{d}_t$ is a time-dependent offset that we learn.

**The learned dynamics model**     Given state/input pairs of the $i$-th trajectory $\tau^{(i)} = \{(\mathbf{s}_t^{(i)}, \mathbf{a}_t^{(i)})\}_{t=0}^T$ from the true system, we can now improve our dynamics model. In particular, if we minimize the prediction error over $\tau^{(i)}$, we obtain

$$\mathbf{d}_t^{(i)} = \mathbf{s}_{t+1}^{(i)} - \left(\mathbf{A}\mathbf{s}_t^{(i)} + \mathbf{B}\mathbf{a}_t^{(i)}\right) \tag{100}$$

The resulting dynamics for the optimal control problem in Eq. (1) become

$$\tilde{f}_t^{(i)}(\mathbf{s}, \mathbf{a}) = \mathbf{s}_{t+1}^{(i)} + \mathbf{A}(\mathbf{s} - \mathbf{s}_t^{(i)}) + \mathbf{B}(\mathbf{a} - \mathbf{a}_t^{(i)}) \tag{101}$$

which are the error dynamics around the trajectory. Now in the noisy case, taking the last trajectory is not necessarily the best thing one can do. E.g., (Schöllig & D'Andrea, 2009) instead integrate the information of all past trajectories via Kalman-filtering. In the fully observed case, one way to think of this is as low-pass filtering $\mathbf{d}_t$ in order to account for the transition noise $\omega$.

**The resulting MBRL problem**     Now, let's plug in all assumptions into the MBRL problem and have a look at how to solve it.

$$\pi_*^{(i+1)} = \underset{\pi = \{\pi_0, \ldots, \pi_T\}}{\arg\min} \sum_{t=0}^T \frac{1}{2}\|\hat{\mathbf{s}}_t - \mathbf{s}_t\|^2$$
$$\mathbf{s}_{t+1} = \mathbf{s}_{t+1}^{(i)} + \mathbf{A}(\mathbf{s} - \mathbf{s}_t^{(i)}) + \mathbf{B}(\mathbf{a} - \mathbf{a}_t^{(i)}) \tag{102}$$
$$\mathbf{a}_t = \pi_t$$
$$\|\pi_t^{(i)} - \pi_t\|^2 \le \epsilon_\pi \quad \forall t$$

where we have flipped the reward's sign to transform it into a minimization problem. In the presented form, this optimization problem has a well-defined unique solution, however, it is a-priori not clear if the trust-region constraint is active for some time-steps. To facilitate an analytical solution to equation 102, we incorporate the constrain on the policy's stepsize by a soft-constraint such that

$$\pi_*^{(i+1)} = \underset{\pi = \{\pi_0, \ldots, \pi_T\}}{\arg\min} \sum_{t=0}^T \frac{1}{2}\|\hat{\mathbf{s}}_t - \mathbf{s}_t\|^2 + \frac{C_t^\pi}{2}\|\pi_t^{(i)} - \pi_t\|^2$$
$$\mathbf{s}_{t+1} = \mathbf{s}_{t+1}^{(i)} + \mathbf{A}(\mathbf{s} - \mathbf{s}_t^{(i)}) + \mathbf{B}(\mathbf{a} - \mathbf{a}_t^{(i)}) \tag{103}$$
$$\mathbf{a}_t = \pi_t,$$

with $C_t^\pi$ being constants that weight the respective trust-region terms.

Generally, the MBRL problem cannot be solved analytically due to the (possibly highly non-linear, non-differentiable, etc.) reward signal and the need for propagating the (uncertain) dynamics model forwards in time. However, using the simplifying assumptions above, the reduced MBRL problem equation 103 can in fact be solved analytically. We can circumvent the equality constraints by predicting the state trajectory using the error corrected dynamics such that we obtain a lifted dynamics formulation similar to the analysis for ILC,

$$\mathbf{s}^{(i)} = \mathbf{F}\mathbf{a}^{(i)} + \mathbf{d}^{(i)}, \quad \text{with} \quad \mathbf{d}^{(i)} = \mathbf{s}^{(i)} - \mathbf{F}\mathbf{a}^{(i)}, \tag{104}$$

with $\mathbf{s}, \mathbf{a}$ denoting the stacked states and actions of the recorded trajectory. Using this notation, the optimization problem reduces to

$$\pi_*^{(i+1)} = \arg\min_\pi \frac{1}{2}\|\hat{\mathbf{s}} - \mathbf{s}^{(i)}\|^2 + \frac{1}{2}\|\pi^{(i)} - \pi\|_{\mathbf{C}^\pi}^2, \tag{105}$$

with $\mathbf{C}^\pi = \text{diag}[C_0^\pi, \ldots, C_H^\pi]$. By inserting equation 104 into equation 105 we obtain the closed-form solution as

$$\pi_*^{(i+1)} = \left(\mathbf{F}^\top\mathbf{F} + \mathbf{C}^\pi\right)^{-1}\mathbf{F}^\top\left(\hat{\mathbf{s}} - \mathbf{d}^{(i)}\right) = \pi_*^{(i)} + \left(\mathbf{F}^\top\mathbf{F} + \mathbf{C}^\pi\right)^{-1}\mathbf{F}^\top\left(\hat{\mathbf{s}} - \mathbf{s}^{(i)}\right), \tag{106}$$

which, clearly, is equivalent to equation 99 for $\mathbf{M} = \mathbf{I}$ and $\mathcal{W} = \mathbf{C}^\pi$.

### E.3 EXTENSIONS

In the previous section, we analyzed the most basic setting for NO-ILC, i.e., linear dynamics, no transition noise in the dynamics and no state/input constraints. Some of these assumptions can easily be liftd to genearlized the presented framework.

**Nonlinear Dynamics**    Instead of the system being defined by fixed matrices $\mathbf{A}, \mathbf{B}$, we can just linearize a non-linear dynamics model $f(\mathbf{s}, \mathbf{a})$ around the last state/input trajectory such that

$$\mathbf{A}_t^{(i)} = \left.\frac{\partial f}{\partial \mathbf{s}}\right|_{\mathbf{s}_t=\mathbf{s}_t^{(i)}, \mathbf{a}_t=\mathbf{a}_t^{(i)}}, \quad \mathbf{B}_t^{(i)} = \left.\frac{\partial f}{\partial \mathbf{a}}\right|_{\mathbf{s}_t=\mathbf{s}_t^{(i)}, \mathbf{a}_t=\mathbf{a}_t^{(i)}} \tag{107}$$

and the corresponding dynamics matrix for the lifted representation becomes (dropping the superscript notation indicating the iteration for clarity)

$$\mathbf{F} = \begin{bmatrix} 0 & \cdots & & & \\ \mathbf{B}_0 & 0 & & & \\ \mathbf{A}_0\mathbf{B}_0 & \mathbf{B}_1 & 0 & & \\ \mathbf{A}_0\mathbf{A}_1\mathbf{B}_0 & \mathbf{A}_0\mathbf{B}_1 & \mathbf{B}_2 & \ddots & \\ \vdots & & & \ddots & \ddots \end{bmatrix} \in \mathbb{R}^{d_\mathcal{S}(T+1) \times d_\mathcal{A}T}. \tag{108}$$

**State/Input Constraints**    Recall that we could solve the quadratic problems equation 98 and equation 103 analytically because we assumed no explicit inequality constraints on the state and inputs. In practice, however, both states and actions are limited by physical or safety constraints typically given by polytopes. While a closed-form solution is not readily available for this case, one can easily employ any numerical solver to deal with such constraints. See, e.g., `https://en.wikipedia.org/w/index.php?title=Quadratic_programming` for a comprehensive list of available solvers.

**Stochastic Dynamics**    Schöllig & D'Andrea (2009) generalize the ILC setting by considering two separate sources of noise: 1) transition noise in the dynamics, i.e., $\mathbf{s}_{t+1} = f(\mathbf{s}_t, \mathbf{a}_t) + \omega_t^f$ as well as 2) varying disturbances, i.e., $\mathbf{d}^{(i+1)} = \mathbf{d}^{(i)} + \omega^d$. Based on this model, a Kalman-Filter in the *iteration-domain* is developed that estimates the respective random variables and the ILC scheme is adapted to account for the estimated quantities.

