# OpenReview forum: "On-Policy Model Errors in Reinforcement Learning"
_ICLR.cc/2022/Conference — ICLR 2022 Poster_

### Official Review · Reviewer_8GLt · 2021-10-27

**Correctness:** 3
**Technical Novelty And Significance:** 2
**Empirical Novelty And Significance:** 2
**Recommendation:** 6
**Confidence:** 3

**Main Review:**

I thought this was a nice paper. The writing was generally clear and the theoretical justifications were interesting. The connection to ILC was also an interesting addition.
While the idea in itself feels rather incremental, it is saved by the thorough analysis, empirical results and extensive ablations.

I enjoyed Sections 1 and 2 and throught they were well written. Although the jump to Section 3 feels rather abstract. I think Section 2 discusses MBRL too broadly when the paper should really focus on the MBPO-style of MBRL since OPC is limited to this section.

Section 3 proposes an improvement bound that can be decomposed onto on-policy and off-policy model-based improvement. These quanitities feel quite abstract to reason about, since they ultimately rest on the environment, choice of approximator and training protocol. However the paper focuses on compounding model biases which is reasonable.
One slighly strange aspect of the OPC model is its deterministic treatment, such that the aleatoric uncertainty is accessed by sampling different buffer indices, which to me indicates a rollout is deterministic after the choice of index b?

The paper assumes that the model error is constant and time-dependent, but there is no empirical study of how well this assumption holds. Given that the true ground-truth simulator is available, the model bias could be evaluated empirically to see how well this assumption holds in practice.

While OPC appears superior to MBPO* in Figure 3, my understanding this that this MBPO sees no real data from the replay buffer. While Figure 11 shows the standard MPBO where it is comparable if not superior. There appears to be many discrepencies between Fig. 3 and FIg. 11, for example
Are these figures meant to report the same results? (i.e. median vs mean?).
Half Cheetah: OPC mean is noisy and below 10 in Fig. 11, but smooth and above 10 in Fig. 3
Ant: MBPO* mean clearly goes up to 6 in Fig. 11, but in Fig. 3 it is at more like 4.

Moreover ,OPC demonstrates signifiant improvement over MBPO on the pybullet environments. However, I am not sure how to feel about these experments, since the dramatic performance shift speaks to the evaluation assumptions in general rather than the performance of OPC. If the algorithm performance does not translate between simulations, it begs the question how reflective these simulated settings are of these algorithms performance.

The feature representation experiment in Figure 4 is also disconcerting, since the sine cosine features contain an inductive bias that should aid generalization.
Is the idea that the model predicts theta, but uses the features as an intermediate transform for the model, or is the model also predicting the featurespace? In which case, predicting sine features over-complicates prediction since you have the manifold constraint.
Is there not a bug in theta being whitened before the feature transform for MBPO?
I dont see how the OPC addition relates to this issue of feature space, so I don't see how this experiment demonstrates the benefits of OPC. Also, as the code was not submitted I cannot check for bugs.

The paper is missing a reference to
Model Predictive Actor-Critic: Accelerating Robot Skill Acquisition with Deep Reinforcement Learning, Morgan et al. ICRA 2021
which does MBPO with path integral MPC rollouts.


**Summary Of The Paper:**

The paper considers model-based reinforcement learning (MBRL) of the MBPO flavour, where model-free RL methods are accelerated using rollouts of a learned model.
THis paper proposes to alleviate (off-policy) model bias per rollout using the (on-policy) data and calls this approach on-policy correction (OPC).
With this perspective, a comparison is made between MBRL and iterative learning control (ILC).
This method is compared to MBPO on standard mujoco tasks and pybullet versions.

**Summary Of The Review:**

While this looks like it is a solid paper on what is arguably a somewhat incremental idea, I have some issues with the experimental results that need ironing out before I can accept.

---

> ### Author Response · Authors · 2021-11-16
> **Initial Response (1/2)**
>
> Thank you for the positive comments about the paper and suggestions for improvements.
> We have added the additional reference to the related work section to provide a more thorough overview of existing literature.
> In the following, we hope to clarify your concerns regarding the experimental results.
>
>
> ### Are rollouts deterministic after sampling batch index?
>
> Yes, indeed. Once a trajectory index $b$ has been sampled, the subsequent rollout transitions deterministically (we mention this right after Equation 6).
> A direct consequence of this fact is that we emulate the true aleatoric uncertainty of the environment through samples, as discussed also in Section 3.4.
> We have adapted the problem statement to make it clear that we have multiple rollouts that allow us to capture aleatoric uncertainty.
>
> ### Assuming constant and time-dependent model error
>
> Thank you for the suggestion. Please have a look at the [general comment](https://openreview.net/forum?id=81e1aeOt-sd&noteId=ShGFzXEtnpJ): we have added a new experiment to evaluate how well a learned model generalizes with and without OPC. We would also like to point out that the model error is batch-dependent, in addition to time-dependent.
>
> ### Discrepancies between Figure 3 and Figure 11
>
> We thank you very much for observing the discrepancies between Figures 3 and 11.
> After carefully checking our plotting scripts, we found an error in the hyperparameter configurations that were used to generate the plots (also compare to Figure 12 that shows the hyperparameter study).
> We updated Figures 3 and 11 in the paper accordingly and apologize for any misconceptions.
> Further, we added a remark that Figure 11 presents the results in terms of mean $\pm$ standard deviation (Figure 3 depicts median and interquartile ranges).
> The reason is that we obtained the original results from Janner et al. directly in terms of mean and standard deviation, instead of the raw data.
>
> ### Are PyBullet results representative for OPC's performance
>
> We agree that simulated settings are not ideal to evaluate an algorithms true performance.
> For a fair comparison between different algorithms, we have to make certain assumptions after all.
> One of them being that an algorithm's performance should at least somewhat translate between simulations, which in fact is often not evaluated in the literature.
> We showcase that for MBPO($\*$) this assumption does not hold for at least four of the PyBullet environments, albeit not for the CartPole environment (Figure 4, second from the right) and OPC can alleviate this issue.

---

> > ### Author Response · Authors · 2021-11-16
> > **Initial Response (2/2)**
> >
> > ### InvertedPendulum: Angle parameterization
> >
> > *Clarifications regarding prediction*
> >
> > As is common in RL, we do not make any assumptions about the state representation itself and accordingly do not explicitly account for different parameterizations.
> > This means that if the angle is directly observed, the dynamics model also predicts the angle at the next time-step.
> > Correspondingly, if the angle is represented in terms of the sin/cos features, the dynamic model predicts the angle also in feature space.
> >
> > *Do sin/cos features over-complicate the predictions?*
> >
> > How to represent angles for learning is a very interesting question in robotics and computer vision alike, albeit out of scope for the presented paper.
> > Nevertheless, we briefly want to mention the following: Evidence suggests that learning to directly predict the angle (one angle in SO(2) and three angles in SO(3), i.e., the Euler angles) seems to be more difficult compared to other representations (Quaternions, sin/cos) [1], [2].
> > The sin/cos transformation for SO(2), which is also employed by the RoboSchool environment, is explicitly mentioned in the motivating example in [2].
> > We also would like to highlight [this insightful discussion](https://stats.stackexchange.com/questions/218407/encoding-angle-data-for-neural-network) on the Statistics Stack Exchange.
> >
> > [1] Grassmann, Reinhard, and Jessica Burgner-Kahrs. "On the Merits of Joint Space and Orientation Representations in Learning the Forward Kinematics in SE (3)." Robotics: science and systems, 2019
> >
> > [2] Zhou, Yi, et al. "On the continuity of rotation representations in neural networks" CVPR, 2019
> >
> > *How does OPC help here?*
> >
> > In accordance with the main message of the paper, the experiment demonstrates that OPC improves the predictive model in the sense that it generates samples that more closely resemble the true state distribution.
> > When the angle is represented in terms of the sin/cos features, we often observe that without OPC the learned model predicts values that do not adhere to the manifold constraint, e.g., $\cos(\theta) > 1$ and $\tan(\theta) \neq \sin(\theta) / \cos(\theta)$.
> > Accordingly, a part of the simulated data cannot be explained under true state distribution, which can lead to slow convergence or even failure to learn a successful policy.
> > In contrast, OPC improves the predictive model such that the simulated data adequately represents the true state distribution more closely.
> > We added an additional experiment in Appendix C.3 that analyses the accuracy of the two prediction models for the sin/cos feature space.
> >
> > *Is there a bug, e.g., due to whitening of the angle?*
> >
> > We are certain that the results are not mere artifacts of a bug due to the following reason:
> > When only considering the original implementations (first and third figure), we do not manually transform the state representation ourselves.
> > The [RoboSchool implementation](https://github.com/benelot/pybullet-gym/blob/master/pybulletgym/envs/roboschool/robots/pendula/interted_pendulum.py#L50) represents the angle in terms of the sin/cos features and the [PyBullet implementation](https://github.com/benelot/pybullet-gym/blob/master/pybulletgym/envs/mujoco/robots/pendula/inverted_pendulum.py#L47) (as well as MuJoCo) encodes the angle directly.
> > We include the transformed state representations (second and fourth figure) to rule out other effects, e.g., initial state distribution, implementation details, etc., that might explain why MBPO($\*$) fails in some cases.

---

> > > ### Comment · Reviewer_8GLt · 2021-11-29
> > > **Response**
> > >
> > > I wish to thank the reviewers for their clarification and additional work.
> > >
> > > I'm glad you found the plotting bug.
> > >
> > > The pybullet state distribution difference should be clearly stated in the paper. I assumed they were the same as Gym. This is a meaningful environment difference to explain the performance difference, but I suspect there might also be hyperparameter tuning required for the MBPO baseline (since the Janner paper does not look at Pybullet).
> > >
> > > Having read the other reviews and the paper again, I'm happy to raise my score.

---

### Official Review · Reviewer_A4Tf · 2021-10-28

**Correctness:** 4
**Technical Novelty And Significance:** 2
**Empirical Novelty And Significance:** 2
**Recommendation:** 5
**Confidence:** 4

**Main Review:**

Strengths:

1. The idea of the paper is clear.

2. The presence of theoretical results and a fitting empirical side.

Weaknesses:

3. The idea of correcting samples with the model sounds immediate, which means more is required of the analysis:

(a) Since there is a trade-off between model based and sample based, how to implement\calculate this trade-off in the proposed correction?

(b) How do the theoretical results compare to model based and replay based results?

4. I don't understand why include the MujoCo experiments if you can't see any improvement there. The improvements in PyBullet are mostly due to the initial state randomization, how does that fit the general theme of the paper?

5. The comparison is made against SAC and MBPO, but why does the first method proposed (replay buffer based simulation) not compared?

6. Consider adding the reference to the highly related (though not very new) work:

Batch Mode Reinforcement Learning based on the Synthesis of Artificial Trajectories

Raphael Fonteneau, Susan A. Murphy, Louis Wehenkel, and Damien Ernst




**Summary Of The Paper:**

The paper proposes a way to combine observed data with a learned model to mitigate the cost of each choice separately - the observed data is over-fitted to a specific policy while the learned model has approximation errors.
The authors mix the two options using probabilistic corrections based on the learned model applied to observed transitions.
The authors also provide convergence results for their algorithm,

**Summary Of The Review:**

The paper considers a straightforward combination of two methods which does not seem highly significant.

The paper is also somewhat lacking in its analysis and empirical claims.

---

> ### Author Response · Authors · 2021-11-16
> **Initial Response**
>
> Thank your for your comments and providing additional a relevant reference.
> We have updated the paper accordingly to provide a more thorough overview of existing work.
>
> ### How to calculate the model/sample-based trade-off in the correction
>
> We're not sure that we understand the question correctly. The correction is defined in a way that we recover samples when applying the same sequence of actions and only use the model to generalize to new actions, so there is no immediate trade-off. If you are referring to our experiments, then the trade-off is governed by the rollout horizon $H$.
>
> ### How are the theoretical results linked to model-based and replay-based results?
>
> Your question on how the theoretical results link to the purely model-based and replay-based results was shared by other reviewers and we have addressed it in a [separate, shared comment](https://openreview.net/forum?id=81e1aeOt-sd&noteId=ShGFzXEtnpJ).
>
> ### MuJoCo experiments
>
> We include an evaluation on the MuJoCo-benchmarks for the following reasons:
>
> 1. In the main paper, we present results for a slightly modified version of MBPO. To guarantee that our modifications do not break the original algorithm, we show that the modified algorithm retains the strong sample efficiency of MBPO (see also Figure 11 in Appendix C.1 for a comparison to the original algorithm). This reason is especially important, because MBPO($\*$) fails to solve the respective tasks for the PyBullet environments.
> 2. For years, the MuJoCo-benchmarks are the de-facto standard to evaluate reinforcement learning algorithms for continuous control problems. Not including these results would impair the credibility of our experimental evaluation. Nevertheless, just evaluating RL algorithms on MuJoCo-benchmarks is prone to overfitting to that particular evaluation metric, which is why we include evaluations on the PyBullet-benchmarks as well.
>
> ### PyBullet experiments
>
> We agree that one difference between the MuJoCo- and PyBullet-benchmarks lies in the magnitude of the initial state's variance.
> However, it is unclear whether this difference is the actual reason why MBPO($\*$) fails for the PyBullet-benchmarks.
> It might also be the case that the underlying dynamics from the MuJoCo simulator can be represented with sufficient accuracy by the ensemble model, whereas this might not work for the PyBullet simulator.
> Either way, we demonstrate with our results that OPC is able to consistently alleviate this problem.
>
> ### Comparison to replay-buffer based simulation
>
> As the policy optimization algorithm (Line 4 in Alg. 1), we employ SAC, independently of the dynamics model we simulate data with, i.e., pure data-based in Eq(3), only model in Eq(4) or OPC in Eq(6).
> Therefore, if we use the replay buffer as dynamics model in Line 3 (Alg 1), the general model-based RL Algorithm reduces to the model-free (standard) SAC.
> We added a clarifying remark the updated paper.

---

> > ### Comment · Reviewer_A4Tf · 2021-11-22
> > **Thank you for your comments**
> >
> > My questions were addressed.

---

### Official Review · Reviewer_XAKT · 2021-11-02

**Correctness:** 3
**Technical Novelty And Significance:** 3
**Empirical Novelty And Significance:** Not applicable
**Recommendation:** 6
**Confidence:** 4

**Main Review:**

Overall, I'd say this is a relatively thorough examination of a simple but seemingly effective technique. As far as I'm aware, the approach is novel in the context of MBRL, and I can see it being widely used by the community, so I recommend acceptance.

**Issue 1:** Issues with Section 3.1/the bound in Eqn (5).

Clarity: The off-policy model error and on-policy error are just measuring the accuracy of $\hat \eta_{n+1}$ and $\hat \eta_n$. If we assume the data is generated by some $\pi_0$ then both $\pi_{n+1}$ and $\pi_n$ will be "off-policy", so really this relies on $\pi_n$ being the data-generating policy, which isn't stated until a few paragraphs later.

Misleading: The bound also implies that our ability to estimate $\eta_n$ is an important part of policy improvement, but in reality... it isn't. Ultimately the only term we care about is the final performance $\eta_{n+1}$ (i.e. eqn (1)), and we cannot, generally, say that the ability to estimate the performance of the data-generating policy will impact the final performance.

The problem is that the bound is presented as a decomposition of policy improvement, but it's actually a bound on the model's estimate of how much better the current policy is better than the data-generating policy. So for example, if we instead bound $\eta_{n+1}$ (as done in Janner et al.), by adding $\eta_n$ to both sides of the equation, we arrive at $\eta_{n+1} \geq \hat \eta_{n+1} - \text{off-policy model error}$, which does not include any on-policy term.

Of course, this doesn't invalidate the technique, as I would still expect that improving an estimate of the on-policy policy would impact the model's ability to estimate an off-policy policy but I'm not sure this is as fundamental as presented by the authors.

**Issue 2:** Theoretical results are nice but don't say anything comparatively. In other words, it's not clear if the presented bound is improved by including OPC.

**Issue 3:** Issues when using OPC off-policy / suggested analysis.

In any MBRL policy improvement context there will be some amount of off-policy evaluation. If we are estimating the on-policy then this cancellation is convenient, however, when estimating a trajectory which is quite different than the on-policy trajectory, i.e. where $s_t$ and $\hat s^b_t$ are not similar, then $\hat s^b_{t+1} - \hat f(\hat s^b_t, \hat a^b_t)$ seems like just noise added to the estimate $\hat f(s_t,a_t)$, and it seems like this technique would be more harmful than beneficial. This situation also arises naturally when rolling out the off-policy trajectory for a few time steps, as we would expect that it will be quite different from the on-policy trajectory.

An interesting experiment would be to test the model's ability for off-policy evaluation (OPE) over varying levels of "off-policy-ness" (i.e. adding noise to the on-policy/behavior policy). This is a common experiment in OPE literature. I would expect that when near on-policy OPC would be helpful, but it's not clear OPC will be helpful when acting more off-policy.

**Strengths**
* Empirical results are strong/convincing. 10 seeds are used.
* Reproducibility is high. The method is simple/easy to implement.
* The appendix is thorough and includes a lot of additional experiments and details.
* I can see the method being widely adopted by the MBRL community.

**Minor Comments**
- I think Figure 1.b would be clearer if the on-policy and off-policy estimates were separated into two separate graphs (in other words, cutting the figure in half).
- Figure 5 is seemingly used to suggest that higher performance = higher data diversity, but I'm not sure how the authors made that jump. I can see that longer rollouts provides a higher performance, but there's no guarantee that rolling out is necessarily more diverse than the data gathered from the initial policy.
- I find Figure 2 unclear. The meaning of dotted lines is not defined and it's not clear how the figure should be interpreted.
- A very similar technique has been proposed in value-based RL [1] (ironically named off-policy corrections instead) and should be cited.
- I think an interesting experiment would be applying OPC to different models (rather than just the models used by MBPO).
- Do the authors have any insight as to why this technique matters much more on the PyBullet domains?

References:
- [1] Harutyunyan, Anna, et al. "Q ($\lambda $) with Off-Policy Corrections." (2016).


**Summary Of The Paper:**

The authors introduce a model-agnostic technique for improving estimates of on-policy data and show improvements on MuJoCo and PyBullet domains.


**Summary Of The Review:**

Overall, I'd say this is a relatively thorough examination of a simple but seemingly effective technique. As far as I'm aware, the approach is novel in the context of MBRL, and I can see it being widely used by the community, so I recommend acceptance. I have concerns regarding Section 3.1 which is the main motivation of the method, which limits my score.

---

> ### Author Response · Authors · 2021-11-16
> **Initial Response (1/2)**
>
>
> Thank your for your detailed comments, references, and suggestions to further improve the paper. We have adapted the paper to include your feedback and suggestions.
>
> ### Issues with Section 3.1 / the bound in Eq (5).
>
> Thanks a lot for the feedback. You are correct that in our context $\pi_{n} = \pi_{D}$ is the behavior policy and we have changed the text to make it clear that we use an on-policy model, but that in our experiments the policy optimization method also uses off-policy model data. Please let us know if you see further room for improving the clarity.
>
> Your questions regarding the relationship to the bound by MBPO was shared by other reviewers and we have answered them jointly in a
> [shared general comment](https://openreview.net/forum?id=81e1aeOt-sd&noteId=ShGFzXEtnpJ).
>
> ### Issues when using OPC off-policy
>
> As you suggested, we added an experiment to evaluate the model's ability for off-policy evaluation over varying levels of "off-policy-ness" in Appendix C.5. We observe that with increasing level of "off-policy-ness" the policy's performance decreases as we would expect. However, this effect is similarly pronounced for both OPC and MBPO($\*$), which indicates that the off-policy data does not necessarily impair OPC to a larger degree.
>
> In addition to the new experiment in Appendix C.5, we would like to highlight the off-policy analysis for the one-dimensional example in Appendix B.4 (Figure 9).
> Therein, we show depict the policy gradient's error as a function of the "off-policy-ness" (x-axis) for different data-generating policies (y-axis) and varying degrees of model errors (from left to right).
> The respective colors represent the signed gradient error as in Figures 2 and 8.
> First, note that with OPC we retain the true policy gradient when the model is exact (center figures) even when we use off-policy data to correct the predictions.
> Further, the analysis shows that (not surprisingly) that the gradient estimates are getting worse for 1) increased model errors and 2) increased "off-policy-ness".
>
> ### Data-diversity as function of rollout horizon
>
> Similar to MBPO, we are using a branched rollout approach, meaning that we start short rollouts from states that were observed on the true system.
> The benefit of this approach is that we 1) alleviate the issue of compounding model errors and 2) still generate states that are close to the true state distribution.
> Now, consider the following three cases:
>
> - 10 reference trajectories with 1000 states from the true environment, 10'000 (true) total states
> - One reference trajectory with 1000 states from the true environment, 1-step predictions starting from each state 10 times, 10'000 (simulated) total states
> - One reference trajectory with 1000 states from the true environment, 10-step predictions starting from each state once, 10'000 (simulated) total states
>
> Clearly, the first case (10'000 steps from the true environment) is superior to both model-based approaches as the state transitions are guaranteed to be from the true state distribution. However, it is less data-efficient.
> Pictorially speaking, 1-step rollouts lead to more dense but less diverse samples of the true state distribution, whereas 10-step rollouts result in a more diverse but less dense samples.
> With the results from Figure 5 and the argument above, we argue that indeed the reason for higher performance is due the data being more diverse for longer rollout horizons.

---

> > ### Author Response · Authors · 2021-11-16
> > **Initial Response (2/2)**
> >
> > ### Clarifying Figure 2
> >
> > In Section 4.1, we investigate the influence of model errors on policy gradients for an environment with one-dimensional, linear dynamics.
> > Figure 2 depicts the policy gradient's error that is inflicted by a model-mismatch (x-axis) for different policies (y-axis).
> > Note that the optimal policy is at $\theta = -1.0$ and the dotted line depicts the contour line for zero gradient error (we will add this to the figure's legend).
> > Without OPC (left figure), we observe large areas with red background, which indicates that the estimated gradient exhibits the wrong sign even for small model errors.
> > With OPC (right figure), we observe that the red region are drastically reduced.
> > This indicates that the policy gradient points in the right direction most of the time, despite the model error.
> > Further, the right figure generally exhibits lighter shades of blue (for example in the bottom left corner), which indicates that the magnitude of the gradient error is reduced as well.
> >
> > We would also like to highlight Appendix B that contains an in-depth analysis of the one-dimensional example. In particular, it includes an analysis of the gradient errors for varying degree of "off-policy-ness".
> >
> > ### Why does OPC work well for PyBullet environments
> >
> > One difference between the two implementations is that the initial state distribution's variance is significantly larger for the PyBullet environments.
> > Consequently, the state distribution in general exhibits more variety such that the learned transition dynamics require better generalization capabilities compared to the MuJoCo environments.
> > As we also confirm by the new experiment in Appendix C.3, OPC reduces the issue of compounding errors and simulates state trajectories that more closely resemble the true state distribution.
> > We believe that this property of OPC translates to more complex environments and accordingly leads to good performance for the PyBullet environments.

---

> > > ### Comment · Reviewer_XAKT · 2021-11-22
> > > **Still have concerns with main motivation**
> > >
> > > Thanks for the response. I appreciate the added off-policy experiment & clarifications on the figures.
> > >
> > > One point regarding Figure 5 is that the paper states "The results indicate that multi-step rollouts generate more diverse data", however the results only indicate a higher performance. Your response provides an (intuitive) argument for why multi-step rollouts generate more diverse data but then the analysis should state "The results indicate that more diverse data improves performance". In other words, there is some backwards logic here.
> > >
> > > However, the response doesn't address my main concern regarding Section 3.1. I'll leave my comments as a reply to the general response.

---

> > > > ### Author Response · Authors · 2021-11-23
> > > > **Backwards logic in conclusion from Figure 5**
> > > >
> > > > We completely agree and see the backwards logic in the statement.
> > > >
> > > > Unfortunately, the deadline for updating the draft on OpenReview has passed, but we will make sure to update the paper in the final version with your recommended statement: "The results indicate that more diverse data improves performance".

---

### Official Review · Reviewer_xLi5 · 2021-11-02

**Correctness:** 4
**Technical Novelty And Significance:** 3
**Empirical Novelty And Significance:** 3
**Recommendation:** 8
**Confidence:** 3

**Main Review:**

I found this paper well structured and easy to read. I liked in particular the deep analysis of the illustrative example in section 4.1 and appendix B. The authors provide also an extensive ablation on what happens when varying the rollout length.

The intuition I got is that this method is able to retain very low error on the last term of equation (5), while at the same time it should also have a low off-policy model error since a model is being used. However, it is not clear to me how OPC behaves compared for instance to MBPO, when it is evaluated on all three terms on the right side of equation (5). Could it happen that while the on-policy error of OPC is low, the off-policy error becomes much bigger? Could it happen that both errors are low, but because of the model choice in OPC the model policy improvement is extremely lower? Perhaps this is something that can be analytically derived for a very simple MDP like the one in section 4.1

**Summary Of The Paper:**

This paper addresses on-policy errors in model-based Reinforcement Learning. The authors decompose the policy improvement bound in terms that depend on the off-policy and on-policy model errors. They first present two major techniques, one purely based on the replay buffer, which asymptotically obtains zero on-policy error, and one based on a learned model, which leverages off-policy data. Then they propose On-Policy Corrections (OPC), a combination of the two techniques which can use a model and still maintain low on-policy error using the replay buffer. This method has asymptotically (with infinite data) zero on-policy error when using deterministic policies. If the policy is stochastic, the error grows exponentially with the trajectory length T. The authors implement their algorithm on top of MBPO and show experimentally that OPC-MBPO has superior performance in some environments.

**Summary Of The Review:**

This paper combines the use of a replay buffer and a model to lower the on-policy model error. The experiments seem convincing and the method is very well analyzed.

While I am not familiar with the related literature and I might have missed something important (hence I lowered my confidence accordingly), I have the impression that this work is solid and would bring a nice contribution to the conference.

---

> ### Author Response · Authors · 2021-11-16
> **Initial Response**
>
> Thank you for the positive comments about the paper and suggestions for improvements.
>
> You are correct that OPC is designed to reduce the right hand side of (5). Your questions about how our bound relates to the one of MBPO and what we can say about it was shared by other reviewers and we have addressed in a [separate, shared comment](https://openreview.net/forum?id=81e1aeOt-sd&noteId=ShGFzXEtnpJ).

---

> > ### Comment · Reviewer_xLi5 · 2021-11-24
> > **Response to authors**
> >
> > I thank the authors for the clarification. I read the other reviews and the authors' responses and I am willing to confirm my score for this submission

---

### Author Response · Authors · 2021-11-16
**General comment (1/2)**

# General Comment

We thank the reviewers for their time and the valuable feedback to our paper. In addition to the individual responses, we would like to point out a key aspect that we think are relevant to all reviews. In particular, a common question was whether OPC actually tightens the overall policy improvement bound (eq 5) compared to pure model-based or replay-based methods.

## Relationship to MBPO

MBPO presents a lower-bound on environment performance, while we present a bound on policy improvement. However, they are closely related if one sets $\pi_{D} = \pi_n$ as the behaviour policy (we have changed the paper to make it more clear that this is our setting). We would like to present an alternative derivation of MBPO's performance lower-bound to the one in Appendix. A by Janner et al. that highlights similarities to our method:
$$
\eta[\pi] \geq \hat{\eta}[\pi] - | \eta[\pi] - \eta[\pi_{D}] | - | \eta[\pi_{D}] - \hat{\eta}[\pi] |
$$
$$
\geq \hat{\eta}[\pi] - | \eta[\pi] - \eta[\pi_{D}] | - | \hat{\eta}[\pi_{D}] - \hat{\eta}[\pi] | -  | \eta[\pi_{D}] - \hat{\eta}[\pi_{D}] |
$$
$$
\geq \hat{\eta}[\pi] - 2  C \epsilon_\pi - | \eta[\pi_{D}] - \hat{\eta}[\pi_{D}] |
$$
$$
\geq \hat{\eta}[\pi] - 2  C \epsilon_\pi - C \gamma \epsilon_{m}
$$
$$
= \hat{\eta}[\pi] - C(2 \epsilon_\pi + \gamma \epsilon_{m})
$$
where $C = 2 r_{max} / (1 - \gamma)^2$, which is the same as the bound by Janner et al. if we group terms. From this perspective, the bound consists of the quality of the on-policy performance estimate with our model under $\pi_{D}$ ($\eta[\pi_{D}] - \hat{\eta}[\pi_{D}]$, which is bounded by $\epsilon_{m}$) together with two off-policy terms that are both bounded by the KL with $\epsilon_\pi$. From that perspective, any changes to the model that improve the on-policy model error would also directly improve the bound by Janner et al.

Since we can only make a statement about performance relative to our behavior policy $\pi_{n}=\pi_{D}$, we found it more informative to present bounds on the policy improvement relative to that policy, rather than lower-bounds on performance on the true environment. The main disadvantage of a lower bound relative to a policy-improvement bound is that from the form it is not apparent how tight the inequality is. In contrast, from our policy improvement bound it follows directly that if the model is perfect $p = \hat{p}$ then optimizing performance on $\hat{p}$ directly translates to improvement on the true environment $p$. Similarly, it shows that if $\pi_{n} = \pi_{n+1}$ the gap is proportional only to the on-policy model error.

We would also like to highlight appendix B. of the paper where we provide additional insights into our toy example. In particular, one can see that with OPC the gap between model-performance and environment-performance is tight on-policy, while without OPC there is a gap corresponding to the model error.

---

> ### Author Response · Authors · 2021-11-16
> **General comment (2/2)**
>
>
> ## Analysis of error terms
>
> Unfortunately, the effect of OPC off-policy is difficult to analyze, since it depends on the generalization properties of the model beyond the data it was trained on. For the special case when the model is perfect ($p = \tilde{p}$), we know that OPC is equivalent to classical MBRL up to the transition noise, which comes from the data in OPC and is thus on-policy. Beyond that, as mentioned above existing MBPO also contain an on-policy model error term (i.e., supervised learning loss) and bound any off-policy contributions via the KL divergence. These bounds hold independently of OPC (though not for deterministic policies, as the KL becomes infinite). Thus one could also motivate reducing the on-policy model error through the lower bound by Janner et al.
>
> Given that off-policy is difficult to analyze without additional assumptions, we focus on the on-policy model error term in our derivations. There, we know that using on-policy data from environment rollouts is the gold-standard and multiple samples are only needed for capture transition noise. For model-based methods, the estimation quality depends on the on-policy model error. We know from Lemma 1 in the paper that, for deterministic policies, combining the model with OPC retains the qualities of the replay buffer and it decays smoothly as the policy becomes more stochastic (Theorem 1).
>
> The reason for the errors is that, as the policy becomes more stochastic, we rely increasingly on the model to generalize from one action to another (even though from the on-policy state-action distribution). We have added a new section in the appendix that evaluates the impact of model errors in OPC for the one-step predictions in appendix A. 5. There, we show that the one-step prediction error under the (integrated) OPC model is
> $$
> \mathrm{min} \Bigg(
>         \mathcal{O} \Big( \mathrm{trace} \, \mathrm{Var} [ \pi(\cdot \mid \hat{\mathbf{s}}\_{0}) ] \Big),
>         \mathcal{O} \Big( \int \| f(\hat{\mathbf{s}}\_{0}, \mathbf{a}) - \tilde{f}(\hat{\mathbf{s}}\_{0}, \mathbf{a}) \|^2 d \pi(\mathbf{a} \mid \hat{\mathbf{s}}\_{0})
>         \Big)
>         \Bigg)
> $$
>
> That is, while the prediction error decreases as the policy variance decreases, the same holds true as the model error decreases. While one should be cautious about comparing (loose) upper bounds, this hints at that OPC manages to improve the on-policy error over purely model-based settings.
>
> ## Empirical Study
>
> In addition to the extended theoretical result, we empirically evaluate the error terms from Eq. (5) on the CartPole environment in Appendix C.4.
> There, we show how the on- and off-policy model errors evolve over the course of policy optimization for both OPC and MBPO($\*$).
> In all but one iteration, the on-policy model error is smaller for OPC, supporting our theoretical results.
> Notably, the same holds true for the off-policy model error, indicating that OPC is also robust with respect to off-policy reference trajectories.
> The effect of off-policy data on OPC is further investigated with a new experiment in Appendix C.5.

---

> ### Comment · Reviewer_XAKT · 2021-11-22
> **This bound doesn't need on-policy terms**
>
> My concern with 3.1 and the bound in Eqn (5) was the performance of the policy did not depend on any on-policy term. This newly presented bound **also** does not actually depend on an on-policy term, because the on-policy term only exists through algebraic manipulation.
>
> For example, if we swap $\pi_D$ in the derivation to any other policy, the exact same bound holds, then why does $|\eta[\pi_D] - \hat \eta[\pi_D]|$ matter? Choosing $\pi_D = \pi$ gives a strictly tighter bound as $\epsilon_\pi=0$. We can always take a lower bound and subtract any positive term, but this does not mean that the positive term impacts the real lower bound. So, I do not see how we can conclude that the on-policy model error necessarily relates to the performance gap between $\eta[\pi]$ and $\hat \eta[\pi]$.

---

> > ### Author Response · Authors · 2021-11-23
> > **On-policy errors**
> >
> > Thank you for clarifying your comment. We completely agree that these bounds are generally not tight so that one needs to be careful about drawing broad conclusions from them.
> >
> > That being said, independently of how we write the bound, for $\pi = \pi_D$ the on-policy model error determines the tightness of the bound. This can also be seen in our toy example in Figure 7, where even in the presence of model errors the bound under OPC is tight on-policy (black-dashed line), so that the green-dashed curve is equal to the blue solid curve. In contrast, the purely model-based estimate (red line) is significantly off.
> >
> > More generally, the on-policy error also matters in practice, since we only collect data from the real environment under $\pi_D$. In MBPO, they use that the model is trained in a supervised-learning setting on on-policy data to bound the on-policy model error term by $\epsilon_m$. In contrast, off-policy predictions (generalization) would require further assumptions on the model.
> >
> > Do you have suggestions on how we could highlight this further in the paper draft?

---

### Decision · Program_Chairs · 2022-01-20

**Decision:**

Accept (Poster)

**Comment:**

This paper presents a study of on-policy data in the context of model-based reinforcement learning and proposes a way to ameliorate the resulting model errors.

This is a timely and interesting contribution, and all reviewers agree on the quality of the manuscript.
Please incorporate all the remaining feedback from the reviewers.

Minor comment: There might be interesting points of contact between this work and the concept of objective mismatch (https://arxiv.org/abs/2002.04523)